# Online Control of Unknown Time-Varying Dynamical Systems

**Edgar Minasyan**
Princeton University, Google AI Princeton
`minasyan@princeton.edu`

**Paula Gradu**
UC Berkeley
`pgradu@berkeley.edu`

**Max Simchowitz**
MIT
`msimchow@mit.edu`

**Elad Hazan**
Princeton University, Google AI Princeton
`ehazan@princeton.edu`

## Abstract

We study online control of time-varying linear systems with unknown dynamics in the nonstochastic control model. At a high level, we demonstrate that this setting is *qualitatively harder* than that of either unknown time-invariant or known time-varying dynamics, and complement our negative results with algorithmic upper bounds in regimes where sublinear regret is possible. More specifically, we study regret bounds with respect to common classes of policies: Disturbance Action (SLS), Disturbance Response (Youla), and linear feedback policies. While these three classes are essentially equivalent for LTI systems, we demonstrate that these equivalences break down for time-varying systems.

We prove a lower bound that no algorithm can obtain sublinear regret with respect to the first two classes unless a certain measure of system variability also scales sublinearly in the horizon. Furthermore, we show that offline planning over the state linear feedback policies is NP-hard, suggesting hardness of the online learning problem.

On the positive side, we give an efficient algorithm that attains a sublinear regret bound against the class of Disturbance Response policies up to the aforementioned system variability term. In fact, our algorithm enjoys sublinear *adaptive* regret bounds, which is a strictly stronger metric than standard regret and is more appropriate for time-varying systems. We sketch extensions to Disturbance Action policies and partial observation, and propose an inefficient algorithm for regret against linear state feedback policies.

## 1 Introduction

The control of linear time-invariant (LTI) dynamical systems is well-studied and understood. This includes classical methods from optimal control such as LQR and LQG, as well as robust $H_\infty$ control. Recent advances study regret minimization and statistical complexity for online linear control, in both stochastic and adversarial perturbation models. Despite this progress, rigorous mathematical guarantees for nonlinear control remain elusive: nonlinear control is both statistically and computationally intractable in general.

In the face of these limitations, recent research has begun to study the rich continuum of settings which lie between LTI systems and generic nonlinear ones. The hope is to provide efficient and robust algorithms to solve the most general control problems that are tractable, and at the same time, to characterize precisely at which degree of nonlinearity no further progress can be made.

35th Conference on Neural Information Processing Systems (NeurIPS 2021), virtual.

This paper studies the control of linear, time-varying (LTV) dynamical systems as one such point along this continuum. This is because the first-order Taylor approximation to the dynamics of any smooth nonlinear system about a given trajectory is an LTV system. These approximations are widely popular because they allow for efficient planning, as demonstrated by the success of iLQR and iLQG methods for nonlinear receding horizon control. We study online control of discrete-time LTV systems, with dynamics and time-varying costs

$$x_{t+1} = A_t x_t + B_t u_t + w_t, \quad c_t(x_t, u_t) : (x, u) \to \mathbb{R}. \tag{1.1}$$

Above, $x_t$ is the state of the system, $u_t$ the control input, $w_t$ the disturbances, and $A_t, B_t$ the system matrices. Our results extend naturally to partial-state observation, where the controller observes linear projections of the state $y_t = C_t x_t$. We focus on the challenges introduced when the system matrices $A_t, B_t$ and perturbations $w_t$ are not known to the learner in advance, and can only be determined by live interaction with the changing systems.

In this setting, we find that the overall change in system dynamics across time characterizes the difficulty of controlling the unknown LTV system. We define a measure, called system variability, which quantifies this. We show both statistical and computational lower bounds as well as algorithmic upper bounds in terms of the system variabilility. Surprisingly, system variability does not impede the complexity of control when the dynamics are known [16].

## 1.1 Contributions

We consider the recently popularized nonstochastic model of online control, and study regret bounds with respect to common classes of policies: Disturbance Action (DAC/SLS [44]), Disturbance Response (DRC/Youla [46]), and linear feedback policies. Planning over the third class of feedback policies in LTI systems admits efficient convex relaxations via the the first two parametrizations, DAC and DRC. This insight has been the cornerstone of both robust [49, 44] and online [3, 39] control.

**Separation of parametrizations.** For linear time-varying systems, however, we find that equivalences between linear feedback, DAC and DRC fail to hold: we show that there are cases where any one of the three parametrizations exhibits strictly better control performance than the other two.

**Regret against convex parametrizations.** Our first set of results pertain to DAC and DRC parametrizations, which are convex and admit efficient optimization. We demonstrate that no algorithm can obtain sublinear regret with respect to these classes when faced with *unknown*, LTV dynamics unless a certain measure of *system variability* also scales sublinearly in the horizon. This is true even under full observation, controllable dynamics, and fixed control cost. This finding is in direct contrast to recent work which shows sublinear regret *is* attainable over LTV system dynamics if they are known [16].

We give an efficient algorithm that attains sublinear regret against these policy classes up to an additive penalty for the aforementioned system variability term found in our lower bound. When the system variability is sufficiently small, our algorithm recovers state-of-the-art results for unknown LTI system dynamics up to logarithmic factors.

In fact, our algorithm enjoys sublinear *adaptive* regret [21], a strictly stronger metric than standard regret which is more appropriate for time-varying systems. We also show that the stronger notion of adaptivity called strongly adaptive regret [11] is out of reach in the partial information setting.

**Regret against state feedback.** Finally, we consider the class of state feedback policies, which are linear feedback with memory length one. We show that full-information optimization over state feedback policies is computationally hard. This suggests that obtaining sublinear regret relative to these policies may be computationally prohibitive, though does not entirely rule out the possibility of improper learning. However, improper learning cannot be done via the DRC or DAC relaxations in light of our policy class separation results. Finally, we include an inefficient algorithm which attains sublinear (albeit nonparametric-rate) regret against state feedback control policies.

## Paper Structure

Discussion of relevant literature and relation to our work can be found in Section 1.2. In Section 2, we formally introduce the setting of LTV nonstochastic control, the policy classes we study and our key result regarding their non-equivalence in the LTV setting (Theorem 2.1). Motivated by this

non-equivalence, the remainder of the paper is split into the study of convex policies (Section 3) and of state feedback policies (Section 4). In Section 3, we show that regret against the DAC and DRC classes cannot be sublinear unless the metric system variability (Definition 3.1) itself is sublinear (Theorem 3.1), and also propose Algorithm 2 whose adaptive regret scales at the rate of our lower bound plus a $T^{2/3}$ term (Theorem 3.4). On the other hand, in Section 4 we show sublinear regret against state feedback policies is technically possible (Theorem 4.1) with a computationally inefficient algorithm, but also provide a computational lower bound (Theorem 4.2) *for planning* which reveals significant difficulties imposed by the LTV dynamics in this scenario as well. Finally, in Section 5 we pose several future directions, concerning both questions in LTV control, as well as the extension to nonlinear control.

## 1.2  Related Work

Our study of LTV systems is motivated by the widespread practical popularity of iterative linearization for nonlinear receding horizon control; e.g., the iLQR [40], iLC [29], and iLQG [41] algorithms. Recent research has further demonstrated that near-optimal solutions to LTV approximations of dynamics confer stability guarantees onto the original nonlinear system of interest [45].

**Low-Regret Control:** We study algorithms which enjoy sublinear regret for online control of LTV systems; that is, whose performance tracks a given benchmark of policies up to a term which is vanishing relative to the problem horizon. [1] initiated the study of online control under the regret benchmark by introducing the online LQR problem: where a learner is faced with an unknown LTI system, fixed costs and i.i.d. Gaussian disturbances, and must attain performance relative to the LQR-optimal policy. Bounds for this setting were later improved and refined in [12, 26, 10, 38], and extended to partial-state observation in [25, 24]. Our work instead adopts the *nonstochastic control setting* [3], where the adversarially chosen (i.e. non-Gaussian) noise is considered to model the drift terms that arise in linearizations of nonlinear terms, and where costs may vary with time. [3] consider known system dynamics, later extended to unknown systems under both full-state [20] and partial-state observation [39, 37]. The study of nonstochastic control of known LTV dynamics was taken up in [16], with parallel work by [32] considering known LTV dynamics under stochastic noise.

**Unknown LTV dynamics:** Our work is the first to consider online (low-regret) control of unknown LTV systems in *any* model. There is, however, a rich body of classical work on adaptive control of LTV systems [28, 42]. These guarantees focus more heavily on error sensitivity and stability; they only permit dynamical recovery up to error that scales linearly in system noise, and thus guarantee only (vacuous) linear-in-horizon regret. More recent work has studied identification (but not online control) of an important LTV class called switching systems [31, 35].

**Online Convex Optimization:** We make extensive use of techniques from the field of online convex optimization [9, 18]. Most relevant to our work is the literature on adapting to changing environments in online learning, which starts from the works of [22, 6]. The notion of adaptive regret was introduced in [21] and significantly studied since as a metric for adaptive learning in OCO [2, 47]. [11] proposed to strengthen adaptive regret and the stronger metric has been shown to imply results over dynamic regret [48].

**Recent nonlinear control literature:** Recent research has also studied provably guarantees in various complementary (but incomparable) models: planning regret in nonlinear control [4], adaptive nonlinear control under linearly-parameterized uncertainty [5], online model-based control with access to non-convex planning oracles [23], and control with nonlinear observation models [27, 13].

## 2  Problem Setting

We study control of a linear time-varying (LTV) system Eq. (1.1) with state $x_t \in \mathbb{R}^{d_x}$, control input $u_t \in \mathbb{R}^{d_u}$ chosen by the learner, and the external disturbance $w_t \in \mathbb{R}^{d_x}$ chosen by Nature. The system is characterized by time-varying matrices $A_t \in \mathbb{R}^{d_x \times d_x}$, $B_t \in \mathbb{R}^{d_x \times d_u}$. For simplicity, the initial state is $x_1 = 0$. At each time $t$, *oblivious*[1] adversary picks the system matrices $(A_t, B_t)$, disturbances $w_t$ and cost functions $c_t : \mathbb{R}^{d_x} \times \mathbb{R}^{d_u} \to \mathbb{R}$. The dynamics $(A_t, B_t)$ are *unknown* to the learner: one observes only the next state $x_{t+1}$ and current cost $c_t(\cdot, \cdot)$ after playing control $u_t$.

---

[1]An oblivious adversary chooses the matrices, costs and perturbations prior to the control trajectory.

**Adaptive Regret.** The goal of the learner is to minimize regret w.r.t. a policy class $\Pi$, i.e. the difference between the cumulative cost of the learner and the best policy $\pi^\star \in \Pi$ in hindsight. Formally, the regret of an algorithm $\mathcal{A}$ with control inputs $u_{1:T}$ and corresponding states $x_{1:T}$, over an interval $I = [r, s] \subseteq [T]$, is defined as

$$\text{Regret}_I(\mathcal{A}; \Pi) = \sum_{t \in I} c_t(x_t, u_t) - \inf_{\pi \in \Pi} \sum_{t \in I} c_t(x_t^\pi, u_t^\pi) . \tag{2.1}$$

Here $u_t^\pi, x_t^\pi$ indicate the control input and the corresponding state when following policy $\pi$. For a randomized algorithm $\mathcal{A}$, we consider the expected regret. In this work, we focus on designing control algorithms that minimize *adaptive* regret, i.e. guarantee a low regret relative to the best-in-hindsight policy $\pi_I^\star \in \Pi$ on *any* interval $I \subseteq [T]$. This performance metric of adaptive regret is more suitable for control over LTV dynamics given its agility to compete against different local optimal policies $\pi_I^\star \in \Pi$ at different times [16].

**Key objects.** A central object in our study is the sequence of Nature's x's $x_{1:T}^{\text{nat}}$ that arises from playing zero control input $u_t = 0$ at each $t \in [T]$, i.e. $x_{t+1}^{\text{nat}} = A_t x_t^{\text{nat}} + w_t$ [39]. define the following operators for all $t$,

$$\Phi_t^{[0]} = \mathbb{I}, \quad \forall h \in [1, t), \ \Phi_t^{[h]} = \prod_{k=t}^{t-h+1} A_k, \quad \forall i \in [0, t), \ G_t^{[i]} = \Phi_t^{[i]} B_{t-i},$$

where the matrix product $\prod_s^r$ with $s \geq r$ is taken in the indicated order $k = s, \ldots, r$. the following identities give an alternative representation for the Nature's x's $x_t^{\text{nat}}$ and state $x_t$ with control input $u_t$ in terms of the *Markov operator* at time $t$, $G_t = [G_t^{[i]}]_{i \geq 0}$:

$$x_{t+1}^{\text{nat}} = \sum_{i=0}^{t-1} \Phi_t^{[i]} w_{t-i}, \quad x_{t+1} = x_{t+1}^{\text{nat}} + \sum_{i=0}^{t-1} G_t^{[i]} u_{t-i} .$$

These operators and the alternative representation capture the dynamics by decoupling the disturbance and the control action effects.

**Assumptions.** We make the three basic assumptions: we require from (i) the disturbances to not blow up the system with no control input, (ii) the system to have decaying effect over time, and (iii) the costs to be well-behaved and admit efficient optimization. Formally, these assumptions are:

**Assumption 1.** For all $t \in [T]$, assume $\|x_t^{\text{nat}}\| \leq R_{\text{nat}}$.

**Assumption 2.** Assume there exist $R_G \geq 1$ and $\rho \in (0, 1)$ s.t. for any $h \geq 0$ and for all $t \in [T]$

$$\sum_{i \geq h} \|G_t^{[i]}\|_{\text{op}} \leq R_G \cdot \rho^h := \psi(h) .$$

**Assumption 3.** Assume the costs $c_t : \mathbb{R}^{d_x} \times \mathbb{R}^{d_u} \to \mathbb{R}$ are general convex functions that satisfy the conditions $0 \leq c_t(x, u) \leq L \max\{1, \|x\|^2 + \|u\|^2\}$, and $\|\nabla c_t(x, u)\| \leq L \max\{1, \|x\| + \|u\|\}$ for some constant $L > 0$, where $\nabla$ denotes any subgradient [7].

The conditions in Assumption 3 allow for functions whose values and gradient grow as quickly as quadratics (e.g. the costs in LQR) , and the $\max\{1, \cdot\}$ term ensures the inclusion of standard bounded and Lipschitz functions as well. Assumptions 1 and 2 arise from the assumption our LTV system is *open-loop stable*; Appendix A.2 extends to the case where a nominal stabilizing controller is known, as in prior work [3, 39]. While these two assumptions may seem unnatural at first, they can be derived from the basic conditions of disturbance norm bound and sequential stability.

**Lemma 2.1.** *Suppose that there exist $C_1 \geq 1, \rho_1 \in (0, 1)$ such that $\|\Phi_t^{[h]}\|_{\text{op}} \leq C_1 \rho_1^h$ for any $h \geq 0$ and all $t \in [T]$, and suppose that $\max_t \|w_t\| \leq R_w$. Then, Assumption 1 holds with $R_{\text{nat}} = \frac{C_1}{1-\rho_1} R_w$, and Assumption 2 holds with $\rho = \rho_1$ and $R_G = \max\{1, \max_t \|B_t\|_{\text{op}} \cdot \frac{C_1}{1-\rho_1}\}$.*

Note that Assumption 2 implies that $\|G_t\|_{\ell_1, \text{op}} = \sum_{i \geq 0} \|G_t^{[i]}\|_{\text{op}} \leq R_G$. It also suggests that for a sufficiently large $h$ the effect of iterations before $t - h$ are negligible at round $t$. This prompts introducing a truncated Markov operator: denote $\bar{G}_t^h = [G_t^{[i]}]_{i < h}$ to be the $h$-truncation of the true Markov operator $G_t$. It follows that their difference is $\|\bar{G}_t^h - G_t\|_{\ell_1, \text{op}} = \sum_{i \geq h} \|G_t^{[i]}\|_{\text{op}} \leq \psi(h)$ negligible in operator norm for a sufficiently large $h$. Define the bounded set of $\bar{h}$-truncated Markov operators to be $\mathcal{G}(h, R_G) = \{G = [G^{[i]}]_{0 \leq i < h} : \|G\|_{\ell_1, \text{op}} \leq R_G\}$ with $\bar{G}_t^h \in \mathcal{G}(h, R_G)$ for all $t$.

## 2.1 Benchmarks and Policy Classes

The performance of an algorithm, measured by Eq. (2.1), directly depends on the policy class $\Pi$ that is chosen as a benchmark to compete against. In this work, we consider the following three policy classes: DRC, DAC, and linear feedback. DRC parameterizes control inputs in terms of Nature's x's $x_t^{\mathrm{nat}}$, DAC does so in terms of the disturbances $w_t$ and linear feedback in terms of the states $x_t$. We express all three in terms of a length-$m$ parameter $M = [M^{[i]}]_{i<m}$ in a bounded ball $\mathcal{M}(m, R_M)$:

$$\mathcal{M}(m, R_M) = \{(M^{[0]}, \ldots, M^{[m-1]}) : \sum_{i=0}^{m-1} \|M^{[i]}\|_{\mathrm{op}} \le R_M\} .$$

**Definition 2.1** (DRC policy class)**.** A DRC control policy $\pi_{\mathrm{drc}}^M$ of length $m$ is given by $u_t^M = \sum_{i=0}^{m-1} M^{[i]} x_{t-i}^{\mathrm{nat}}$ where $M = [M^{[i]}]_{i<m}$ is the parameter of the policy. Define the bounded DRC policy class as $\Pi_{\mathrm{drc}}(m, R_M) = \{\pi_{\mathrm{drc}}^M : M \in \mathcal{M}(m, R_M)\}$.

**Definition 2.2** (DAC policy class)**.** A DAC control policy $\pi_{\mathrm{dac}}^M$ of length $m$ is given by $u_t^M = \sum_{i=0}^{m-1} M^{[i]} w_{t-i}$ where $M = [M^{[i]}]_{i<m}$ is the parameter of the policy. Define the bounded DAC policy class as $\Pi_{\mathrm{dac}}(m, R_M) = \{\pi_{\mathrm{dac}}^M : M \in \mathcal{M}(m, R_M)\}$.

**Definition 2.3** (Feedback policy class)**.** A feedback control policy $\pi_{\mathrm{feed}}^M$ of length $m$ is given by $u_t^M = \sum_{i=0}^{m-1} M^{[i]} x_{t-i}$ where $M = [M^{[i]}]_{i<m}$ is the parameter of the policy. Define the bounded feedback policy class as $\Pi_{\mathrm{feed}}(m, R_M) = \{\pi_{\mathrm{feed}}^M : M \in \mathcal{M}(m, R_M)\}$. In the special case of memory $m = 1$, denote the *state feedback* policy class as $\Pi_{\mathrm{state}} = \Pi_{\mathrm{feed}}(m = 1)$.

**Convexity.** Both the DRC and DAC policy classes are *convex parametrizations*: a policy $\pi \in \Pi_{\mathrm{drc}} \cup \Pi_{\mathrm{dac}}$ outputs controls $u_t$ that are linear in the *policy-independent* sequences $x_{1:T}^{\mathrm{nat}}$ and $w_{1:T}$, and thus the mapping from parameter $M$ to resulting states and inputs (resp. costs) is affine (resp. convex). Hence, we refer to these as the *convex classes*. In contrast, feedback policies select inputs based on *policy-dependent* states, and are therefore non-convex [15].

We drop the arguments $m, R_M$ when they are clear from the context. The state feedback policies $\Pi_{\mathrm{state}}$ encompass the $\mathcal{H}_2$ and $\mathcal{H}_\infty$ optimal control laws under full observation. For LTI systems, DRC and DAC are equivalent [46, 44] and approximate all linear feedback policies to arbitrarily high precision [3, 39]. However, we show that these relationships between the classes break down for LTV systems: there exist scenarios where any one of the three classes strictly outperforms the other two.

**Theorem 2.1** (Informal)**.** *For each class $\Pi$ in $\{\Pi_{\mathrm{drc}}, \Pi_{\mathrm{dac}}, \Pi_{\mathrm{feed}}\}$ there exists a sequence of well-behaved $(A_t, B_t, w_t, c_t)$ such that a policy $\pi^\star \in \Pi$ suffers $0$ cumulative cost, but each of the other two classes $\Pi' \in \{\Pi_{\mathrm{drc}}, \Pi_{\mathrm{dac}}, \Pi_{\mathrm{feed}}\} \setminus \Pi$ suffers $\Omega(T)$ cost on all their constituent policies $\pi \in \Pi'$.*

The formal theorem that includes the definition of a well-behaved instance sequence and the final statement dependence on $m, R_M$ along with its proof can be found in Appendix F.1.

**Notation.** The norm $\|\cdot\|$ refers to Euclidean norm unless otherwise stated, $[n]$ is used as a shorthand for $[1, n]$, $T$ is used as a subscript shorthand for $[T]$. The asymptotic notation $\mathcal{O}(\cdot), \Omega(\cdot)$ suppress all terms independent of $T$, $\widetilde{\mathcal{O}}(\cdot)$ additionally suppresses terms logarithmic in $T$. We define $\widetilde{\mathcal{O}}^\star(\cdot)$ to suppress absolute constants, polynomials in $R_{\mathrm{nat}}, R_G, R_M$ and logarithms in $T$.

## 3 Online Control over Convex Policies

This section considers online control of unknown LTV systems so as to compete with the convex DRC and DAC policy classes. The fundamental quantity which appears throughout our results is the *system variability*, which measures the variation of the time-varying Markov operators $G_t$ over intervals $I$.

**Definition 3.1.** Define the system variability of an LTV dynamical system with Markov operators $\mathbf{G} = G_{1:T}$ over a contiguous interval $I \subseteq [T]$ to be

$$\mathrm{Var}_I(\mathbf{G}) = \min_G \frac{1}{|I|} \sum_{t \in I} \|G - G_t\|_{\ell_2, F}^2 = \frac{1}{|I|} \sum_{t \in I} \|G_I - G_t\|_{\ell_2, F}^2,$$

where $\|\cdot\|_{\ell_2, F}$ indicates the $\ell_2$ norm of the fully vectorized operator and $G_I = |I|^{-1} \sum_{t \in I} G_t$ is the empirical average of the operators that correspond to $I$. Recall that $\mathrm{Var}_T(\mathbf{G})$ corresponds to $I = [T]$.

Our results in this section for both upper and lower bounds focus on expected regret: high probability results are possible as well with more technical effort using standard techniques.

## 3.1 A Linear Regret Lower Bound

Our first contribution is a negative one: that the regret against the class of either DAC or DRC policies cannot scale sublinearly in the time horizon. Informally, our result shows that the regret against these classes scales as $T\sigma$, where $\sigma^2$ is the system variability.

More precisely, for any $\sigma^2 \in (0, 1/8]$, we construct a distribution $\mathcal{D}_\sigma$ over sequences $(A_t, B_t, c_t, w_t)$, formally specified in Appendix F.2. Here, we list the essential properties of $\mathcal{D}_\sigma$: *(i)* $A_t \equiv 0$, *(ii)* $c_t \equiv c$ is a fixed cost satisfying Assumption 3 with $L \leq 4$, *(iii)* the matrices $(B_t)$ are i.i.d., with $\|B_t\|_{\mathrm{op}} \leq 2$ almost surely, and $\mathbb{E}[\|B_t - \mathbb{E}[B_t]\|_{\mathrm{F}}^2] = \sigma^2$, and *(iv)* $\|w_t\| \leq 4$ for all $t$. These conditions imply that Assumptions 1 and 2 hold for $R_G = 2, \rho = 0, R_{\mathrm{nat}} = 4$. The condition $A_t \equiv 0$ implies that $x_t^{\mathrm{nat}} = w_t$ for all $t$, so the classes DRC and DAC are equivalent and the lower bound holds over both. Moreover, by Jensen's inequality, this construction ensures that

$$\mathbb{E}[\mathrm{Var}_I(\mathbf{G})] = |I|^{-1}\mathbb{E}[\min_G \sum_{t \in I} \|G - G_t\|_{\ell_2, F}^2]$$
$$= |I|^{-1}\mathbb{E}[\min_B \sum_{t \in I} \|B - B_t\|_{\mathrm{F}}^2] \leq \mathbb{E}[\|B_t - \mathbb{E}[B_t]\|_{\mathrm{F}}^2] = \sigma^2.$$

In particular, $\mathbb{E}[\mathrm{Var}_T(\mathbf{G})] \leq \sigma^2$. For the described construction, we show the following lower bound:

**Theorem 3.1.** *Let $C$ be a universal, positive constant. For any $\sigma \in (0, 1/8]$ and any online control algorithm $\mathcal{A}$, there exists a DRC policy $\pi^\star \in \Pi_{\mathrm{drc}}(1, 1)$ s.t. expected regret incurred by $\mathcal{A}$ under the distribution $\mathcal{D}_\sigma$ and cost $c(x, u)$ is at least*

$$\mathbb{E}_{\mathcal{D}_\sigma, \mathcal{A}}[\mathrm{Regret}_T(\mathcal{A}; \{\pi^\star\})] \geq C \cdot T\sigma \geq C \cdot T \cdot \sqrt{\mathbb{E}[\mathrm{Var}_T(\mathbf{G})]},$$

A full construction and proof of Theorem 3.1 is given in Appendix F.2. In particular, for $\sigma = 1/8$, we find that no algorithm can attain less than $\Omega(T)$ expected regret; a stark distinction from either unknown LTI [39, 20] or known LTV [16] systems.

## 3.2 Estimation of Time-Varying Vector Sequences

To devise an algorithmic upper bound that complements the result in Theorem 3.1, we first consider the setting of online prediction under a partial information model. This setting captures the *system identification* phase of LTV system control and is used to derive the final control guarantees. Formally, consider the following repeated game between a learner and an *oblivious* adversary: at each round $t \in [T]$, the adversary picks a target vector $\mathbf{z}_t^\star \in \mathcal{K}$ from a convex decision set $\mathcal{K}$ contained in a 0-centered ball of radius $R_z$; simultaneously, the learner selects an estimate $\hat{\mathbf{z}}_t \in \mathcal{K}$ and suffers quadratic loss $\ell_t(\hat{\mathbf{z}}_t) = \|\hat{\mathbf{z}}_t - \mathbf{z}_t^\star\|^2$. The only feedback the learner has access to is via the following noisy and costly oracle.

**Oracle 1** (Noisy Costly Oracle). At each time $t \in [T]$, the learner selects a decision $b_t \in \{0, 1\}$ indicating whether a query is sent to the oracle. If $b_t = 1$, the learner receives an unbiased estimate $\tilde{\mathbf{z}}_t$ as response such that $\|\tilde{\mathbf{z}}_t\| \leq \tilde{R}_z$ and $\mathbb{E}[\tilde{\mathbf{z}}_t \mid \mathcal{F}_t, b_t = 1] = \mathbf{z}_t^\star$. The filtration $\mathcal{F}_t$ is the sigma algebra generated by $\tilde{\mathbf{z}}_{1:t-1}, b_{1:t-1}$ and the choices of the oblivious adversary $\mathbf{z}_{1:T}^\star$. A completed query results in a unit cost $\lambda > 0$ for the learner.

The performance metric of an online prediction algorithm $\mathcal{A}_{\mathrm{pred}}$ is expected quadratic loss regret along with the extra cumulative oracle query cost. It is defined over each interval $I = [r, s] \subseteq [T]$ as

$$\mathrm{Regret}_I(\mathcal{A}_{\mathrm{pred}}; \lambda) = \mathbb{E}_{\mathcal{F}_{1:T}}\left[\sum_{t \in I} \ell_t(\hat{\mathbf{z}}_t) - \min_{\mathbf{z} \in \mathcal{K}} \sum_{t \in I} \ell_t(\mathbf{z}) + \lambda \sum_{t \in I} b_t\right]. \quad (3.1)$$

To attain *adaptive* regret, i.e. bound Eq. (3.1) for each interval $I$, we propose Algorithm 1 constructed as follows. First, suppose we wanted non-adaptive (i.e. just $I = [T]$) guarantees. In this special case, we propose to sample $b_t \sim \mathrm{Bernoulli}(p)$ for an appropriate parameter $p \in (0, 1)$, and perform a gradient descent update on the importance-weighted square loss $\tilde{\ell}_t(\mathbf{z}) = \frac{1}{2p}\mathbb{I}\{b_t = 1\}\|\mathbf{z} - \tilde{\mathbf{z}}_t\|^2$. To

---

**Algorithm 1** Adaptive Estimation Algorithm (ADA-PRED)

---

1: **Input:** parameter $p$, decision set $\mathcal{K}$

2: **Initialize:** $\hat{\mathbf{z}}_1^{(1)} \in \mathcal{K}$, working dictionary $\mathcal{S}_1 = \{(1 : \hat{\mathbf{z}}_1^{(1)})\}$, $q_1^{(1)} = 1$, parameter $\alpha = \frac{p}{(R_z + \tilde{R}_z)^2}$

3: **for** $t = 1, \ldots, T$ **do**

4:   Play iterate $\hat{\mathbf{z}}_t = \sum_{(i, \hat{\mathbf{z}}_t^{(i)}) \in \mathcal{S}_t} q_t^{(i)} \hat{\mathbf{z}}_t^{(i)}$

5:   Draw/Receive $b_t \sim \mathrm{Bernoulli}(p)$

6:   **if** $b_t = 1$ **then**

7:    Request estimate $\tilde{\mathbf{z}}_t$ from Oracle 1

8:    Let $\tilde{\ell}_t(\mathbf{z}) = \frac{1}{2p}\|\mathbf{z} - \tilde{\mathbf{z}}_t\|^2$ and $\tilde{\nabla}_t = \frac{1}{p}(\mathbf{z}_t - \tilde{\mathbf{z}})$

9:   **else**

10:    Let $\tilde{\mathbf{z}}_t \leftarrow \emptyset$ and $\tilde{\ell}_t(\mathbf{z}) = 0$ and $\tilde{\nabla}_t = 0$

11:   Update predictions $\hat{\mathbf{z}}_{t+1}^{(i)} \leftarrow \mathrm{Proj}_{\mathcal{K}}(\hat{\mathbf{z}}_t^{(i)} - \eta_t^{(i)} \tilde{\nabla}_t)$ for all $(i, \hat{\mathbf{z}}_t^{(i)}) \in S_t$

12:   Form new dictionary $\tilde{\mathcal{S}}_{t+1} = (i, \hat{\mathbf{z}}_{t+1}^{(i)})_{i \in \mathrm{keys}(\mathcal{S}_t)}$

13:   Construct proxy new weights $\bar{q}_{t+1}^{(i)} = \frac{t}{t+1} \cdot \frac{q_t^{(i)} e^{-\alpha \tilde{\ell}_t(\hat{\mathbf{z}}_t^{(i)})}}{\sum_{j \in \mathrm{keys}(\mathcal{S}_t)} q_t^{(j)} e^{-\alpha \tilde{\ell}_t(\hat{\mathbf{z}}_t^{(j)})}}$ for all $i \in \mathrm{keys}(\mathcal{S}_t)$

14:   Add new instance $\tilde{\mathcal{S}}_{t+1} \leftarrow \tilde{\mathcal{S}}_{t+1} \cup (t+1, \hat{\mathbf{z}}_{t+1}^{(t+1)})$ for arbitrary $\hat{\mathbf{z}}_{t+1}^{(t+1)} \in \mathcal{K}$ with $\bar{q}_{t+1}^{(t+1)} = \frac{1}{t+1}$

15:   Prune $\tilde{\mathcal{S}}_{t+1}$ to form $\mathcal{S}_{t+1}$ (see Appendix C.1)

16:   Normalize $q_{t+1}^{(i)} = \frac{\bar{q}_{t+1}^{(i)}}{\sum_{j \in \mathrm{keys}(\mathcal{S}_{t+1})} \bar{q}_{t+1}^{(j)}}$

---

extend this method to enjoy adaptive regret guarantees, we adopt the approach of [21]: the core idea in this approach is to initiate an instance of the base method at each round $t$ and use a weighted average of the instance predictions as the final prediction (Line 4). The instance weights are multiplicatively updated according to their performance (Line 13). To ensure computational efficiency, the algorithm only updates instances from a working dictionary $\mathcal{S}_t$ (Line 11). These dictionaries are pruned each round (Line 15) such that $|\mathcal{S}_t| = O(\log T)$ (see Appendix C.1 for details).

**Theorem 3.2.** *Given access to queries from Oracle 1 and with stepsizes $\eta_t^{(i)} = \frac{1}{t-i+1}$, Algorithm 1 enjoys the following adaptive regret guarantee: for all $I = [r, s] \subseteq [T]$,*

$$\mathrm{Regret}_I(\text{ADA-PRED}; \lambda) \leq \frac{2(R_z + \tilde{R}_z)^2(1 + \log s \cdot \log |I|)}{p} + \lambda p |I| . \qquad (3.2)$$

When $I = [T]$, the optimal choice of parameter $p = \log T / \sqrt{\lambda T}$ yields regret scaling roughly as $\sqrt{\lambda T \log^2 T}$. Unfortunately, this gives regret scaling as $\sqrt{T}$ for all interval sizes: to attain $\sqrt{|I|}$ regret on interval $I$, the optimal choice of $p$ would yield $\sim T / \sqrt{|I|}$ regret on $[T]$, which is considerably worse for small $|I|$. One may ask if there exists a *strongly adaptive* algorithm which adapts $p$ as well, so as to enjoy regret polynomial in $|I|$ for all intervals $I$ *simultaneously* [11]. The following result shows this is not possible:

**Theorem 3.3** (Informal). *For all $\gamma > 0$ and $\lambda > 0$, there exists no online algorithm $\mathcal{A}$ with feedback access to Oracle 1 that enjoys strongly adaptive regret of $\mathrm{Regret}_I(\mathcal{A}; \lambda) = \tilde{O}(|I|^{1-\gamma})$.*

Hence, in a sense, Algorithm 1 is as adaptive as one could hope for: it ensures a regret bound for all intervals $I$, but not a strongly adaptive one. The lower bound construction, formal statement, and proof of Theorem 3.3 are given in Appendix C.2.

### 3.3 Adaptive Regret for Control of Unknown Time-Varying Dynamics

We now apply our adaptive estimation algorithm (Algorithm 1) to the online control problem. Our proposed algorithm, Algorithm 2, takes in two sub-routines: a prediction algorithm $\mathcal{A}_{\mathrm{pred}}$ which enjoys low prediction regret in the sense of the previous section, and a control algorithm $\mathcal{A}_{\mathrm{ctrl}}$ which has low regret for control of *known* systems. Our master algorithm trades off between the two in epochs $\tau = 1, 2, \ldots$ of length $h$: each epoch corresponds to one step of $\mathcal{A}_{\mathrm{pred}}$ indexed by $[\tau]$.

At each epoch, the algorithm receives Markov operator estimates from $\mathcal{A}_{\mathrm{pred}}$ (Line 3) and makes a binary decision $b_{[\tau]} \sim \mathrm{Bernoulli}(p)$. If $b_{[\tau]} = 1$, then it explores using i.i.d. Rademacher inputs (Line 7), and sends the resulting estimator to $\mathcal{A}_{\mathrm{pred}}$ (Line 14). This corresponds to one query from Oracle 1. Otherwise, it selects inputs in line with $\mathcal{A}_{\mathrm{ctrl}}$ (Line 9), and does not give a query to $\mathcal{A}_{\mathrm{pred}}$ (Line 16). Regardless of exploration decision, the algorithm feeds costs, current estimates of the Markov operator and Nature's x's based on the Markov operator estimates to $\mathcal{A}_{\mathrm{ctrl}}$ (Lines 10-12), which it uses to select inputs and update its parameter.

The prediction algorithm $\mathcal{A}_{\mathrm{pred}}$ is taken to be ADA-PRED with the decision set $\mathcal{K} = \mathcal{G}(h, R_G)$: the projection operation onto it and the ball $\mathbb{B}_{R_{\mathrm{nat}}}$ is done by clipping when the norm of the argument exceeds the indicated bound. The control algorithm $\mathcal{A}_{\mathrm{ctrl}}$ is taken to be DRC-OGD [39] for *known* systems.

**Theorem 3.4.** *For $h = \dfrac{\log T}{\log \rho^{-1}}$, $p = T^{-1/3}$ and $m \leq \sqrt{T}$, on any contiguous interval $I \subseteq [T]$, Algorithm 2 enjoys the following adaptive regret guarantee:*

$$\mathbb{E}\left[\mathrm{Regret}_I(\text{ADA-CTRL}); \Pi_{\mathrm{drc}}(m, R_M)\right] \leq \widetilde{\mathcal{O}}^{\star}\left(Lm\left(|I|\sqrt{\mathbb{E}[\mathrm{Var}_I(\mathbf{G})]} + d_u T^{2/3}\right)\right) \qquad (3.3)$$

*Proof Sketch.* The analysis proceeds by reducing the regret incurred to that over a known system, accounting for: 1) the additional exploration penalty $(O(p|I|))$, 2) the system misspecification induced error $(\sim \sum_{t \in I} \|\hat{G}_t - \bar{G}_t^h\|_{\ell_1, \mathrm{op}})$, and 3) truncation errors $(\sim \psi(h)|I|)$. Via straightforward computations, the system misspecification error can be expressed in terms of the result in Theorem 3.2, ultimately leading to an error contribution $\sim |I|\sqrt{\mathbb{E}[\mathrm{Var}_I(\mathbf{G})]} + p^{-1/2}|I|^{1/2}$. The analysis is finalized by noting that the chosen $p$ ideally balances $p|I|$ and $p^{-1/2}|I|^{1/2}$, and that the chosen $h$ ensures that the truncation error is negligible. The full proof can be found in Appendix D. $\qquad \square$

The adaptive regret bound in Eq. (3.3) has two notable terms. Note that the first term $|I|\sqrt{\mathbb{E}[\mathrm{Var}_I(\mathbf{G})]}$ for $I = [T]$ matches the regret lower bound in Theorem 3.1. Furthermore, our algorithm is adaptive in this term for all intervals $I$. On the other hand, for unknown LTI systems with $\mathrm{Var}_I(\mathbf{G}) = 0$, the algorithm recovers the state-of-the-art bound of $T^{2/3}$ [20]. However, the $T^{2/3}$ term is not adaptive to the intervals $I$ consistent with the lower bound against strongly adaptive algorithms in Theorem 3.3.

---

**Algorithm 2** DRC-OGD with Adaptive Exploration (ADA-CTRL)

---

1: **Input:** $p, h, \mathcal{A}_{\mathrm{pred}} \leftarrow$ ADA-PRED$(p, \hat{G}_0, \mathcal{G}(h, R_G))$, $\mathcal{A}_{\mathrm{ctrl}} \leftarrow$ DRC-OGD$(m, R_{\mathcal{M}})$
2: **for** $\tau = 1, \ldots, T/h$ **do**  $\quad\quad\quad\quad\quad\quad\quad\quad\quad\quad\quad\quad\quad\quad\quad\quad$ ▷ let $t_\tau = (\tau - 1)h + 1$
3: $\quad$ Set $\hat{G}_{t_\tau}, \hat{G}_{t_\tau+1}, \ldots, \hat{G}_{t_\tau+h-1}$ equal to $\tau$-th iterate $\hat{G}_{[\tau]}$ from $\mathcal{A}_{\mathrm{pred}}$
4: $\quad$ Draw $b_{[\tau]} \sim \mathrm{Bernoulli}(p)$
5: $\quad$ **for** $t = t_\tau, \ldots, t_\tau + h - 1$ **do**
6: $\quad\quad$ **if** $b_{[\tau]} = 1$ **then**
7: $\quad\quad\quad$ Play control $u_t \sim \{\pm 1\}^{d_u}$
8: $\quad\quad$ **else**
9: $\quad\quad\quad$ Play control $u_t$ according to the $t$-th input chosen by $\mathcal{A}_{\mathrm{ctrl}}$
10: $\quad\quad$ Suffer cost $c_t(x_t, u_t)$, observe new state $x_{t+1}$
11: $\quad\quad$ Extract $\hat{x}_{t+1}^{\mathrm{nat}} = \mathrm{Proj}_{\mathbb{B}_{R_{\mathrm{nat}}}}\left(x_{t+1} - \sum_{i=0}^{h-1} \hat{G}_t^{[i]} u_{t-i}\right)$
12: $\quad\quad$ Feed cost, Markov operator and Nature's x estimates $(c_t, \hat{G}_t, \hat{x}_{t+1}^{\mathrm{nat}})$ to $\mathcal{A}_{\mathrm{ctrl}}$.
13: $\quad$ **if** $b_{[\tau]} = 1$ **then**
14: $\quad\quad$ Feed $(b_{[\tau]}, \tilde{G}_{[\tau]})$ to $\mathcal{A}_{\mathrm{pred}}$, where $\tilde{G}_{[\tau]}^{[i]} = x_{t_\tau+h} u_{t_\tau+h-i}^\top$, $i = 0, 1, \ldots, h - 1$.
15: $\quad$ **else**
16: $\quad\quad$ Feed $(b_{[\tau]}, \tilde{G}_{[\tau]})$ to $\mathcal{A}_{\mathrm{pred}}$, where $\tilde{G}_{[\tau]} \leftarrow \emptyset$.

---

## 4  Online Control over State Feedback

Given the impossibility of sublinear regret against DRC/DAC without further restrictions on system variability, this section studies whether sublinear regret is possible against the class of linear feedback

policies. For simplicity, we focus on the *state feedback* policies $u_t = Kx_t$, that is, linear feedback policies with memory $m = 1$ (Definition 2.3). We note that state feedback policies were the class which motivated the relaxation to DAC policies in the first study of nonstochastic control [3].

We present two results, rather qualitative in nature. First, we show that obtaining *sublinear regret is, in the most literal sense, possible*. The following result considers regret relative to a class $\mathcal{K}$ of static feedback controllers which satisfy the restrictive assumption that each $K \in \mathcal{K}$ stabilizes the time varying dynamics $(A_t, B_t)$; see Appendix E for the formal algorithm, assumptions, and guarantees. We measure the regret against this class $\mathcal{K}$:

$$\mathrm{Regret}_T(\mathcal{K}) := \sum_{t=1}^{T} c_t(x_t, u_t) - \inf_{K \in \mathcal{K}} \sum_{t=1}^{T} c_t(x_t^K, u_t^K),$$

where $(x_t^K, u_t^K)$ are the iterates arising under the control law $u_t = Kx_t$.

**Theorem 4.1** (Sublinear regret against state-feedback). *Under a suitable stabilization assumption, there exists a computationally inefficient control algorithm which attains sublinear expected regret:*

$$\mathbb{E}[\mathrm{Regret}_T(\mathcal{K})] \le e^{\Omega(d_x d_u/2)} \cdot T^{1 - \frac{1}{2(d_x d_u + 3)}}.$$

Above, $\Omega(\cdot)$ suppresses a universal constant and exponent base, both of which are made explicit in a formal theorem statement in Appendix E. The bound follows by running the EXP3 bandit algorithm on a discretization of the set $\mathcal{K}$ (high probability regret can be obtained by instead using EXP3.P [8]). The guarantee in Theorem 4.1 is neither practical nor sharp; its sole purpose is to confirm the possibility of sublinear regret. Due to the bandit reduction and exponential size of the cover of $\mathcal{K} \subset \mathbb{R}^{d_u \times d_x}$, the algorithm is computationally inefficient and suffers a *nonparametric* rate of regret [33]: $\epsilon$-regret requires $T = \epsilon^{-\Omega(\mathrm{dimension})}$.

One may wonder if one can do much better than this naive bandit reduction. For example, is there structure that can be leveraged? For LTV systems, we show that there is strong evidence to suggest that, at least from a computational standpoint, attaining polynomial regret (e.g. $T^{1-\alpha}$ for $\alpha > 0$ independent of dimension) is computationally prohibitive.

**Theorem 4.2.** *There exists a reduction from* MAX-3SAT *on $m$-clauses and $n$-literals to the problem of finding a state-feedback controler $K$ which is within a small constant factor of optimal for the cost $\sum_{t=1}^{T} c_t(x_t^K, x_t^K)$ on a sequence of sequentially stable LTV systems and convex costs $(A_t, B_t, c_t)$ with no disturbance ($w_t \equiv 0$), with state dimension $n+1$, input dimension $2$, and horizon $T = \Theta(mn)$. Therefore, unless* P = NP, *the latter cannot be solved in time polynomial in $n$ [17].*

A more precise statement, construction, and proof are given in Appendix F.4. Theorem 4.2 demonstrates that solving the *offline optimization* problem over state feedback controllers $K$ to within constant precision is NP-Hard. In particular, this means that any sublinear regret algorithm which is *proper and convergent*, in the sense that $u_t = K_t x_t$ for some sequence $K_t$ converges to a limit as $T \to \infty$, must be computationally inefficient. This is true *even if the costs and dynamics are known in advance*. Our result suggests it is computationally hard to obtain sublinear regret, but it does not rigorously imply it. For example, there may be more clever convex relaxations (other than DRC and DAC, which provably cannot work) that yield efficient and sublinear regret. Secondly, this lower bound does not rule out the possibility of an computationally inefficient algorithm which nevertheless attains polynomial regret.

## 5  Discussion and Future Work

This paper provided guarantees for and studied the limitations of sublinear additive regret in online control of an unknown, linear time-varying (LTV) dynamical system.

Our setting was motivated by the fact that the first-order Taylor approximation (Jacobian linearization) of smooth, nonlinear systems about any smooth trajectory is LTV. One would therefore hope that low-regret guarantees against LTV systems may imply convergence to first-order stationary points of general nonlinear control objectives [34], which in turn may enjoy stability properties [45]. Making this connection rigorous poses several challenges. Among them, one would need to extend our low-regret guarantees against oblivious adversaries to hold against adaptive adversaries, the latter

modeling how nonlinear system dynamics evolve in response to the learner's control inputs. This may require parting from our current analysis, which leverages the independence between exploratory inputs and changes in system dynamics.

Because we show that linear-in-$T$ regret is unavoidable for changing systems with large system variability, at least for the main convex policy parametrizations, it would be interesting to study our online setting under other measures of performance. In particular, the *competive ratio*, or the *ratio* of total algorithm cost to optimal cost in hindsight (as opposed to the *difference* between the two measured by regret) may yield a complementary set of tradeoffs, or lead to new and exciting principles for adaptive controller design. Does system variability play the same deciding roles in competive analysis as it does in regret? And, in either competitive or regret analyses, what is the correct measure of system variability (e.g. variability in which norm/geometry, or of which system parameters) which best captures sensitivity of online cost to system changes?

## Acknowledgments

Elad Hazan and Edgar Minasyan have been supported in part by NSF grant #1704860. This work was done in part when Paula Gradu was at Google AI Princeton and Princeton University. Max Simchowitz is generously supported by an Open Philanthropy AI fellowship.

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
