# Contents

# A  Extensions

## A.1  Affine Offsets

For many systems, performance improves dramatically for controllers with constant affine terms, that is

$$u_t^M = \bar{u}^M + \sum_{t=0}^{h-1} M^{[i]} x_{t-i}^{\mathrm{nat}},$$

where $\bar{u}^M$ is a constant affine term encoded by $M = (M^{[0]}, \ldots, M^{[h-1]}, \bar{u}^M)$. All our arguments apply more generally to control policies of this form. Moreover, we can even allow linear combinations of time varying terms:

$$u_t^M = \sum_{t=0}^{h-1} M^{[i]} x_{t-i}^{\mathrm{nat}} + \sum_{t=0}^{h'-1} M^{[i]'} \psi_i(t)$$

$$M = (M^{[0]}, \ldots, M^{[h-1]}, M^{[0]'}, \ldots, M^{[h'-1]'}),$$

where now $\psi_i(t)$ are fixed, possibly time varying basis functions (which do not depend on $M$). The case of constant affine terms corresponds to $h' = 1$, and $\psi_i(t) = 1$ for all $t$.

## A.2  Changing Stabilizing Controllers

Our results extend naturally to the following setting: for each time $t = 1, 2, \ldots, T$, the algorithm has access to a static feedback control policy $K_t^{\mathrm{stb}}$ such that the closed loop matrices $(A_t + B_t K_t^{\mathrm{stb}})$ are sequentially stable, that is

$$\Phi_{s+h,s}^{\mathrm{stb}} := \prod_{i=s}^{s+h} (A_t + B_t K_t^{\mathrm{stb}})$$

has geometric decay. We let $x_t^{\mathrm{nat}}$ denote the iterates produced by the updates

$$x_{t+1}^{\mathrm{nat}} = (A_t + B_t K_t^{\mathrm{stb}}) x_t^{\mathrm{nat}}, u_t^{\mathrm{nat}} = K x_t^{\mathrm{nat}}, \quad x_1^{\mathrm{nat}} = 0.$$

We compute the stabilized policies of the form

$$u_t^M = K_t^{\mathrm{stb}} x_t + \sum_{i=0}^{h-1} M^{[i]} x_{t-i}^{\mathrm{nat}}.$$

To facillicate the extension, we define the stabilized Markov operator

$$G_{\star,t}^{[0]} = \begin{bmatrix} 0 \\ I_{d_u} \end{bmatrix}, \quad G_{\star,t}^{[i]} := \begin{bmatrix} I_{d_x} \\ K_t \end{bmatrix} \Phi_{t,t-i+1}^{\mathrm{stb}} B_{t-i},$$

This Markov operator satisfies

$$\begin{bmatrix} x_t \\ u_t \end{bmatrix} = \begin{bmatrix} x_t^{\mathrm{nat}} \\ u^{\mathrm{nat}} \end{bmatrix} + \sum_{i=0}^{t-1} G_{\star,t}^{[i]} \tilde{u}_t, \quad \text{s.t. } u_t = K_t x_t + \tilde{u}_t.$$

With similar techniques, we obtain estimates $\widehat{G}_t$, back out estimates of the Nature's sequence $(x^{\mathrm{nat}}, u^{\mathrm{nat}})$ via

$$\begin{bmatrix} \hat{x}_t^{\mathrm{nat}} \\ \hat{u}^{\mathrm{nat}} \end{bmatrix} = \mathrm{clip}_r \left( \begin{bmatrix} x_t \\ u_t \end{bmatrix} - \sum_{i=0}^{h} \widehat{G}_t^{[i]} (u_{t-i} - K_{t-i}^{\mathrm{stb}} x_{t-i}) \right),$$

for a truncation radius $r > 0$ suitably chosen. Recall that we apply this $\mathrm{clip}_r(\cdot)$ operator to ensure the estimates of the Nature's sequenc does not grow unbounded and exert feedback. We then select inputs

$$\hat{u}_t(M) = K_t^{\mathrm{stb}} x_t + \sum_{i=0}^{h-1} M^{[i]} \hat{x}_{t-i}^{\mathrm{nat}},$$

Markov operator with low adaptive regret, and apply our OCO-with-memory algorithm to losses

$$\widehat{f}_t(M) = c_t \left( \begin{bmatrix} \hat{x}_t^{\mathrm{nat}} \\ \hat{u}_t^{\mathrm{nat}} \end{bmatrix} - \sum_{i=0}^{h} \widehat{G}_t^{[i]} \hat{u}_{t-i}(M) \right).$$

### A.3 Partial Observation

Our results further extend to partially observed systems. We explain this extension for sequentially stable systems; extensions to sequentialy stabilized systems by time varying linear dynamic controllers follows from the exposition in [39], Appendix C.

For partially observed systems, we have the same state transition dynamics $x_{t+1} = A_t x_t + B_t u_t + w_t$, but, for a time-varying observation matrix $C_t$ and process noise $e_t$, we observe outputs $y_t \in \mathbb{R}^{d_y}$

$$y_t = C_t x_t + e_t.$$

Costs $c_t(y_t, u_t)$ are suffered on input and outputs. As for full observation, the Nature's sequence $x^{\text{nat}}$ and $y^{\text{nat}}$ correspond to states $x_t$ and outputs $y_t$ which arise under identically zero input $u_t \equiv 0$. The DRC parametrization selects linear conmbinations of Nature's y's:

$$u_t^M = \sum_{i=0}^{m-1} M^{[i]} y_{t-i}^{\text{nat}}.$$

Recalling

$$\Phi_{s+h,s} := \prod_{i=s}^{s+h} (A_t + B_t),$$

the relevant Markov operators $G_{\star t}$ are the ones mapping inputs to outputs:

$$G_{\star,t}^{[0]} = 0, \quad G_{\star,t}^{[i]} = C_t \Phi_{t,t-i+1} B_{t-i},$$

With similar techniques, we obtain estimates $\widehat{G}_t$, back out estimates of the Nature's sequence $y^{\text{nat}}$ via

$$\hat{y}^{\text{nat}} = \text{clip}_r \left( y_t - \sum_{i=0}^{h} \widehat{G}_t^{[i]} u_{t-i} \right)$$

for truncation radius $r > 0$ suitably chosen, we select inputs

$$\hat{u}_t(M) = \sum_{i=0}^{h-1} M^{[i]} \hat{y}_{t-i}^{\text{nat}}),$$

update parameters with the OCO-with-memory losses

$$\widehat{f}_t(M) = \ell_t(\hat{y}_t^{\text{nat}} - \sum_{i=0}^{h} \widehat{G}_t^{[i]} M^{[i]} \hat{u}_{t-i}(M), u_t).$$

### A.4 The DAC parametrization

Here we sketch an algorithm to compete with DAC-parametrized control policies [3]. For simplicity, we focus on sequentially stable system, though the discussion extends to systems sequentially stabilized by sequences of controllers $(K_t^{\text{stb}})$. Note that DAC *does not* apply under partial observation.

Recall that, in the DAC parametrization, the inputs are selected as linear combinations of past disturbances:

$$u_t^M = \sum_{i=0}^{h-1} M^{[i]} w_{t-i-1}.$$

To implement DAC, we therefore need empiricals estimate $\hat{w}_t$ of $w_t$. As per [20], it suffices to construct estimates $(\hat{A}_t, \hat{B}_t)$ of $(A_t, B_t)$, and choose

$$\hat{w}_t = \text{clip}_r \left( x_t - \hat{A}_{t-1} x_{t-1} - \hat{B}_{t-1} u_{t-1} \right),$$

again clipped at a suitable radius $r > 0$ to block compounding feedback. Given these estimates, our algorithm extends to DAC control in the expected way.

How does one obtain the estimates $(\hat{A}_t, \hat{B}_t)$? First, we observe that since

$$G_{\star,t}^{[0]} = 0, \quad G_{\star,t}^{[i]} = \Phi_{t,t-i+1} B_{t-i}, \quad \Phi_{s+h,s} = \prod_{i=s}^{s+h} A_i,$$

we have $B_t = G_{\star,t}^{[1]}$. Hence, we can select $\hat{B}_t$ as $\widehat{G}_t^{[1]}$.

The estimate of $A_t$ is more involved. For linear, *time-invariant* systems, $A$ can be recovered from $G_\star$ via the Ho-Kalman procedure as is does in [20] (see also [30, 36]). For time-varying systems, this become more challenging. Ommitting details in the interest of brevity, one can use the robustness properties of Ho-Kalman to argue that if the system matrices are slow moving (an assumption required for low regret), $G_{\star,t}$ is close to a stationarized analogue $\bar{G}_{\star,t}$ given by

$$\bar{G}_{\star,t}^{[i]} := A_t^{i-1} B_t.$$

Hence, we can view any estimate $\widehat{G}_t$ of $G_{\star,t}$ as an estimate of $\bar{G}_{\star,t}$, and apply Ho-Kalman to the latter.

# B   Adapive Regret for Time-Varying DRC-OGD

We first extend the DRC-OGD algorithm from [39] to the setting of *known* linear time-varying dynamics. We spell out Algorithm 3 and prove it attains $\tilde{\mathcal{O}}\left(\sqrt{T}\right)$ adaptive regret over general convex costs under fully adversarial noise (Theorem B.1) with respect to the DRC policy class. The main technique, using OGD over the DRC parametrization, remains unchanged from the original paper and we show it generalizes naturally to LTV systems.

---

**Algorithm 3** Disturbance Response Control via Online Gradient Descent (DRC-OGD)

---
1: **Input:** stepsize $\eta$, memory $m$, radius $R_\mathcal{M}$
2: Initialize $M_1 \in \mathcal{M}(m, R_\mathcal{M})$ arbitrarily
3: Receive initial state $x_1$, set $x_1^{\text{nat}} = x_1$ and $x_{\leq 0}^{\text{nat}} = 0$
4: **for** $t = 1 \ldots T$ **do**
5:     Play control $u_t = \sum_{i=0}^{m-1} M_t^{[i]} x_{t-i}^{\text{nat}}$
6:     Suffer $c_t(x_t, u_t)$ and observe cost function $c_t(\cdot, \cdot)$
7:     Construct $f_t(M_0, \ldots, M_h) \doteq c_t(\hat{x}_t(M_{0:h-1}), u_t(M_h))$ and let $\tilde{f}_t(M) \doteq f_t(M, \ldots, M)$
8:     Update $M_{t+1} \leftarrow \Pi_\mathcal{M}\left(M_t - \eta \nabla \tilde{f}_t(M_t)\right)$
9:     Receive new system $G_t$
10:     Receive new state $x_{t+1}$ and extract $x_{t+1}^{\text{nat}} = x_{t+1} - \sum_{i=0}^{t-1} G_t^{[i]} u_{t-i}$ or receive $x_{t+1}^{\text{nat}}$

---

**Theorem B.1.** *Running Algorithm 3 with* $\eta = \frac{\sqrt{d_{\min}} R_\mathcal{M}^2}{2LR_{\text{sys}}^2 (h+1)^{5/4} \sqrt{T}}$ *guarantees the following regret bound on every interval* $I = [r, s]$:

$$\sum_{t=r}^{s} c_t(x_t, u_t) - \min_{\pi \in \Pi_{\text{drc}}} \sum_{t=r}^{s} c_t(x_t^\pi, u_t^\pi) \leq 6LR_{\text{sys}}^2 \left(3\sqrt{d_{\min}} m(h+1)^{5/4}\sqrt{T} + \psi(h)|I|\right)$$

*where* $R_{\text{sys}} = R_G R_\mathcal{M} R_{\text{nat}}$ *and* $d_{\min} = \min\{d_x, d_u\}$.

## B.1   Adaptive Regret of OGD for functions with memory

We first prove that OGD with a fixed stepsize attains $O(\sqrt{T})$ adaptive regret for functions with memory.

---

**Algorithm 4** Online Gradient Descent for OCOwMem (Mem-OGD)

---
1: **Input:** stepsize $\eta$, memory $m$, set $\mathcal{K}$
2: Initialize $x_1 \in \mathcal{K}$ arbitrarily, set $x_{\leq 0} = x_1$ by convention
3: **for** $t = 1 \ldots T$ **do**
4:      Play $x_t$
5:      Suffer $f_t(x_{t-h}, \ldots, x_t)$ and observe loss function $f_t(\cdot, \ldots, \cdot)$
6:      Construct proxy loss function $\tilde{f}_t(x) \doteq f_t(x, \ldots, x)$
7:      Update $x_{t+1} = \Pi_{\mathcal{K}} \left( x_t - \eta \nabla \tilde{f}_t(x_t) \right)$

---

**Theorem B.2.** *Let $\{f_t : \mathcal{K}^{h+1} \to [0,1]\}_{t=1}^T$ be a sequence of $L$ coordinate-wise Lipschitz loss functions with memory such that $\tilde{f}_t$ (Line 6) is convex. Then, on any interval $I = [r,s] \subseteq [T]$, Algorithm 4 enjoys the following adaptive policy regret guarantee:*

$$\sum_{t=r}^{s} f_t(x_{t-h}, \ldots, x_t) - \min_{x \in \mathcal{K}} \sum_{t=r}^{s} \tilde{f}_t(x) \leq \frac{D^2}{\eta} + 2\eta L^2 (h+1)^{5/2} |I|$$

*where $D = diam(\mathcal{K})$.*

First we state and prove the following well-known fact about vanilla projected OGD over (memory-less) loss functions:

**Fact B.1.** *Let $\{\tilde{f}_t : \mathcal{K} \to [0,1]\}_{t=1}^T$ be a sequence of convex loss functions with $\|\nabla \tilde{f}(x)\| \leq G$. Then, on any interval $I = [r,s] \subseteq [T]$, projected OGD enjoys the following guarantee:*

$$\sum_{t=r}^{s} \tilde{f}_t(x_t) - \min_{x \in \mathcal{K}} \sum_{t=r}^{s} \tilde{f}(x) \leq \frac{D^2}{\eta} + \eta |I| G^2$$

*where $D = diam(\mathcal{K})$.*

*Proof.* Consider an arbitary interval $I = [r,s] \subseteq [T]$ Let $\mathbf{x}^\star = \arg\min_{x \in \mathcal{K}} \sum_{t=r}^s \tilde{f}_t(x)$ and denote $\nabla_t \doteq \nabla \tilde{f}_t(x_t)$ for simplicity. By convexity we have

$$\tilde{f}_t(x_t) - \tilde{f}_t(x^\star) \leq \nabla_t^\top (x_t - x^\star) \tag{B.1}$$

By the Pythagorean theorem

$$\begin{aligned}
\|x_{t+1} - x^\star\|^2 &= \|\Pi_{\mathcal{K}}(x_t - \eta \nabla_t) - x^\star\|^2 \\
&\leq \|x_t - \eta \nabla_t - x^\star\|^2 \\
&\leq \|x_t - x^\star\|^2 + \eta^2 \|\nabla_t\|^2 - 2\eta \nabla_t^\top (x_t - x^\star)
\end{aligned} \tag{B.2}$$

Hence we can bound the interval regret as:

$$\begin{aligned}
\Rightarrow 2 \sum_{t=r}^{s} (f_t(x_t) - f_t(x^\star)) &\leq 2 \sum_{t=r}^{s} \nabla_t^\top (x_t - x^\star) && \textcolor{blue}{Eq. \ (B.1)} \\
&\leq \sum_{t=r}^{s} \left( \frac{\|x_t - x^\star\|^2 - \|x_{t+1} - x^\star\|^2}{\eta} + \eta G^2 \right) && \textcolor{blue}{Eq. \ (B.2)} \\
&= \frac{\|x_r - x^\star\|^2}{\eta} + |I| \eta G^2 \\
&\leq \frac{D^2}{\eta} + \eta |I| G^2
\end{aligned}$$

which yields the desired adaptive regret bound. $\qquad\square$

Using this simple fact and Lipschitzness we are able to easily prove the desired guarantee for Algorithm 4.

*Proof of Theorem B.2.* First note that Algorithm 4 is just doing gradient descent on the proxy convex loss functions $\tilde{f}_t : \mathcal{K} \to [0, 1]$. Hence, as long as we identify the gradient bound we can apply Fact B.1 to get a bound on $\tilde{f}$-regret. Observe that

$$
\begin{aligned}
|\tilde{f}_t(x) - \tilde{f}_t(y)| &= |f_t(x, \ldots, x) - f_t(y, \ldots, y)| \\
&\leq |f_t(x, \ldots, x) - f_t(y, \ldots, x)| + |f_t(y, x, \ldots, x) - f_t(y, y, \ldots, x)| + \\
&\quad \ldots + |f_t(y, \ldots, y, x) - f_t(y, \ldots, y)| \\
&\leq L(h+1)\|x - y\|
\end{aligned}
$$

so $\tilde{f}_t$ is $L(h+1)$-Lipschitz and hence has a gradient bound of $L(h+1)$. So we can apply Fact B.1 to get

$$
\sum_{t=r}^{s} \tilde{f}_t(x_t) - \min_{x \in \mathcal{K}} \sum_{t=r}^{s} \tilde{f}(x) \leq \frac{D^2}{\eta} + \eta |I| L^2 (h+1)^2 \tag{B.3}
$$

We can use this to bound the adaptive policy regret. First note that

$$
\|x_t - x_{t-1}\| = \|\eta \nabla \tilde{f}_t(x_{t-1})\| \leq \eta L(h+1)
$$

and by the triangle inequality

$$
\|x_t - x_{t-i}\| \leq \sum_{j=1}^{i} \|x_{t-j+1} - x_{t-j}\| \leq \eta L(h+1) \cdot i \tag{B.4}
$$

Using Eq. (B.4) and Lipschitzness we have:

$$
\begin{aligned}
f_t(x_{t-h}, \ldots, x_t) - \tilde{f}_t(x_t) &\leq L\|(x_{t-h}, \ldots, x_t) - (x_t, \ldots, x_t)\| \\
&\leq L \sqrt{\sum_{i=1}^{h} \|x_t - x_{t-i}\|^2} \\
&\leq \eta L^2 (h+1) \cdot \sqrt{\sum_{i=1}^{h} i^2} \\
&\leq \eta L^2 (h+1)^{5/2} \tag{B.5}
\end{aligned}
$$

$\square$

Combining everything we get

$$
\sum_{t=r}^{s} f_t(x_{t-h}, \ldots, x_t) - \min_{x \in \mathcal{K}} \sum_{t=r}^{s} \tilde{f}_t(x) = \underbrace{\sum_{t=r}^{s} f_t(x_{t-h}, \ldots, x_t) - \tilde{f}_t(x_t)}_{\text{(dist. to proxy loss [Eq. (B.5)])}} + \underbrace{\sum_{t=r}^{s} \tilde{f}_t(x_t) - \min_{x \in \mathcal{K}} \sum_{t=r}^{s} \tilde{f}_t(x_t)}_{(\tilde{f}\text{-regret [Eq. (B.3)])}}
$$

$$
\leq 2\eta L^2 (h+1)^{5/2} |I| + \frac{D^2}{\eta}
$$

## B.2 Proof of Theorem B.1

We first prove that the constructed loss function satisfies key properties for efficient optimization.

**Lemma B.2** (Convexity)**.** *The loss functions $\tilde{f}_t$ constructed in Appendix B of Algorithm 3 are convex in $M$.*

*Proof.* By definition we have that:

$$\hat{x}_t(M) = x_t^{\text{nat}} + \sum_{i=1}^{h} G_t^{[i]} u_{t-i}$$

$$= x_t^{\text{nat}} + \sum_{i=1}^{h} G_t^{[i]} \left( \sum_{j=0}^{m-1} M^{[j]} x_{t-i-j}^{\text{nat}} \right) \tag{B.6}$$

which is affine in $M$. Even more simply, we have $u_t(M) = \sum_{i=0}^{m-1} M^{[i]} x_{t-i}^{\text{nat}}$.

Since $\hat{x}_t(M)$ and $u_t(M)$ are affine, and, respectively, linear functions of $M$ and composition with the convex cost $c_t$ preserves convexity we get the desired property. $\qquad\square$

**Lemma B.3** (Lipschitzness). *The loss functions $f_t$ constructed in [Appendix B] of [Algorithm 3] are $L_f$ coordinate-wise Lipschitz for $L_f = 3LR_{nat}^2 R_G^2 R_{\mathcal{M}} \sqrt{m}$.*

*Proof.* Observe that by [Eq. (B.6)] we have $\|\hat{x}_t(M_{0:h})\| \leq R_{\text{nat}}(1 + R_G R_{\mathcal{M}})$. Straightforwardly, $\|u_t(M_h)\| \leq R_{\text{nat}} R_{\mathcal{M}}$ as well.

For an arbitrary $i \in \overline{0, h}$, denoting $M_{0:h} = (M_0, \ldots, M_i, \ldots M_h)$, $\tilde{M}_{0:h} = (M_0, \ldots, \tilde{M}_i, \ldots, M_h)$ and using the sub-quadratic lipschitzness of the costs we have:

$$|f_t(M_{0:h}) - f_t(\tilde{M}_{0:h})| = |c_t(\hat{x}_t(M_{0:h}), u_t(M_{0:h})) - c_t(\hat{x}_t(\tilde{M}_{0:h}), u_t(\tilde{M}_{0:h}))|$$

$$\leq 3LR_{\text{nat}} R_G R_{\mathcal{M}} \left\| G_t^{[h-i]} \left( \sum_{j=0}^{m-1} (M_i^{[j]} - \tilde{M}_i^{[j]}) x_{t-i-j}^{\text{nat}} \right) \right\| \quad (i < h)$$

$$\text{or} \leq 3LR_{\text{nat}} R_G R_{\mathcal{M}} \left( \sum_{j=0}^{m-1} (M_i^{[j]} - \tilde{M}_i^{[j]}) x_{t-i-j}^{\text{nat}} \right) \quad (i = h)$$

$$\leq 3LR_{\text{nat}}^2 R_G^2 R_{\mathcal{M}} \sqrt{m} \|M_i - \tilde{M}_i\|_F$$

so the function is coordinate-wise Lipschitz with constant $L_f = 3LR_{\text{nat}}^2 R_G^2 R_{\mathcal{M}} \sqrt{m}$. $\qquad\square$

**Lemma B.4** (Euclidean Diameter). *The euclidean diameter of $\mathcal{M}(m, R_{\mathcal{M}})$ is at most $D = 2\sqrt{m \cdot \min\{d_x, d_u\}} R_{\mathcal{M}}$.*

*Proof.* That for an arbitrary $M \in \mathcal{M}(m, R_{\mathcal{M}})$, we have

$$\|M\|_F = \sqrt{\sum_{i=0}^{m-1} \|M^{[i]}\|_F^2}$$

$$\leq \sqrt{m \cdot \max_{i \in [m-1]} \|M^{[i]}\|_F^2}$$

$$= \sqrt{m} \max_{i \in [m-1]} \min\{d_x, d_u\} \|M^{[i]}\|_{op}$$

$$\leq \sqrt{m \cdot \min\{d_x, d_u\}} \|M\|_{\ell_1, op}$$

$$\leq \sqrt{m \cdot \min\{d_x, d_u\}} R_{\mathcal{M}}$$

and the euclidean diameter is at most twice the maximal euclidean norm, concluding our statement. $\qquad\square$

The three lemmas above will allow us to use the results in [Appendix B.1] to obtain adaptive regret guarantees in terms of $f_t$ which truncated the effect on the state of actions further than $h$ in the past. To convert guarantees in terms of $f_t$ to ones in terms of $c_t$, we prove that the effect of the past is minimal:

**Lemma B.5** (Truncation Error). *For a changing DRC policy that acts according to $M_1, \ldots, M_t$ up to time $t$ we have that:*

$$c_t(x_t, u_t) - c_t\left(\hat{x}_t(M_{t-h:t-1}), u_t(M_t)\right) \leq 3LR_{nat}^2 R_{\mathcal{M}}^2 R_G \psi(h)$$

*Proof.* By the sub-quadratic Lipschitzness (and noting $\|u_t\|_2 \leq R_{\mathcal{M}} R_{\text{nat}}$, $\|x_t\|_2 \leq R_nat(1 + R_G R_{\mathcal{M}})$, and $u_t = u_t(M_t)$) we have:

$$
\begin{aligned}
c_t(x_t, u_t) - c_t\left(\hat{x}_t(M_{t-h:t-1}), u_t(M_t)\right) &\leq 3LR_{\text{nat}} R_G R_{\mathcal{M}} \|x_t(M_{1:t-1} - \hat{x}_t(M_{t-h:t-1})\| \\
&= 3LR_{\text{nat}} R_G R_{\mathcal{M}} \|\sum_{i=h}^{t-1} G_{t-1}^{[i]} u_{t-1-i}\| \\
&\leq 3LR_{\text{nat}}^2 R_{\mathcal{M}}^2 R_G \psi(h)
\end{aligned}
$$

$\square$

Having proven all these preliminary results the proof of the main theorem is immediate:

*Proof of Theorem B.1.* By the definition of the proxy loss in Line 7 of Algorithm 3, we can expand the regret of Algorithm 3 over interval $I = [r, s]$ as:

$$
\begin{aligned}
\text{Regret} &= \sum_{t=r}^{s} c_t(x_t, u_t) - \min_{M \in \mathcal{M}} \sum_{t=r}^{s} c_t(x_t^M, u_t^M) \\
&= \underbrace{\sum_{t=r}^{s} c_t(x_t, u_t) - \sum_{t=r}^{s} c_t(\hat{x}_t(M_{t-h:t-1}), u_t(M_t))}_{\text{(truncation error I)}} + \underbrace{\sum_{t=r}^{s} f_t(M_{t-h:t}) - \min_{M \in \mathcal{M}} \tilde{f}_t(M)}_{\text{(f-regret)}} \\
&\quad + \underbrace{\min_{M \in \mathcal{M}} \sum_{t=r}^{s} c_t(\hat{x}_t(M), u_t(M)) - \min_{M \in \mathcal{M}} \sum_{t=r}^{s} c_t(x_t^M, u_t^M)}_{\text{(truncation error II)}}
\end{aligned}
$$

The first truncation error is bounded directly by Lemma B.5. For the second truncation error, let $M^\star = \arg\min_{M \in \mathcal{M}} \sum_{t=r}^{s} c_t(x_t^M, u_t^M)$. Clearly we have

$$\min_{M \in \mathcal{M}} \sum_{t=r}^{s} c_t(\hat{x}_t(M), u_t(M)) \leq \sum_{t=r}^{s} c_t(\hat{x}_t(M^\star), u_t(M^\star))$$

and hence we can apply Lemma B.5 to bound:

$$
\begin{aligned}
\text{truncation error II} &= \min_{M \in \mathcal{M}} \sum_{t=r}^{s} c_t(\hat{x}_t(M), u_t(M)) - \min_{M \in \mathcal{M}} \sum_{t=r}^{s} c_t(x_t^M, u_t^M) \\
&\leq \sum_{t=r}^{s} c_t(\hat{x}_t(M^\star), u_t(M^\star)) - \sum_{t=r}^{s} c_t(x_t^{M^\star}, u_t^{M^\star}) \\
&\leq 3LR_{\text{nat}}^2 R_{\mathcal{M}}^2 R_G \psi(h)|I|
\end{aligned}
$$

Finally, due to Lemma B.2, Lemma B.3 and Lemma B.4 we can apply Theorem B.2 to get

$$\text{f-regret} \leq \frac{D^2}{\eta} + 2\eta L_f^2 (h+1)^{5/2}|I|$$

Summing everything up and plugging in the Lipschitz and diameter constants, we have:

$$\text{Regret} \leq 6LR_{\text{nat}}^2 R_{\mathcal{M}}^2 R_G \psi(h)|I| + \frac{4m \min\{d_x, d_u\} R_{\mathcal{M}}^2}{\eta} + 18\eta(L^2 R_{\text{nat}}^4 R_G^4 R_{\mathcal{M}}^2)m(h+1)^{5/2}|I|$$

Setting $\eta \doteq \frac{\sqrt{\min\{d_x, d_u\}}}{2LR_G^2 R_{\text{nat}}^2 (h+1)^{5/4}\sqrt{T}}$, we get

$$\text{Regret} \leq 6LR_{\text{nat}}^2 R_{\mathcal{M}}^2 R_G \psi(h)|I| + 17\sqrt{\min\{d_x, d_u\}}(LR_G^2 R_{\text{nat}}^2 R_{\mathcal{M}}^2) \cdot (h+1)^{5/4}m \cdot \sqrt{T}$$
$$\leq 6LR_{\text{sys}}^2 \left(3\sqrt{d_{\min}}m(h+1)^{5/4}\sqrt{T} + \psi(h)|I|\right)$$

where we denote $R_{\text{sys}} \doteq R_G R_{\mathcal{M}} R_{\text{nat}}$ and $d_{\min} \doteq \min\{d_x, d_u\}$.  $\square$

## C   Estimation of Time-Varying Vector Sequences

In this section we segway into the setting of online prediction under a partial information model. The goal is to estimate a sequence of vectors under limited noisy feedback where the feedback access is softly restricted via additional cost. As shown in the following section, this setting captures the *system identification* phase of controlling an unknown time-varying dynamical system. We first extensively study the simplified setting as below, and afterwards transfer our findings into meaningful results in control.

Formally, consider the following repeated game between a learner and an *oblivious* adversary: at each round $t \in [T]$, the adversary picks a target vector $\mathbf{z}_t^\star \in \mathcal{K}$ from a convex decision set $\mathcal{K}$ contained in a $0$-centered ball of radius $R_z$; simultaneously, the learner selects an estimate $\hat{\mathbf{z}}_t \in \mathcal{K}$ and suffers quadratic loss $f_t(\hat{\mathbf{z}}_t) = \|\hat{\mathbf{z}}_t - \mathbf{z}_t^\star\|^2$. The only feedback the learner has access to is via the following noisy and costly oracle.

**Oracle 2** (Noisy Costly Oracle). At each time $t \in [T]$, the learner selects a decision $b_t \in \{0, 1\}$ indicating whether a query is sent to the oracle. If $b_t = 1$, the learner receives an unbiased estimate $\tilde{\mathbf{z}}_t$ as response such that $\|\tilde{\mathbf{z}}_t\| \leq \tilde{R}_z$ and $\mathbb{E}[\tilde{\mathbf{z}}_t \mid \mathcal{F}_t, b_t = 1] = \mathbf{z}_t^\star$ where $\mathcal{F}_t$ is the filtration sigma algebra generated by the entire sequence $\mathbf{z}_{1:T}^\star$ and the past $\tilde{\mathbf{z}}_{1:t-1}, b_{1:t-1}$. A completed query results in a unit cost for the learner denoted $b_t$ as well by abuse of notation.

The idea behind this setting is to model a general estimation framework for a time-varying system which focuses only on exploration. Committing to exploration, however, cannot realistically be free hence the additional cost for the number of calls to Oracle 2. Our goal is to design an algorithm $\mathcal{A}$ that minimizes the quadratic loss regret along with the extra oracle cost, defined over each interval $I = [r, s] \subseteq [T]$ as

$$\text{Regret}_I(\mathcal{A}; \lambda) = \mathbb{E}\left[\sum_{t \in I} f_t(\hat{\mathbf{z}}_t)\right] - \min_{\mathbf{z} \in \mathcal{K}} \sum_{t \in I} f_t(\mathbf{z}) + \lambda\mathbb{E}\left[\sum_{t \in I} b_t\right], \qquad \text{(C.1)}$$

where $\lambda \geq 0$ is a scaling constant independent of the horizon $T$. For the $I = [T]$ entire interval, we use $T$ as a subscript instead of $[T]$. The expectation above is taken over both the (potential) randomness of the algorithm and the stochasticity of the oracle responses; it is taken in the round order $t = 1, \ldots, T$ at each round conditioning on the past iterations.

In terms of estimation itself, the metric to consider over interval $I$ is given by $\text{Regret}_I(\mathcal{A}; 0)$ that ignores the oracle call costs. Furthermore, we observe that the best-in-hindsight term in (C.1) is in fact a fundamental quantity of the vector sequence as defined below. This formulation will be used, and is more appropriate, when transferring our findings to the setting of control.

**Definition C.1.** Define the variability of a time-varying vector sequence $\mathbf{z}_{1:T}$ over an interval $I \subseteq [T]$ to be

$$\text{Var}_I(\mathbf{z}_{1:T}) = \frac{1}{|I|} \min_{\mathbf{z} \in \mathcal{K}} \sum_{t \in I} \|\mathbf{z} - \mathbf{z}_t\|^2 = \frac{1}{|I|} \sum_{t \in I} \|\bar{\mathbf{z}}_I - \mathbf{z}_t\|^2,$$

where $\bar{\mathbf{z}}_I = |I|^{-1} \sum_{t \in I} \mathbf{z}_t \in \mathcal{K}$ is the empirical average of the members of the sequence that correspond to $I$.

This definition concludes the setup of our abstraction to general estimation of vector sequences. Regarding algorithmic results, we first present a base method that achieves logarithmic regret over the entire trajectory $[1, T]$. The idea is for the learner to uniformly query Oracle 2 with probability $p$: once an estimate $\tilde{\mathbf{z}}_t$ is received, construct a stochastic gradient with expectation equal to the true gradient, and perform a gradient update. The algorithm is described in detail in Algorithm 5, and its guarantee given in the theorem below.

---

**Algorithm 5** Base Estimation Algorithm

---

1: **Input:** $p, \hat{\mathbf{z}}_1 \in \mathcal{K}$
2: **for** $t = 1, \ldots, T$ **do**
3:     **Play** iterate $\hat{\mathbf{z}}_t$
4:     **Draw/Receive** $b_t \sim \text{Bernoulli}(p)$
5:     **if** $b_t = 1$ **then**
6:         **Receive** estimate $\tilde{\mathbf{z}}_t$ from Oracle 2
7:         **Construct** importance weighted gradient $\tilde{\nabla}_t := \frac{1}{p}(\hat{\mathbf{z}}_t - \tilde{\mathbf{z}}_t)$
8:     **else**
9:         **Set** $\tilde{\nabla}_t = 0$.
10:     **Update** $\hat{\mathbf{z}}_{t+1} = \text{Proj}_{\mathcal{K}}(\hat{\mathbf{z}}_t - \eta_t \tilde{\nabla}_t), \eta_t = \frac{1}{t}$.

---

**Theorem C.1.** *Given access to queries from Oracle 2, with stepsizes $\eta_t = \frac{1}{t}$, Algorithm 5 enjoys the following regret guarantee:*

$$\text{Regret}_T(Algorithm\ 5; \lambda) \leq \frac{(R_z + \tilde{R}_z)^2 (1 + \log T)}{p} + \lambda p T . \tag{C.2}$$

*Proof of Theorem C.1.* To prove the bound in the theorem, we construct the following proxy loss functions: if $b_t = 1$ denote $\tilde{f}_t(\mathbf{z}) = \frac{1}{2p}\|\mathbf{z} - \tilde{\mathbf{z}}_t\|^2$, otherwise for $b_t = 0$ denote $\tilde{f}_t(\mathbf{z}) = 0$. The stochastic gradients of these functions at the current iterate can be written as $\nabla_{\mathbf{z}} \tilde{f}_t(\hat{\mathbf{z}}_t) = \tilde{\nabla}_t = \frac{\mathbb{I}\{b_t=1\}}{p}(\hat{\mathbf{z}}_t - \tilde{\mathbf{z}}_t)$ and are used by the algorithm in the update rule. The idealized gradients are $\nabla_t = (\hat{\mathbf{z}}_t - \mathbf{z}_t^\star)$ which we would use given access to the true targets $\mathbf{z}_t^\star$. Recall that $\mathcal{F}_t$ denotes the sigma-algebra generated by the true target sequence $\mathbf{z}_{1:T}^\star$, as well as randomness of the past rounds $b_{1:t-1}$ and $\tilde{\mathbf{z}}_{1:t-1}$. Note then that $\hat{\mathbf{z}}_t$ is $\mathcal{F}_t$ measurable. We characterize two essential properties of the stochastic gradients:

**Lemma C.1.** *Let $\bar{\mathbf{z}}^\star \in \mathcal{K}$ be the minimizer of $\sum_{t=1}^T \|\mathbf{z} - \mathbf{z}_t^\star\|^2$, i.e. empirical average of $\mathbf{z}_{1:T}^\star$. Then,*

$$\mathbb{E}[\langle \tilde{\nabla}_t, \hat{\mathbf{z}}_t - \bar{\mathbf{z}}^\star \rangle] = \mathbb{E}[\langle \nabla_t, \hat{\mathbf{z}}_t - \bar{\mathbf{z}}^\star \rangle] .$$

*Moreover, $\mathbb{E}[\|\tilde{\nabla}_t\|^2] \leq (R_z + \tilde{R}_z)^2/p$.*

*Proof.* Using the Oracle 2 assumption on $\tilde{\mathbf{z}}_t$, we get

$$\mathbb{E}[\tilde{\nabla}_t \mid \mathcal{F}_t] = \frac{1}{p}\mathbb{E}[\mathbb{I}\{b_t = 1\}(\hat{\mathbf{z}}_t - \tilde{\mathbf{z}}_t) \mid \mathcal{F}_t]$$

$$= \frac{1}{p}\mathbb{E}[\mathbb{I}\{b_t = 1\} \cdot \hat{\mathbf{z}}_t \mid \mathcal{F}_t] - \frac{1}{p}\mathbb{E}[\mathbb{I}\{b_t = 1\} \cdot \tilde{\mathbf{z}}_t \mid \mathcal{F}_t]$$

$$= \hat{\mathbf{z}}_t - \mathbb{E}[\tilde{\mathbf{z}}_t \mid \mathcal{F}_t, b_t = 1] \overset{(i)}{=} \hat{\mathbf{z}}_t - \mathbf{z}_t^\star = \nabla_t,$$

where $(i)$ uses the unbiasedness property of Oracle 2. Next, since $\bar{\mathbf{z}}^\star$ is determined by $\mathbf{z}_{1:T}^\star$ it is therefore $\mathcal{F}_t$ measurable for all $t$. Thus, $\hat{\mathbf{z}}_t - \bar{\mathbf{z}}^\star$ is $\mathcal{F}_t$ measurable, so

$$\mathbb{E}[\langle \tilde{\nabla}_t, \hat{\mathbf{z}}_t - \bar{\mathbf{z}}^\star \rangle] = \mathbb{E}\left[\langle \mathbb{E}[\tilde{\nabla}_t \mid \mathcal{F}_t], \hat{\mathbf{z}}_t - \bar{\mathbf{z}}^\star \rangle\right] = \mathbb{E}[\langle \nabla_t, \hat{\mathbf{z}}_t - \bar{\mathbf{z}}^\star \rangle] .$$

Finally, using the norm bound $\|\hat{\mathbf{z}}_t\| \leq R_z$ since $\hat{\mathbf{z}}_t \in \mathcal{K}$ and the assumption that $\|\tilde{\mathbf{z}}_t\| \leq \tilde{R}_z$ from Oracle 2, we conclude

$$\mathbb{E}[\|\tilde{\nabla}_t\|^2] = \frac{1}{p^2}\mathbb{E}[\mathbb{I}\{b_t = p\}\|\tilde{\mathbf{z}}_t - \hat{\mathbf{z}}_t\|^2] \leq \frac{1}{p^2}\mathbb{E}[\mathbb{I}\{b_t = p\}(R_z + \tilde{R}_z)^2] = \frac{(R_z + \tilde{R}_z)^2}{p} .$$

$\square$

The rest of the theorem proof mirrors that of Theorem 3.3 in [19] but accounting for the stochastic gradient. We can view Algorithm 5 as running online stochastic gradient descent over strongly convex functions on losses $\frac{1}{2}f_t(\mathbf{z}) = \frac{1}{2}\|\mathbf{z} - \mathbf{z}_t^\star\|^2$ with true gradient $\nabla_t$ and stochastic gradient $\tilde{\nabla}_t$ at the iterate $\hat{\mathbf{z}}_t$. Since the losses $\frac{1}{2}f_t$ are 1-strongly convex, $\mathbb{E}[b_t] = p$ and using the claim from Lemma C.1 we get,

$$\frac{1}{2}\text{Regret}_T = \frac{1}{2}\mathbb{E}\left[\sum_{t=1}^T (f_t(\hat{\mathbf{z}}_t) - f_t(\bar{\mathbf{z}}^\star) + \lambda \cdot b_t)\right] \le \mathbb{E}\left[\sum_{t=1}^T (\langle \nabla_t, \hat{\mathbf{z}}_t - \bar{\mathbf{z}}^\star\rangle - \frac{1}{2}\|\hat{\mathbf{z}}_t - \bar{\mathbf{z}}^\star\|^2)\right] + \frac{1}{2}\lambda p T$$

$$= \frac{1}{2}\mathbb{E}\left[\sum_{t=1}^T (2\langle \tilde{\nabla}_t, \hat{\mathbf{z}}_t - \bar{\mathbf{z}}^\star\rangle - \|\hat{\mathbf{z}}_t - \bar{\mathbf{z}}^\star\|^2)\right] + \frac{1}{2}\lambda p T.$$

The update rule is given as $\hat{\mathbf{z}}_{t+1} = \text{Proj}_{\mathcal{K}}(\hat{\mathbf{z}}_t - \eta_t \tilde{\nabla}_t)$, so from the Pythagorean theorem for the projection

$$\|\hat{\mathbf{z}}_{t+1} - \bar{\mathbf{z}}^\star\|^2 \le \|\hat{\mathbf{z}}_t - \eta_t \tilde{\nabla}_t - \bar{\mathbf{z}}^\star\|^2 = \|\hat{\mathbf{z}}_t - \bar{\mathbf{z}}^\star\|^2 + \eta_t^2\|\tilde{\nabla}_t\|^2 - 2\eta_t\langle\tilde{\nabla}_t, \hat{\mathbf{z}}_t - \bar{\mathbf{z}}^\star\rangle.$$

$$2\langle\tilde{\nabla}_t, \hat{\mathbf{z}}_t - \bar{\mathbf{z}}^\star\rangle \le \frac{\|\hat{\mathbf{z}}_t - \bar{\mathbf{z}}^\star\|^2 - \|\hat{\mathbf{z}}_{t+1} - \bar{\mathbf{z}}^\star\|^2}{\eta_t} + \eta_t\|\tilde{\nabla}_t\|^2.$$

Combining the above bounds results in

$$\frac{1}{2}\text{Regret}_T \le \frac{1}{2}\mathbb{E}\left[\sum_{t=1}^T \left(\frac{\|\hat{\mathbf{z}}_t - \bar{\mathbf{z}}^\star\|^2 - \|\hat{\mathbf{z}}_{t+1} - \bar{\mathbf{z}}^\star\|^2}{\eta_t} - \|\hat{\mathbf{z}}_t - \bar{\mathbf{z}}^\star\|^2\right) + \sum_{t=1}^T \eta_t\|\tilde{\nabla}_t\|^2\right] + \frac{1}{2}\lambda p T.$$

The telescoping sum inside the parentheses is equal to 0, the gradient term is bounded $\mathbb{E}[\|\tilde{\nabla}_t\|^2] \le \frac{(R_z + \tilde{R}_z)^2}{p}$ according to Lemma C.1 and the stepsize sum is bounded by $\sum_{t=1}^T \eta_t \le 1 + \log T$, yielding the final result

$$\text{Regret}_T(Algorithm\ 5; \lambda) \le \frac{(R_z + \tilde{R}_z)^2(1 + \log T)}{p} + \lambda p T.$$

$\square$

## C.1 Adaptive Regret Bound

The guarantee in Theorem C.1 ensures that the predicted sequence $\hat{\mathbf{z}}_{1:T}$ performs comparably to the empirical mean $\bar{\mathbf{z}}^\star$ of the entire target sequence $\mathbf{z}_{1:T}^\star$. However, that doesn't imply much about the performance of Algorithm 5 on a given local interval $I \subseteq [T]$ since $\bar{\mathbf{z}}_I^\star$ can be very different from $\bar{\mathbf{z}}^\star$. Hence, we would like to extend our results to hold for any interval $I$, i.e. derive adaptive regret results as introduced in [21]. To do so we will use the approach of [21] using Algorithm 5 as a subroutine. The resulting algorithm, presented in Algorithm 6, suffers only a logarithmic computational overhead over Algorithm 5 with its performance guarantee stated in the theorem below.

**Theorem C.2.** *Taking the base estimation algorithm $\mathcal{A}$ to be Algorithm 5 and given access to queries from Oracle 2, Algorithm 6 enjoys the following guarantee:*

$$\forall I = [r, s] \subseteq [T], \quad \text{Regret}_I(Algorithm\ 6; \lambda) \le \frac{2(R_z + \tilde{R}_z)^2(1 + \log s \cdot \log |I|)}{p} + \lambda p|I|.$$
(C.3)

**Corollary C.1.** *The estimation error over each interval $I = [r, s] \subseteq [T]$ is bounded as follows,*

$$\mathbb{E}\left[\sum_{t \in I}\|\hat{\mathbf{z}}_t - \mathbf{z}_t^\star\|^2\right] \le \text{Var}_I(\mathbf{z}_{1:T}^\star) + \frac{2(R_z + \tilde{R}_z)^2(1 + \log s \cdot \log |I|)}{p}.$$

*Proof of Theorem C.2.* First observe that $\tilde{\ell}_t$ is $\alpha$-exp concave with $\alpha = \frac{p}{(R_z + \tilde{R}_z^2)}$. This is evident given its construction: $\tilde{\ell}_t(\mathbf{z}) = \frac{1}{2p}\|\mathbf{z} - \tilde{\mathbf{z}}_t\|^2$ with $\|\mathbf{z}\| \le R_z$ since $\mathbf{z} \in \mathcal{K}$ and $\|\tilde{\mathbf{z}}_t\| \le \tilde{R}_z$ according

---
**Algorithm 6** Adaptive Estimation Algorithm
---
1: **Input:** parameter $p$, decision set $\mathcal{K}$, base estimation algorithm $\mathcal{A}$, $\hat{\mathbf{z}}_1 \in \mathcal{K}$
2: **Initialize:** $\mathcal{A}_1 \leftarrow \mathcal{A}(p, \hat{\mathbf{z}}_1)$, working set $\mathcal{S}_1 = \{1\}$, $q_1^{(1)} = 1$, parameter $\alpha = \frac{p}{(R_z + \bar{R}_z)^2}$
3: **for** $t = 1, \ldots, T$ **do**
4:     **Compute** predictions $\hat{\mathbf{z}}_t^{(i)} \leftarrow \mathcal{A}_i$ for $i \in \mathcal{S}_t$
5:     **Play** iterate $\hat{\mathbf{z}}_t = \sum_{i \in \mathcal{S}_t} q_t^{(i)} \hat{\mathbf{z}}_t^{(i)}$
6:     **Draw/Receive** $b_t \sim \text{Bernoulli}(p)$
7:     **if** $b_t = 1$ **then**
8:         **Request** estimate $\tilde{\mathbf{z}}_t$ from Oracle 2
9:         **Let** $\tilde{\ell}_t(\mathbf{z}) = \frac{1}{2p} ||\mathbf{z} - \tilde{\mathbf{z}}_t||^2$
10:     **else**
11:         **Let** $\tilde{\mathbf{z}}_t \leftarrow \emptyset$ and $\tilde{\ell}_t(\mathbf{z}) = 0$
12:     **Update** expert algorithms $\mathcal{A}_i(b_t, \tilde{\mathbf{z}}_t)$ for all $i \in \mathcal{S}_t$
13:     **Form** new set $\tilde{\mathcal{S}}_{t+1} = (i)_{i \in \mathcal{S}_t}$
14:     **Construct** proxy new weights $\bar{q}_{t+1}^{(i)} = \frac{t}{t+1} \cdot \frac{q_t^{(i)} e^{-\alpha \tilde{\ell}_t(\hat{\mathbf{z}}_t^{(i)})}}{\sum_{j \in \mathcal{S}_t} q_t^{(j)} e^{-\alpha \tilde{\ell}_t(\hat{\mathbf{z}}_t^{(j)})}}$ for all $i \in \mathcal{S}_t$
15:     **Add** new instance $\tilde{\mathcal{S}}_{t+1} \leftarrow \tilde{\mathcal{S}}_{t+1} \cup t + 1$ for arbitrary $\mathcal{A}_{t+1} \leftarrow \mathcal{A}(p, \hat{\mathbf{z}}_1^{(t+1)} = \hat{\mathbf{z}}_1)$ with $\bar{q}_{t+1}^{(t+1)} = \frac{1}{t+1}$
16:     **Prune** $\tilde{\mathcal{S}}_{t+1}$ to form $\mathcal{S}_{t+1}$
17:     **Normalize** $q_{t+1}^{(i)} = \frac{\bar{q}_{t+1}^{(i)}}{\sum_{j \in \mathcal{S}_{t+1}} \bar{q}_{t+1}^{(j)}}$
---

to Oracle 2. The rest of the algorithm uses the approach of [21], in particular Algorithm 1, over exp concave functions to derive the guarantee in the theorem statement.

We note that Claim 3.1 in [21] holds identically in our case, i.e. for any $I = [r, s]$ the regret of Algorithm 6 with respect to $\mathcal{A}_r$ is bounded by $\frac{2}{\alpha}(\ln r + \ln |I|)$ if $\mathcal{A}_r$ stays in the working set. We combine this fact with the bound given in Theorem C.1 to get that Algorithm 6 enjoys regret $\frac{3}{\alpha}(\log r + \log |I|)$ over $I = [r, s]$ if $\mathcal{A}_r$ stays in the working set $\mathcal{S}_t$ throughout $I$. Finally, an induction argument along with the working set properties detailed in Appendix C.1.1 identical to that of Lemma 3.2 in [21] yields the desired result for $\tilde{\ell}_t$. Notice that this is our desired result in expectation,

**Observation C.2.** *We have the following identity for any $t$ and $r \le t$:*

$$\mathbb{E}\left[\tilde{\ell}_t(\hat{\mathbf{z}}_t) - \tilde{\ell}_t(\hat{\mathbf{z}}_t^{(r)})\right] = \mathbb{E}\left[\ell_t(\hat{\mathbf{z}}_t) - \ell_t(\hat{\mathbf{z}}_t^{(r)})\right]$$

*Proof of Observation C.2.* We can expand:

$$2\mathbb{E}\left[\tilde{\ell}_t(\hat{\mathbf{z}}_t) - \tilde{\ell}_t(\hat{\mathbf{z}}_t^{(r)})\right] = 2p \cdot \mathbb{E}\left[\tilde{\ell}_t(\hat{\mathbf{z}}_t) - \tilde{\ell}_t(\hat{\mathbf{z}}_t^{(r)})|b_t = 1\right] + 2(1-p) \cdot \mathbb{E}\left[\tilde{\ell}_t(\hat{\mathbf{z}}_t) - \tilde{\ell}_t(\hat{\mathbf{z}}_t^{(r)})|b_t = 0\right]$$

$$= p \cdot \mathbb{E}\left[\frac{1}{p} \cdot \left(||\hat{\mathbf{z}}_t||^2 + \langle\hat{\mathbf{z}}_t, \tilde{\mathbf{z}}_t\rangle + ||\tilde{\mathbf{z}}_t||^2 - ||\hat{\mathbf{z}}_t^{(r)}||^2 - \langle\hat{\mathbf{z}}_t^{(r)}, \tilde{\mathbf{z}}_t\rangle - ||\tilde{\mathbf{z}}_t||^2\right)\right] + 0$$

$$= \mathbb{E}\left[||\hat{\mathbf{z}}_t||^2 - ||\hat{\mathbf{z}}_t^{(r)}||^2\right] + \mathbb{E}\left[\langle\hat{\mathbf{z}}_t - \hat{\mathbf{z}}_t^{(r)}, \tilde{\mathbf{z}}_t\rangle\right]$$

By the linearity of expectation, the fact that $\hat{\mathbf{z}}_t, \hat{\mathbf{z}}_t^{(r)}$ are completely determined given $\mathcal{F}_{t-1}$, and the law of total expectation we have that

$$\mathbb{E}\left[\langle\hat{\mathbf{z}}_t - \hat{\mathbf{z}}_t^{(r)}, \tilde{\mathbf{z}}_t\rangle\right] = \mathbb{E}\left[\mathbb{E}\left[\langle\hat{\mathbf{z}}_t - \hat{\mathbf{z}}_t^{(r)}, \tilde{\mathbf{z}}_t\rangle|\mathcal{F}_{t-1}\right]\right]$$

$$= \mathbb{E}\left[\langle\hat{\mathbf{z}}_t - \hat{\mathbf{z}}_t^{(r)}, \mathbf{z}_t^\star\rangle\right]$$

Plugging this in above we, adding and subtracting $\|\mathbf{z}_t^\star\|^2$, and rearranging we have:

$$2\mathbb{E}\left[\tilde{\ell}_t(\hat{\mathbf{z}}_t) - \tilde{\ell}_t(\hat{\mathbf{z}}_t^{(r)})\right] = \mathbb{E}\left[\|\hat{\mathbf{z}}_t\|^2 - \|\hat{\mathbf{z}}_t^{(r)}\|^2 + \langle\hat{\mathbf{z}}_t - \hat{\mathbf{z}}_t^{(r)}, \mathbf{z}_t^\star\rangle + \|\mathbf{z}\|^2 - \|\mathbf{z}\|^2\right]$$

$$= \mathbb{E}\left[\|\hat{\mathbf{z}}_t\|^2 + \langle\hat{\mathbf{z}}_t, \mathbf{z}_t^\star\rangle + \|\mathbf{z}\|^2 - \|\hat{\mathbf{z}}_t^{(r)}\|^2 - \langle\hat{\mathbf{z}}_t^{(r)}, \mathbf{z}_t^\star\rangle - \|\mathbf{z}\|^2\right]$$

$$= 2\mathbb{E}\left[\ell_t(\hat{\mathbf{z}}_t) - \ell_t(\hat{\mathbf{z}}_t^{(r)})\right]$$

as desired. $\qquad\square$

Combining Observation C.2 with the fact that $\tilde{\ell}_t$ are $\alpha$-exp concave for $\alpha = \frac{p}{(R_z+\bar{R}_z)^2}$ we conclude the final statement of Theorem C.2. $\qquad\square$

### C.1.1 Working Set Construction

Our Algorithm 6 makes use of the working sets $\{\mathcal{S}_t\}_{t\in[T]}$ along with its properties in Claim C.3. In this section, we show the explicit construction of these working sets as in [21] and prove the claim.

**Claim C.3.** *The following properties hold for the working sets $S_t$ for all $t \in [T]$: (i) $|\mathcal{S}_t| = O(\log T)$; (ii) $[s, (s+t)/2] \cap \mathcal{S}_t \neq \emptyset$ for any $s \in [t]$; (iii) $\mathcal{S}_{t+1}\backslash\mathcal{S}_t = \{t+1\}$; (iv) $|\mathcal{S}_t\backslash\mathcal{S}_{t+1}| \leq 1$.*

For any $i \in [T]$, let it be given as $i = r2^k$ with $r$ odd and $k$ nonnegative. Denote $m = 2^{k+2} + 1$, then $i \in S_t$ if and only if $t \in [i, i+m]$. This fully describes the construction of the working sets $\{S_t\}_{t\in[T]}$, and we proceed to prove its properties.

*Proof of Claim C.3.* For all $t \in [T]$ we show the following properties of the working sets $S_t$.

(i) $|S_t| = O(\log T)$: if $i \in S_t$ then $1 \leq i = r2^k \leq t$ which implies that $0 \leq k \leq \log_2 t$. For each fixed $k$ in this range, if $r2^k = i \in S_t$ then $i \in [t-2^{k+2}-1, t]$ by construction. Since $[t-2^{k+2}-1, t]$ is an interval of length $2^{k+2} + 2 = 4 \cdot 2^k + 2$, it can include at most 3 numbers of the form $r2^k$ with $r$ odd. Thus, there is at most 3 numbers $i = r2^k \in S_t$ for each $0 \leq k \leq \log_2 t$ which means that $|S_t| = O(\log t) = O(\log T)$.

(ii) $[s, (s+t)/2] \cap S_t \neq \emptyset$ for all $s \in [t]$: this trivially holds for $s = t-1, t$. Let $2^l \leq (t-s)/2$ be the largest such exponent of 2. Since the size of the interval $[s, (s+t)/2]$ is $\lfloor(t-s)/2\rfloor$, then there exists $u \in [s, (s+t)/2]$ that divides $2^l$. This means that the corresponding $m \geq 2^{l+2} + 1 > t - s$ for $u \geq s$ is large enough so that $t \in [u, u+m]$, and consequently, $u \in S_t$.

(iii) $S_{t+1}\backslash S_t = \{t+1\}$: let $i \in S_{t+1}$ and $i \notin S_t$, which is equivalent to $t+1 \in [i, i+m]$ and $t \notin [i, i+m]$. Clearly, $i = t+1$ satisfies these conditions and is the only such number.

(iv) $|S_t\backslash S_{t+1}| \leq 1$: suppose there exist two $i_1, i_2 \in S_t\backslash S_{t+1}$. This implies that $i_1 + m_1 = t = i_2 + m_2$ which in turn means $2^{k_1}(r_1 + 4) = 2^{k_2}(r_2 + 4)$. Since both $r_1 + 4, r_2 + 4$ are odd, then $k_1 = k_2$, and consequently, $r_1 = r_2$ resulting in $i_1 = i_2$. Thus, there can not exist two different members of $S_t\backslash S_{t+1}$ which concludes that $|S_t\backslash S_{t+1}| \leq 1$. $\qquad\square$

## C.2 No Strong Adaptivity

Notice that even though the guarantee of Theorem C.2 applies to all intervals $I$, it does not entail meaningful guarantees for all. The reason is the choice of parameter $p$: if one wishes to optimize $\text{Regret}_T$ then $p = \mathcal{O}(T^{-1/2})$ implies $\mathcal{O}(\sqrt{T})$ regret, but this choice is meaningless for intervals with length $|I| << \sqrt{T}$; on the other hand, optimizing the bound for small intervals leads to large bounds for the entire horizon. One might then ask whether there exist methods with *strongly adaptive* guarantees, and we answer this question with a negative.

**Theorem C.3.** *For any $\gamma > 0$ and oracle cost $\lambda > 0$, there exists no online algorithm $\mathcal{A}$ with feedback access to Oracle 2 that enjoys the following strongly adaptive regret guarantee:* $\text{Regret}_I(\mathcal{A}; \lambda) = \tilde{\mathcal{O}}(|I|^{1-\gamma})$.

*Proof.* The proof of this impossibility results follows a simple construction: the idea behind it is that strongly adaptive guarantees imply both large and small amount of exploration. Let us suppose

there exists such an algorithm $\mathcal{A}$ and arrive at a contradiction: $\forall I = [r, s] \subseteq [T]$ algorithm $\mathcal{A}$ has a regret bound $\text{Regret}_I(\mathcal{A}; \lambda) \leq C \cdot |I|^{1-\gamma} = \tilde{\mathcal{O}}(|I|^{1-\gamma})$ over any oblivious sequence $\mathbf{z}^\star_{1:T}$ where $C$ depends on problem parameters, $\lambda$ and $\log T$.

Construct the following oblivious sequence: let $k = T^{1-\gamma/2}$ and $I_1, \dots, I_k$ be consecutive disjoint intervals such that $\cup_{j \in [k]} I_j = [T]$, $I_j \cap I_l = \emptyset$ for all $j \neq l$, and $|I_j| = T/k = T^{\gamma/2}$ for all $j \in [k]$ (w.l.o.g. we assume $T$ divides $k$). Now for each interval $I_j$, $j \in [k]$, sample a fresh $q_j \in \{\pm 1\} \sim Rad(1/2)$ and let $\mathbf{z}^\star_t = q_j$ for all $t \in I_j$.

According to the assumed guarantee, the overall regret is bounded as $\text{Regret}_T(\mathcal{A}; \lambda) leq C \cdot T^{1-\gamma}$ which by definition implies that $\sum_{t=1}^{T} b_t \leq \frac{C}{\lambda} T^{1-\gamma} < k$ where the last inequality is true for sufficiently large horizon $T$. Since there are $k$ consecutive disjoint intervals $I_1, \dots, I_k$ and less than $k$ overall calls to Oracle 2, there exists an interval $I \in \{I_1, \dots, I_k\}$ such that $\sum_{t \in I} b_t = 0$.

On the other hand, the assumed guarantee for $\mathcal{A}$ implies that the interval $I$ of size $|I| = T^{\gamma/2}$ enjoys sublinear regret, i.e. $\text{Regret}_I(\mathcal{A}; \lambda) = o(|I|)$. We show that this is a contradiction given that $\sum_{t \in I} b_t = 0$. As there were no oracle calls for the interval $I$, the predictions of $\mathcal{A}$, $\hat{\mathbf{z}}_t$ over $t \in I$, are independent from the Rademacher sample of the interval $q_I$: this is true since the samples for each interval in $I_1, \dots, I_k$ are independent. Therefore, $\hat{\mathbf{z}}_t \perp q_I$ for all $t \in I$ which means that since the best loss in hindsight over $I$ is equal to 0 as $\mathbf{z}^\star_t = q_I$ for all $t \in I$,

$$\text{Regret}_I(\mathcal{A}; \lambda) \geq \mathbb{E}_{q_I}\left[\sum_{t \in I} \ell_t(\hat{\mathbf{z}}_t)\right] = \sum_{t \in I} \mathbb{E}_{q_I}[\|\hat{\mathbf{z}}_t - q_I\|^2] = \Omega(|I|) .$$

Hence, for the interval $I$, the regret of $\mathcal{A}$ cannot be sublinear, which contradict the assumption that $\mathcal{A}$ exhibits strongly adaptive guarantees. This concludes that no strongly adaptive online algorithm exists in the described partial information model. $\qquad\square$

# D   Adaptive Regret for Control of Changing Unknown Dynamics

In this section we give our full control algorithm which attains sublinear regret with respect to $\Pi_{\text{drc}}$ up to an additive system variability term. A key component is the system estimation for which we will use Algorithm 6 and its guarantees from Appendix C. More specifically, our algorithm is based on the canonical explore-exploit approach: it explores with some probability $p$ by inputting random controls into the system, and otherwise outputs a control according to DRC-OGD (Algorithm 3). Note that due to the long-term consequences which appear in control, we need to explore for $h$ consecutive steps in order to get an estimate for the $h$-truncation of the Markov operator. Hence, our algorithm will determine whether it explores or exploits in blocks of length $h$. Furthermore, we will define the set of Markov operator of length $h$ and $\ell_1, op$-norm bounded by $R_G$ as $\mathcal{G}(h, R_G)$:

$$\mathcal{G}(h, R_G) \doteq \{G = (G^{[0]}, \dots, G^{[h-1]}) \in \mathbb{R}^{h \times d_x \times d_u} \text{ s.t. } \sum_{i=0}^{h-1} \|G\|_{op} \leq R_G\}$$

**Remark D.1.** Note that the radius of $\mathcal{G}(h, R_G)$ is bounded by $\bar{R}_G = \sqrt{h \cdot d_{\min}} R_G$ where $d_{\min} = \min\{d_x, d_u\}$.

*Proof of Remark D.1.* $\forall G \in \mathcal{G}(h, R_G)$, we have:

$$\|G\|_F = \sqrt{\sum_{i=0}^{h-1} \|G^{[i]}\|_F^2}$$
$$\leq \sqrt{h} \sqrt{\max_i \|G^{[i]}\|_F^2}$$
$$\leq \sqrt{h \min\{d_x, d_u\}} \max_i \|G^{[i]}\|_{op}$$
$$\leq \sqrt{h \min\{d_x, d_u\}} R_G$$

and denoting $d_{\min} = \min\{d_x, d_u\}$ yields the result. $\qquad\square$

**Remark D.2.** By abuse of notation, we will consider $G^{[i]} \doteq 0$ for $G \in \mathcal{G}(h, R_G)$.

**Remark D.3.** For simplicity, we assume $T$ divisible by $h$ (this is w.l.o.g. up to an extra $O(h)$ cost which for us is negligble).

We spell out the full procedure in Algorithm 7 and give its guarantee below in Theorem D.1.

---

**Algorithm 7** DRC-OGD with Exploration

---

1: **Input:** $p, h, \hat{G}_0, \mathcal{A} \leftarrow$ *Algorithm* $6(p, \hat{G}_0, \mathcal{G}(h, R_G)), \mathcal{C} \leftarrow$ *Algorithm* $3(\eta, m, R_{\mathcal{M}})$
2: **for** $\tau_1 = 0, \ldots, T/h - 1$ **do**
3:      Request $\hat{G}_{\tau_1 \cdot h + 1} \leftarrow \mathcal{A}$ and set $\hat{G}_{\tau_1 \cdot h + 2}, \ldots, \hat{G}_{(\tau_1+1) \cdot h} \leftarrow \hat{G}_{\tau_1 \cdot h + 1}$
4:      Draw $b_{\tau_1+1} \sim \text{Bernoulli}(p)$
5:      **for** $\tau_2 = 1, \ldots, h$ **do**                             $\triangleright$ let $t \doteq \tau_1 \cdot h + \tau_2$
6:          **if** $b_{\tau_1} = 1$ **then**
7:              Play control $u_t \sim \{\pm 1\}^{d_u}$
8:          **else**
9:              Play control $u_t \leftarrow \mathcal{C}$
10:          Suffer cost $c_t(x_t, u_t)$, observe new state $x_{t+1}$
11:          Extract $\hat{x}_{t+1}^{\text{nat}} = \text{Proj}_{\mathbb{B}_{R_{\text{nat}}}} \left[ x_{t+1} - \sum_{i=0}^{h-1} \hat{G}_t^{[i]} u_{t-i} \right]$
12:          Update $\mathcal{C} \leftarrow (c_t, \hat{G}_t, \hat{x}_{t+1}^{\text{nat}})$
13:      **if** $b_{\tau_1} = 1$ **then**
14:          Let $\tilde{G}_{(\tau_1+1) \cdot h}^{[i]} = x_{(\tau_1+1) \cdot h + 1} u_{(\tau_1+1) \cdot h - i}^\top$ for $i = \overline{0, h-1}$
15:      **else**
16:          Let $\tilde{G}_{(\tau_1+1) \cdot h} \leftarrow \emptyset$
17:      Update $\mathcal{A} \leftarrow (b_{\tau_1+1}, \tilde{G}_{(\tau_1+1) \cdot h})$

---

**Theorem D.1.** *For* $h = \dfrac{\log T}{\log \rho^{-1}}$, $p = T^{-1/3}$ *and* $m \leq \sqrt{T}$, *on any contiguous interval* $I \subseteq [T]$,
*Algorithm 2 enjoys the following adaptive regret guarantee[2]:*

$$\mathbb{E}\left[\text{Regret}_I(\text{ADA-CTRL}); \Pi_{\text{drc}}(m, R_M)\right] \leq \widetilde{\mathcal{O}}^\star \left( Lm \left( |I| \sqrt{\mathbb{E}[\text{Var}_I(\mathbf{G})]} + d_u T^{2/3} \right) \right)$$

The proof of this theorem will proceed in terms of a quantity which we call *total system variability* which captures the total (rather than average) deviation from the mean operator for each interval. More precisely,

**Definition D.1.** Define the *total* system variability of an LTV dynamical system with Markov operators $\mathbf{G} = G_{1:T}$ over a contiguous interval $I \subseteq [T]$ to be

$$\text{Var}_I^{\text{tot}}(\mathbf{G}) = \min_G \sum_{t \in I} \|G - G_t\|_{\ell_2, F}^2 = \sum_{t \in I} \|G_I - G_t\|_{\ell_2, F}^2,$$

where $\| \cdot \|_{\ell_2, F}$ indicates the $\ell_2$ norm of the fully vectorized operator and $G_I = |I|^{-1} \sum_{t \in I} G_t$ is the empirical average of the operators that correspond to $I$.

### D.1 Estimation of the Markov Operator

Note that the estimation component of Algorithm 7 directly operates in the setting of Appendix C and effectively solves the problem of adaptively estimating the sequence $\bar{G}_{1 \cdot h}, \ldots, \bar{G}_{(T/h) \cdot h}$ of the $h$-truncations of the true Markov operators $G_{1 \cdot h}, \ldots, G_{(T/h) \cdot h}$. To formally be able to apply Theorem C.2, we first show that the estimates sent to Algorithm 6 satisfy the properties of Oracle 2.

**Claim D.1.** *The estimators* $\tilde{G}_{\tau_1 \cdot h}$, $\tau_1 = \overline{1, T/h}$ *satisfy the properties of Oracle 2 with* $\tilde{R}_G = \sqrt{h \cdot d_u}(R_{\text{nat}} + R_G \max\{\sqrt{d_u}, R_{\text{nat}} R_{\mathcal{M}}\})$.

*Proof.* There are only two things to prove:

---

[2]For precise constants please see Eq. (D.2).

1. **Boundedness.** Because we clip the nature's x estimates $\hat{x}_t^{\text{nat}}$ to $R_{\text{nat}}$ and when the DAC policy lies in $\mathcal{M}(m, R_{\mathcal{M}})$, we have that if $b_t = 0$, $||u_t|| \leq R_{\text{nat}}R_{\mathcal{M}}$. If $u_t$ is an exploratory action then $||u_t|| = \sqrt{d_u} \leq \max\{\sqrt{d_u}, R_{\text{nat}}R_{\mathcal{M}}\}$ by design. By the equation of the progression of the state, we have that

$$||x_t|| \leq R_{\text{nat}} + R_G \max\{\sqrt{d_u}, R_{\text{nat}}R_{\mathcal{M}}\} \tag{D.1}$$

and by Cauchy Scwarz we get

$$||\tilde{G}_{\tau_1 \cdot h}^{[i]}||_F = ||u_{\tau_1 \cdot h-i}^\top x_{\tau_1 \cdot h+1}|| \leq \sqrt{d_u}(R_{\text{nat}} + R_G \max\{\sqrt{d_u}, R_{\text{nat}}R_{\mathcal{M}}\})$$

and hence

$$||\tilde{G}_{\tau_1 \cdot h}||_F \leq \sqrt{h \cdot d_u}(R_{\text{nat}} + R_G \max\{\sqrt{d_u}, R_{\text{nat}}R_{\mathcal{M}}\})$$

2. **Unbiasedness.** Plugging in $x_{\tau_1 \cdot h} = x_{\tau_1 \cdot h}^{\text{nat}} + \sum_{i=0}^t G_{\tau_1 \cdot h}^{[i]} u_{\tau_1 \cdot h-i}$, we get exactly that $\mathbb{E}[\tilde{G}_{\tau_1 \cdot h}^{[i]}] = G_{\tau_1 \cdot h}^{[i]}$. Since this holds for the selected truncation $h$, we have $\mathbb{E}[\tilde{G}_{\tau_1 \cdot h}] = \bar{G}_{\tau_1 \cdot h}$ for $\bar{G}$ as defined.

$\square$

Hence we can simply apply the guarantees of [Appendix C](#) to obtain the following guarantee:

**Corollary D.1.** *On any interval $J = [k, l] \subseteq [T/h]$, we have that:*

$$\mathbb{E}\left[\sum_{\tau=k}^l ||\tilde{G}_{\tau \cdot h} - \bar{G}_{\tau \cdot h}||_F^2\right] \leq \text{Var}_{J \cdot h}^{\text{tot}}(\bar{G}_{1:T}) + \text{Regret}_J(\mathcal{A}(p, \bar{R}_G, \tilde{R}_G; 0))$$

*where we use $J \cdot h$ to denote the set $[k \cdot h, \ldots, l \cdot h]$.*

However, to properly analyze the additional regret introduced in the control framework by our estimation error, we need to convert [Corollary D.1](#) into a guarantee in terms of $\ell_1, op$ norm which holds for any contiguous interval $I = [r, s] \subseteq [T]$. This step is rather straightforward and only relies on a few basic properties/observations which we collect in [Observation D.2](#) below.

**Observation D.2.** *We will make the following observations:*

1. *$||A||_{op} \leq ||A||_F$ for any matrix $A$,*

2. *$\text{Var}_J^{\text{tot}}(\mathbf{z}_{1:T}) \leq \text{Var}_I^{\text{tot}}(\mathbf{z}_{1:T})$ for any set of indices $J \subseteq I$, and any sequence $\mathbf{z}_{1:T}$,*

3. *$||\hat{G}_t - \bar{G}_t||_F^2 = \sum_{i=1}^h ||\hat{G}_t^{[i]} - \bar{G}_t^{[i]}||_F^2$,*

4. *$||G_t - \bar{G}_t||_{\ell_1, op} \leq \psi(h)$,*

5. *For any $I = [r, s]$, $\sum_{t=r+1}^s ||\bar{G}_t - \bar{G}_{t-1}||_F^2 \leq 4\text{Var}_I^{\text{tot}}(\bar{G}_{1:T}) \leq 4\text{Var}_I^{\text{tot}}(\mathbf{G})$.*

*Proof.* Properties 1-4 follow from the definitions of the relevant quantities, or are general well-known facts. For 5, by the triangle inequality, we have that, for any $\bar{G}$,

$$||\bar{G}_t - \bar{G}_{t-1}||_F \leq ||\bar{G}_t - \bar{G}||_F + ||\bar{G}_{t-1} - \bar{G}||_F$$

$$\Rightarrow ||\bar{G}_t - \bar{G}_{t-1}||_F^2 \leq 2(||\bar{G}_t - \bar{G}||_F^2 + ||\bar{G}_{t-1} - \bar{G}||_F^2)$$

summing the above from $r+1$ to $s$ and taking $\bar{G}$ to be the sample mean over $I$ yields the desired result. The second inequality simply follows by the fact that truncation can only decrease variance. $\square$

As a first step, we first bound the Frobenius norm error of the truncated operators over an arbitrary contiguous interval $I = [r, s] \subseteq [T]$.

**Lemma D.3.** *On any interval $I = [r, s] \subseteq [T]$ with $r - 1, s$ divisible by $h$[3], we can bound the Frobenius estimation error of the truncated operators as:*

$$\mathbb{E}\left[\sum_{t=r}^{s} ||\hat{G}_t - \bar{G}_t||_F^2\right] \leq 10h^2\text{Var}_I^{\text{tot}}(\mathbf{G}) + 2h\text{Regret}_I(\mathcal{A}(p, \bar{R}_G, \tilde{R}_G; 0))$$

*Proof.* By Algorithm 7, we can write

$$\sum_{t=r}^{s} ||\hat{G}_t - \bar{G}_t||_F^2 = \sum_{\tau_1=(r-1)/h}^{s/h-1} \sum_{\tau_2=1}^{h} ||\hat{G}_{\tau_1 \cdot h + \tau_2} - \bar{G}_{\tau_1 \cdot h + \tau_2}||_F^2$$

$$= \sum_{\tau_1=(r-1)/h}^{s/h-1} \sum_{\tau_2=1}^{h} ||\hat{G}_{(\tau_1+1)\cdot h} - \bar{G}_{\tau_1 \cdot h + \tau_2}||_F^2$$

$$\leq 2 \sum_{\tau_1=(r-1)/h}^{s/h-1} \sum_{\tau_2=1}^{h} \left( ||\hat{G}_{(\tau_1+1)\cdot h} - \bar{G}_{(\tau_1+1)\cdot h}||_F^2 + ||\bar{G}_{(\tau_1+1)\cdot h} - \bar{G}_{\tau_1 \cdot h + \tau_2}||_F^2 \right)$$

$$= 2h \underbrace{\sum_{\tau=r/h}^{s/h} ||\hat{G}_{\tau \cdot h} - \bar{G}_{\tau \cdot h}||_F^2}_{\text{(Algorithm 6 estimation error)}} + 2 \underbrace{\sum_{\tau_1=(r-1)/h}^{s/h-1} \sum_{\tau_2=1}^{h} ||\bar{G}_{(\tau_1+1)\cdot h} - \bar{G}_{\tau_1 \cdot h + \tau_2}||_F^2}_{(\bar{G} \text{ movement})}$$

For the first term, we will simply apply the above corollary after taking expectation. We will therefore focus on bounding the $\bar{G}$ movement term.

$$\forall \tau_2: \quad ||\bar{G}_{(\tau_1+1)\cdot h} - \bar{G}_{\tau_1 \cdot h + \tau_2}||_F^2 \leq \left( \sum_{i=1}^{h-1} ||\bar{G}_{\tau_1 \cdot h + i + 1} - \bar{G}_{\tau_1 \cdot h + i}||_F \right)^2 \qquad (\triangle\text{-ineq.})$$

$$\leq h \sum_{i=1}^{h-1} ||\bar{G}_{\tau_1 \cdot h + i + 1} - \bar{G}_{\tau_1 \cdot h + i}||_F^2 \qquad (\text{C.S.})$$

This implies that

$$\sum_{\tau_2=1}^{h} ||\bar{G}_{(\tau_1+1)\cdot h} - \bar{G}_{\tau_1 \cdot h + \tau_2}||_F^2 \leq h^2 \sum_{i=1}^{h-1} ||\bar{G}_{\tau_1 \cdot h + i + 1} - \bar{G}_{\tau_1 \cdot h + i}||_F^2$$

So finally we have that

$$\bar{G}\text{-movement} \leq h^2 \sum_{t=r+1}^{s} ||\bar{G}_t - \bar{G}_t||_F^2$$

$$\leq 4h^2 \text{Var}_I^{\text{tot}}(\mathbf{G}) \qquad\qquad \textit{Observation D.2} (5)$$

Finally, taking expectation, plugging in Corollary D.1 and noting that for $J = [r/h, s/h]$, $\text{Var}_{J \cdot h}^{\text{tot}}(\bar{G}_{1:T}) \leq \text{Var}_I^{\text{tot}}(\mathbf{G})$ (Observation D.2 (2)) and $\text{Regret}_J(\mathcal{A}(p, \bar{R}_G, \tilde{R}_G; 0)) \leq \text{Regret}_I(\mathcal{A}(p, \bar{R}_G, \tilde{R}_G; 0))$, we get:

$$\mathbb{E}\left[\sum_{t=r}^{s} ||\hat{G}_t - \bar{G}_t||_F^2\right] \leq 10h^2 \mathbb{E}[\text{Var}_I^{\text{tot}}(\mathbf{G})] + 2h\text{Regret}_I(\mathcal{A}(p, \bar{R}_G, \tilde{R}_G; 0))$$

$\square$

---

[3]This is w.l.o.g. and only assumed for simplicity of presentation.

**Lemma D.4.** *On any interval $I = [r, s] \subseteq [T]$ with $r - 1, s$ divisible by $h$, we can bound the squared $\ell_1, op$ estimation error of the truncated operators as:*

$$\mathbb{E}\left[\sum_{t=r}^{s} ||\hat{G}_t - \bar{G}_t||^2_{\ell_1,op}\right] \leq 20h^3\mathbb{E}[\text{Var}_I^{\text{tot}}(\mathbf{G})] + 4h^2\text{Regret}_I(\mathcal{A}(p, \bar{R}_G, \tilde{R}_G; 0))$$

*Proof.* We have that

$$\sum_{t=r}^{s} ||\hat{G}_t - \bar{G}_t||^2_{\ell_1,op} \leq \sum_{t=r}^{s}\left(\sum_{i=0}^{h-1} ||\hat{G}_t^{[i]} - \bar{G}_t^{[i]}||_{op}\right)^2 \qquad\qquad \triangle\text{-ineq.}$$

$$\leq 2\sum_{t=r}^{s}\left(\sum_{i=0}^{h-1} ||\hat{G}_t^{[i]} - \bar{G}_t^{[i]}||_{op}\right)^2$$

$$\leq 2h\sum_{t=r}^{s}\sum_{i=0}^{h-1} ||\hat{G}_t^{[i]} - \bar{G}_t^{[i]}||^2_F \qquad\qquad \textit{Observation D.2 (1) \& C.S.}$$

$$= 2h\sum_{t=r}^{s} ||\hat{G}_t - \bar{G}_t||^2_F \qquad\qquad\qquad \textit{Observation D.2 (3)}$$

Taking expectation and plugging in the bound in Lemma D.3 yields the promised result. $\qquad\square$

Finally we can use Lemma D.4 and Cauchy-Schwarz to get a result in terms of the linear (rather than squared) $\ell_1, op$ error accumulated over an interval:

**Proposition D.5.** *On any interval $I = [r, s] \subseteq [T]$ with $r - 1, s$ divisible by $h$, we can bound the squared $\ell_1, op$ estimation error of the truncated operators as:*

$$\mathbb{E}\left[\sum_{t=r}^{s} ||\hat{G}_t - \bar{G}_t||_{\ell_1,op}\right] \leq 5h^{3/2}|I|^{1/2}\sqrt{\text{Var}_I^{\text{tot}}(\mathbf{G})} + 2h|I|^{1/2}\text{Regret}_I^{1/2}(\mathcal{A}(p, \bar{R}_G, \tilde{R}_G; 0))$$

*Proof.* By Cauchy-Schwarz and Jensen (since $\sqrt{\cdot}$ is concave) we have:

$$\mathbb{E}\left[\sum_{t=r}^{s} ||\hat{G}_t - \bar{G}_t||_{\ell_1,op}\right] \leq |I|^{1/2}\mathbb{E}\left[\left(\sum_{t=r}^{s} ||\hat{G}_t - \bar{G}_t||^2_{\ell_1,op}\right)^{1/2}\right]$$

$$\leq |I|^{1/2}\left(\mathbb{E}\left[\sum_{t=r}^{s} ||\hat{G}_t - \bar{G}_t||^2_{\ell_1,op}\right]\right)^{1/2}$$

$$\leq |I|^{1/2}\sqrt{20h^3\mathbb{E}[\text{Var}_I^{\text{tot}}(\mathbf{G})] + 4h^2\text{Regret}_I(\mathcal{A}(p, \bar{R}_G, \tilde{R}_G; 0))}$$

$$\leq 5h^{3/2}|I|^{1/2}\sqrt{\mathbb{E}[\text{Var}_I^{\text{tot}}(\mathbf{G})]} + 2h|I|^{1/2}\text{Regret}_I^{1/2}(\mathcal{A}(p, \bar{R}_G, \tilde{R}_G; 0))$$

$\qquad\square$

## D.2 Error Sensitivity

We now analyze concretely how the $G_t$ estimation errors induce additional regret over the case of *known* systems. We can decompose the expected regret over an interval $I = [r, s]$ as:

$$\text{Regret}_I = \sum_{t=r}^{s} c_t(x_t, u_t) - \min_{\pi \in \Pi_{\text{drc}}} \sum_{t=r}^{s} c_t(x_t^\pi, u_t^\pi)$$

$$= \underbrace{\sum_{t=r}^{s} c_t(x_t, u_t) - c_t(\hat{x}_t(M_{t-h:t-1}), \hat{u}_t(M_t))}_{\text{(realized iterate error)}}$$

$$+ \underbrace{\sum_{t=r}^{s} c_t(\hat{x}_t(M_{t-h:t-1}), \hat{u}_t(M_t)) - \inf_{M \in \mathcal{M}} \sum_{t=r}^{s} c_t(\hat{x}_t(M), \hat{u}_t(M))}_{(\text{regret} := \widehat{\text{Regret}}_I)}$$

$$+ \underbrace{\inf_{M \in \mathcal{M}} \sum_{t=r}^{s} c_t(\hat{x}_t(M), \hat{u}_t(M)) - \inf_{M \in \mathcal{M}} \sum_{t=r}^{s} c_t(x_t(M), u_t(M))}_{\text{(comparator error)}}$$

First let us bound the realized iterate error which bounds the difference between what actually happened and what would have happened in the fictive $(\hat{G}_t, \hat{x}_t^{\text{nat}})$ system (without exploration).

**Lemma D.6.** *We can bound the difference between the true $x_t^{\text{nat}}$ and the extracted $\hat{x}_t^{\text{nat}}$ as:*

$$\|x_t^{\text{nat}} - \hat{x}_t^{\text{nat}}\| \le R_{\text{nat}} R_{\mathcal{M}} \left( \|\bar{G}_{t-1} - \hat{G}_{t-1}\|_{\ell_1, op} + \psi(h) \right)$$

*Proof.* Since $x^{\text{nat}} \in \mathbb{B}_{R_{\text{nat}}}$ and because (as argued earlier) $\|u_t\| \le R_{\text{nat}} R_{\mathcal{M}}$, we have:

$$\|x_t^{\text{nat}} - \hat{x}_t^{\text{nat}}\| = \left\| x_t^{\text{nat}} - \text{Proj}_{\mathbb{B}_{R_{\text{nat}}}} \left[ x_t - \sum_{i=0}^{h-1} \hat{G}_{t-1}^{[i]} u_{t-1-i} \right] \right\|$$

$$\le \left\| x_t^{\text{nat}} - x_t + \sum_{i=0}^{h-1} \hat{G}_{t-1}^{[i]} u_{t-1-i} \right\| \qquad \text{(Pythagoras)}$$

$$= \left\| \sum_{i=0}^{h-1} (G_{t-1}^{[i]} - \hat{G}_{t-1}^{[i]}) u_{t-1-i} + \sum_{i=h}^{t-2} G_{t-1}^{[i]} u_{t-1-i} \right\|$$

$$\le R_{\text{nat}} R_{\mathcal{M}} \left( \|\bar{G}_{t-1} - \hat{G}_{t-1}\|_{\ell_1, op} + \psi(h) \right)$$

$\square$

**Lemma D.7** (Realized Iterate Error). *For $I = [r, s]$ with $r - 1, s$ divisible by $h$, we can bound the realized iterate error as:*

$$\sum_{t=r}^{s} c_t(x_t, u_t) - c_t(\hat{x}_t(M_{t-h:t-1}), \hat{u}_t(M_t)) \le 2h \sum_{\tau_1 = (r-1)/h}^{s/h} b_{\tau_1 - 1} + 4L\sqrt{d_u} \frac{R_{\text{sys}}^2}{R_G} \left( \sum_{t=r}^{s} \|\bar{G}_{t-1} - \hat{G}_{t-1}\|_{\ell_1, op} + \psi(h)|I| \right)$$

*where we denote $R_{\text{sys}} \doteq R_G R_{\text{nat}} R_{\mathcal{M}}$.*

*Proof.* Consider the cost difference on a $h$-block indexed by $t = \overline{\tau_1 \cdot h + 1, \tau_1 \cdot h + h}$. Consider the following three cases:

1. $b_{\tau_1} = 1$: in this case we are exploring and cannot give a better guarantee than

$$c_t(x_t, u_t) - c_t(\hat{x}_t(M_{t-h:t-1)}, \hat{u}_t(M_t)) \le 1$$

2. $b_{\tau_1} = 0, b_{\tau_1 - 1} = 1$: while we are not exploring during the current round, and hence $u_t = \hat{u}_t(M_t)$ for $t = \overline{\tau_1 \cdot h + 1, \tau_1 \cdot h + h}$, we have explored in the previous round and

therefore $u_t$ and $\hat{u}_t(M_t)$ may be arbitrarily far for $t \leq \tau_1 \cdot h$. This can induce $x_t$ and $\hat{x}_t(M_{t-h:t-1})$ to be quite far, especially the closer we get to $\tau_1 \cdot h + 1$. As such, in this event, we will also simply bound

$$c_t(x_t, u_t) - c_t(\hat{x}_t(M_{t-h:t-1}), \hat{u}_t(M_t)) \leq 1$$

3. $b_{\tau_1} = 0, b_{\tau_1-1} = 0$: finally, in this case we have that $u_t = \hat{u}_t(M_t)$ for $t = \overline{\tau_1 \cdot h - h + 1, \tau_1 \cdot h + h}$. We can expand:

$$x_t = x_t^{\text{nat}} + \sum_{i=0}^{h-1} G_{t-1}^{[i]} u_{t-1-i} + \sum_{i=h}^{t-2} G_{t-1}^{[i]} u_{t-1-i}$$

and

$$\hat{x}_t(M_{t-h:t-1}) = \hat{x}_t^{\text{nat}} + \sum_{i=0}^{h-1} \hat{G}_{t-1}^{[i]} \hat{u}_{t-1-i}(M_{t-1-i})$$

By the observation above, for $t = \overline{\tau_1 \cdot h + 1, \tau_1 \cdot h + h}$, we have

$$\|x_t - \hat{x}_t(M_{t-h:t-1})\| \leq \|x_t^{\text{nat}} - \hat{x}_t^{\text{nat}}\| + R_{\text{nat}} R_{\mathcal{M}} \left( \|\bar{G}_{t-1} - \hat{G}_{t-1}\|_{\ell_1, op} + \psi(h) \right)$$

$$\leq 2 R_{\text{nat}} R_{\mathcal{M}} \left( \|\bar{G}_{t-1} - \hat{G}_{t-1}\|_{\ell_1, op} + \psi(h) \right) \qquad \textit{Lemma D.6}$$

Hence, by the sub-quadratic Lipschitzness of the cost and Eq. (D.1) we have

$$c_t(x_t, u_t) - c_t(\hat{x}_t(M_{t-h:t-1}), \hat{u}_t(M_t)) \leq 4L\sqrt{d_u} R_G R_{\text{nat}}^2 R_{\mathcal{M}}^2 \left( \|\bar{G}_{t-1} - \hat{G}_{t-1}\|_{\ell_1, op} + \psi(h) \right)$$

So for any $\tau_1 = \overline{0, T/h}$, we have

$$\sum_{t=\tau_1 \cdot h+1}^{\tau_1 \cdot h+h} c_t(x_t, u_t) - c_t(\hat{x}_t(M_{t-h:t-1}), \hat{u}_t(M_t)) \leq \mathbf{1}_{b_{\tau_1}=1} \cdot h + \mathbf{1}_{b_{\tau_1}=0, b_{\tau_1-1}=1} \cdot h$$

$$+ \mathbf{1}_{b_{\tau_1}=0, b_{\tau_1-1}=0} 4L\sqrt{d_u} R_G R_{\text{nat}}^2 R_{\mathcal{M}}^2 \sum_{t=\tau_1 \cdot h+1}^{\tau_1 \cdot h+h} \|\bar{G}_{t-1} - \hat{G}_{t-1}\|_{\ell_1, op}$$

$$+ \mathbf{1}_{b_{\tau_1}=0, b_{\tau_1-1}=0} 4L\sqrt{d_u} R_G R_{\text{nat}}^2 R_{\mathcal{M}}^2 h \cdot \psi(h)$$

$$\leq (b_{\tau_1} + b_{\tau_1-1}) \cdot h$$

$$+ 4L\sqrt{d_u} R_G R_{\text{nat}}^2 R_{\mathcal{M}}^2 \left( \sum_{t=\tau_1 \cdot h}^{(\tau_1+1) \cdot h-1} \|\bar{G}_t - \hat{G}_t\|_{\ell_1, op} + h\psi(h) \right)$$

summing over $\tau_1$ yields the desired result.

$\square$

**Lemma D.8** (Comparator Error). *We can bound the comparator error as:*

$$\textit{(comparator error)} \leq 8L\sqrt{d_u} R_{\text{nat}}^2 R_G^2 R_{\mathcal{M}}^3 h \left( m \sum_{t=r-h-m}^{s} \|\bar{G}_t - \hat{G}_t\|_{\ell_1, op} + \psi(h) \right)$$

*Proof.* Let $M^* = \arg\min_{M \in \mathcal{M}} \sum_{t=r}^{s} \sum_{t=r}^{s} c_t(x_t(M), u_t(M))$. We have that:

$$\textit{(comparator error)} \leq \sum_{t=r}^{s} c_t(\hat{x}_t(M^*), \hat{u}_t(M^*)) - c_t(x_t(M^*), u_t(M^*))$$

$$\leq 2L\sqrt{d_u} R_{\text{nat}} R_G R_{\mathcal{M}} \sum_{t=r}^{s} \left( \|\hat{x}_t(M^*) - x_t(M^*)\| + \|\hat{u}_t(M^*)) - u_t(M^*)\| \right)$$

We have that

$$\|\hat{u}_t(M^\star) - u_t(M^\star)\| = \|\sum_{i=0}^{m-1} M^{\star,[i]}(\hat{x}_{t-i}^{\mathrm{nat}} - x_{t-i}^{\mathrm{nat}})\|$$

$$\leq R_{\mathrm{nat}} R_{\mathcal{M}}^2 \left( \sum_{\tau=t-m}^{t-1} \|\bar{G}_\tau - \hat{G}_\tau\|_{\ell_1,op} + \psi(h) \right)$$

With a bit more computation, we can also bound the difference in the states. First, let us expand the expression of the states:

$$x_t(M^\star) = x_t^{\mathrm{nat}} + \sum_{i=0}^{h-1} G_{t-1}^{[i]} u_{t-1-i}(M^\star) + \sum_{i=h}^{t-2} G_{t-1}^{[i]} u_{t-1-i}(M^\star)$$

$$\hat{x}_t(M^\star) = \hat{x}_t^{\mathrm{nat}} + \sum_{i=0}^{h-1} \hat{G}_{t-1}^{[i]} \hat{u}_{t-1-i}(M^\star)$$

The only new thing we need to bound is:

$$\sum_{i=0}^{h-1} G_{t-1}^{[i]} u_{t-1-i}(M^\star) - \sum_{i=0}^{h-1} \hat{G}_{t-1}^{[i]} \hat{u}_{t-1-i}(M^\star) = \sum_{i=0}^{h-1} G_{t-1}^{[i]} u_{t-1-i}(M^\star) - \sum_{i=0}^{h-1} \hat{G}_{t-1}^{[i]} u_{t-1-i}(M^\star)$$

$$+ \sum_{i=0}^{h-1} \hat{G}_{t-1}^{[i]} u_{t-1-i}(M^\star) - \sum_{i=0}^{h-1} \hat{G}_{t-1}^{[i]} \hat{u}_{t-1-i}(M^\star)$$

$$\leq R_{\mathrm{nat}} R_{\mathcal{M}} \|\bar{G}_{t-1} - \hat{G}_{t-1}\|_{\ell_1,op} + R_G \sum_{i=1}^{h} \|u_{t-i}(M^\star) - \hat{u}_{t-i}(M^\star)\|$$

$$\leq 2 R_G R_{\mathrm{nat}} R_{\mathcal{M}}^2 h \left( m \sum_{\tau=t-h-m}^{t-1} \|\bar{G}_\tau - \hat{G}_\tau\|_{\ell_1,op} + \psi(h) \right)$$

Plugging this into our expressions for $x_t(M^\star)$, and using previous bounds we have

$$\|\hat{x}_t(M^\star) - x_t(M^\star)\| \leq 3 R_G R_{\mathrm{nat}} R_{\mathcal{M}}^2 h \left( m \sum_{\tau=t-h-m}^{t-1} \|\bar{G}_\tau - \hat{G}_\tau\|_{\ell_1,op} + \psi(h) \right)$$

Hence we can finalize that:

$$(\text{comparator error}) \leq 8 L \sqrt{d_u} R_{\mathrm{nat}}^2 R_G^2 R_{\mathcal{M}}^3 h \left( m \sum_{t=r-h-m}^{s} \|\bar{G}_t - \hat{G}_t\|_{\ell_1,op} + \psi(h) \right)$$

$\square$

### D.3 Proof of Theorem D.1

*Proof of Theorem D.1.* We have that

$$\text{Regret}_I \leq \underbrace{18L\sqrt{d_{\min}}R_G^2 R_\mathcal{M}^2 R_{\text{nat}}^2 m(h+1)^{5/4}\sqrt{T}}_{\text{(Known System Regret)}} + \underbrace{2h\sum_{\tau=(r-1)/h}^{s/h} b_{\tau-1}}_{\text{(Exploration Penalty)}}$$

$$+ \underbrace{12LR_{\text{nat}}^2 R_G^2 R_\mathcal{M}^3 hm\left(\sum_{t=r}^s \|\bar{G}_t - \hat{G}_t\|_{\ell_1,op} + 2R_G(h+m)\right)}_{\text{(System Misspecification Induced Error)}}$$

$$+ \underbrace{18L\sqrt{d_u}R_G^2 R_\mathcal{M}^3 R_{\text{nat}}^2 \psi(h)h(|I|+h+m)}_{\text{(Truncation Error)}}$$

Taking expectation and plugging in [Proposition D.5] we get:

$$\mathbb{E}[\text{Regret}_I] \leq \widetilde{\mathcal{O}}\bigg( L\sqrt{d_u}R_{\text{sys}}^2 m\sqrt{T} + p|I|$$

$$+ LR_{\text{sys}}^2 R_\mathcal{M}m\left(|I|^{1/2}\sqrt{\mathbb{E}[\text{Var}_I^{\text{tot}}(\mathbf{G})]} + d_u R_{\text{sys}}|I|^{1/2}p^{-1/2} + R_G m\right)$$

$$+ L\sqrt{d_u}R_{\text{sys}}^2 R_\mathcal{M}R_G m\bigg) \tag{D.2}$$

$$= \widetilde{\mathcal{O}}^\star\left(Lm\left(|I|\sqrt{\mathbb{E}[\text{Var}_I(\mathbf{G})]} + d_u T^{2/3}\right)\right)$$

where $R_{\text{sys}} = R_G R_\mathcal{M} R_{\text{nat}}$, using $\text{Var}_I(\mathbf{G}) = |I|\text{Var}_I^{\text{tot}}(\mathbf{G})$, and that for the chosen $h$ we have $\psi(h) \leq R_G T^{-1}$. $\qquad\square$

# E  Sublinear Regret for State Feedback

We demonstrate that it is (information-theoretically) possible to achieve sublinear (though large) regret against a benchmark of stabilizing static feedback control policies.

We suppose there is a subset $\mathcal{K} \subset \mathbb{R}^{d_u \times d_x}$ of feedback policies $K$, and our goal is to obtain regret compared to the best $K \in \mathcal{K}$:

$$\text{Reg}_T(\mathcal{K}) := \sum_{t=1}^T c_t(x_t, u_t) - \inf_{K \in \mathcal{K}} \sum_{t=1}^T c_t(x_t^K, u_t^K),$$

where $(x_t^K, u_t^K)$ are the iterates arising under the control law $u_t = Kx_t$.

For this setting, we propose an algorithm the classic Exp3exponential weights algorithm (see, e.g. Chapter 3 of [8]) on an $\varepsilon$-cover $\mathcal{K}_\varepsilon$ of $\mathcal{K}$ in the operator norm. We maintain a constant controller $K$ on intervals of length $H$, and feed the losses on those intervals to the Exp3 algorithm. Pseudocode is given in [Algorithm 8].

We state our regret bound under the (quite restrictive) assumption that all policies $K \in \mathcal{K}$ are sequentially stabilizing. Formally, given a sequence of controllers $K \in \mathcal{K}$, we define

$$\Phi_{s:t}(K) := \prod_{i=s}^t (A_i + K_i B_i) = (A_t + KB_t)\cdot(A_{t-1} + KB_{t-1})\cdots\cdots(A_s + KB_s).$$

We assume that $\Phi_{s:t}(K)$ exhibits geometric decay uniformly over all times for any fixed $K$:

**Assumption 4.** There exists $c_\star \geq 1$ and $\rho_\star \in (1/2, 1)$ such that for any indices $s \leq t$ and any fixed $K \in \mathcal{K}$, $\|\Phi_{s:t}(K)\|_{\text{op}} \leq c_\star \rho_\star^{t-s}$. We define the constant

$$R_\mathcal{K} := 1 + \max\{\|K\| : K \in \mathcal{K}\}$$

**Theorem E.1.** *Suppose [Assumptions 3] and [4] holds, and for some $R_w \geq 1$ and $R_B \geq 0$, $\max_t \|w_t\| \leq R_w$, and $\max_t \|B_t\| \leq R_B$. In addition, suppose $T$ is large enough that*

---

**Algorithm 8** Exponentially Weighted Control
---
1: **Input:** window length $H$, step size $\eta > 0$, finite $\varepsilon$-cover $\mathcal{K}_\varepsilon \subset \mathcal{K}$, initial estimate $K_1 \in \mathcal{K}_\varepsilon$
2: **Initialize** $\mathcal{L}_1(K) = 0$ and $p_1(K) = 1/|\mathcal{K}_\varepsilon|$, $\forall K \in \mathcal{K}_\varepsilon$,
3: **for** $t = 1, \dots, T$ **do**
4:     **Play** select $u_t = K_t x_t$.
5:     **Recieve** $\hat{c}_t = c_t(x_t, u_t)$
6:     **if** $t \mod H = 0$ **then**
7:         **Set** $n = t/H$, $\ell_n = \sum_{i=t-H+1}^{t} \hat{c}_i$
8:         **Set** $\mathcal{L}_{n+1}(K_t) = \frac{1}{p_n(K_t)}\ell_n + \mathcal{L}_n(K_t)$
9:         **Set** $\mathcal{L}_{n+1}(K) = \mathcal{L}_n(K)$ for all $K \in \mathcal{K}_\varepsilon/\{K_t\}$
10:       **Set** $p_{n+1}(K) = \frac{\exp(\eta\mathcal{L}_{n+1}(K))}{\sum_{K' \in \mathcal{K}_\varepsilon} \exp(\eta\mathcal{L}_{n+1}(K'))}$
11:       **Sample** $K_{t+1} \sim p_{n+1}(\cdot)$.
12:     **else**
13:       Set $K_{t+1} = K_t$

---

$\mathcal{C}_\star \rho_\star^{T/4} \leq 1/2$. Then, Algorithm 8 with horizon $H = \lceil T^{1/4} \rceil$, appropriate an step size $\eta$ and minimal $\varepsilon$-covering $\mathcal{K}_\varepsilon$ of $\mathcal{K}$ enjoys the following regret bound:

$$\mathbb{E}[\text{Reg}_T(\mathcal{K})] \leq L\mathcal{C}_1(5R_\mathcal{K})^{d_x d_u/2} \cdot T^{1-\frac{1}{2(d_x d_u+3)}},$$

where $\mathcal{C}_1 = \mathcal{O}\left(\frac{\mathcal{C}_\star^5}{(1-\rho_\star)^3} R_\mathcal{K}^3 R_w^2 (1 + R_B)\sqrt{d_u d_x}\right)$.

The theorem is established by a reduction to online multi-arm bandits in Appendix E.1 below.

**Remark E.1** (Extensions of Theorem E.1)**.** The following analysis extends to policies of the form $u_t = (K_t^{\text{stb}} + K)x_t + v$, where $(K_t^{\text{stb}})$ is a fixed sequence of control policies determined a priori, $K \in \mathcal{K} \subset \mathbb{R}^{d_u \times d_x}$ is a feedback parameter, and $v \in \mathcal{V} \subset \mathbb{R}^{d_u}$ is a bounded affine term. Letting $(x_t^{K,c}, u_t^{K,c})$ denote the iterates produced by such a policy, our notion of regret is

$$\text{Reg}_T(\mathcal{K} \times \mathcal{V}) := \sum_{t=1}^{T} c_t(x_t, u_t) - \inf_{(K,c) \in \mathcal{K} \times \mathcal{V}} \sum_{t=1}^{T} c_t(x_t^{K,c}, u_t^{K,c}),$$

The only assumptions we require in general is that $\mathcal{V}$ is bounded, and that $\mathcal{K}$, combined with $(K_i^{\text{stb}})$, are sequentially stabilizing in the sense that, for any $s \leq t$, the fixed $(K_i^{\text{stb}})$ sequence, and any $K_{s:t}^{\text{stb}} \in \mathcal{K}^{t-s+1}$, it holds that the products

$$\Phi_{s:t}^{\text{stb}}(K_{s:t}) := \prod_{i=s}^{t}(A_i + (K_i^{\text{stb}} + K_i)B_i)$$

exhibit geometric decay. $\qquad\qquad\qquad\qquad\qquad\qquad\qquad\qquad\qquad\qquad\qquad \square$

### E.1   Proof of Theorem E.1

In what follows, assume that $H = T^{1/4}$ evenly divides $T$. For every index $n \in \mathbb{N}$, define $t_n = 1 + (n-1)H$. To avoid confusion, we $x_t^{\text{alg}}, u_t^{\text{alg}}$ denote the iterates produced by the algorithm. We define the sequence which begins at state $x_{t_n}^{\text{alg}}$ at time $t_n$, and rolls forward under controller $K$ for future times:

$$\bar{x}_{n;t_n}(K) = x_{t_n}^{\text{alg}}, \quad \bar{x}_{n;t+1}(K) = (A_t + B_t K)\bar{x}_{n;t}(K) + w_t, \quad t \geq t_n$$
$$\bar{u}_{n;t_n}(K) = K\bar{x}_{n;t_n}(K).$$

Observe that, since we select a new controller $K_{t_n}$ just before each time $t_n$, we have

$$(\bar{x}_{n;t_n}(K_{t_n}), \bar{u}_{n;t_n}(K_{t_n})) = (x_t^{\text{alg}}, u_t^{\text{alg}}), \quad \forall t \in [t_n, t_{n+1} - 1].$$

Therefore, defining the losses,

$$\ell_n(K) = \sum_{t=t_n}^{t_{n+1}-1} c_t(\bar{x}_{n;t_n}(K), \bar{u}_{n;t_n}(K)),$$

we have

$$\ell_n(K_{t_n}) = \sum_{t=t_n}^{t_{n+1}-1} c_t(x_t^{\mathsf{alg}}, u_t^{\mathsf{alg}}).$$

Therfore, we may decompose the regret as

$$\mathbb{E}[\mathrm{Reg}_T(\mathcal{K})] = \underbrace{\mathbb{E}\left[\sum_{n=1}^{T/H} \ell_n(K_{t_n})\right] - \inf_{K \in \mathcal{K}_\varepsilon} \mathbb{E}\left[\sum_{n=1}^{T/H} \ell_n(K)\right]}_{R_1}$$

$$+ \underbrace{\inf_{K \in \mathcal{K}_\varepsilon} \mathbb{E}\left[\sum_{n=1}^{T/H} \ell_n(K)\right] - \inf_{K \in \mathcal{K}_\varepsilon} \sum_{t=1}^{T} c_t(x_t^K, u_t^K)}_{R_2}$$

$$+ \underbrace{\inf_{K \in \mathcal{K}_\varepsilon} \sum_{t=1}^{T} c_t(x_t^K, u_t^K)] - \inf_{K \in \mathcal{K}} \sum_{t=1}^{T} c_t(x_t^K, u_t^K)}_{R_3}.$$

Here, $R_1$ is the *simple regret* on the $\ell_n$ sequence, $R_2$ is the extend to which the $\ell_n$ sequence approximates regret against controller $K \in \mathcal{K}_\varepsilon$ in the covering, and finally $R_3$ bounds the regret of the covering against the full set $\mathcal{K}$. Here, expectations are over the randomness in the algorithm, and due to the obliviousness of the adversary, we may assume that $(c_t, x_t^K, u_t^K)$ are deterministic and chosen in advance. We bound each of the three terms in sequence. Before proceeding, we use the following estimates:

**Lemma E.1** (Key Term Bounds). *Suppose that $H = T^{1/4}$ is sufficiently large that $\mathcal{C}_\star \rho_\star^H \le 1/2$. Moreover, let $R_\star = \frac{\mathcal{C}_\star}{1-\rho_\star}$. Then,*

(a) *For all $K \in \mathcal{K}$ and $t \in [T]$, $\|x_t^K\| \le R_\star R_w$*

(b) *For any $t \ge t_n$ and $K \in \mathcal{K}$, $\|\bar{x}_{n;t}(K)\| \le 2\mathcal{C}_\star R_\star R_w$*

(c) *$\|(x_t^K, u_t^K)\| \le R_\mathcal{K} R_\star R_w$ and $\|(\bar{x}_{n;t}(K), \bar{u}_{n;t}(K))\| \le 2R_\mathcal{K} \mathcal{C}_\star R_\star R_w$.*

*Proof. Part a:* Unfolding the dynamics, and bounding $\|w_t\| \le R_w$ and $\Phi$ via Assumption 4,

$$\|x_t^K\| = \left\| \sum_{s=1}^{t-1} \Phi_{s+1:t}(K) w_t \right\| \le R_w \mathcal{C}_\star \sum_{s \ge 0} \rho_\star^s = \frac{\mathcal{C}_\star R_w}{\rho_\star} := R_\star R_w.$$

Next, we bound $\|x_{t_n}^{\mathsf{alg}}\|$ for some $n$,

$$\|x_{t_n}^{\mathsf{alg}}\| = \left\| \sum_{i=1}^{H} \Phi_{t_{n-1}+i:t_n-1}(K_{t_{n-1}}) w_{t_n-i} + \Phi_{t_{n-1}:t_n}(K_{t_{n-1}}) x_{t_{n-1}}^{\mathsf{alg}} \right\|$$

$$\le R_\star R_w + \mathcal{C}_\star \rho_\star^H \|x_{t_n-1}^{\mathsf{alg}}\|.$$

If $H$ is sufficiently large that $\mathcal{C}_\star \rho_\star^H \le 1/2$, then the above is just

$$\|x_{t_n}^{\mathsf{alg}}\| \le R_\star R_w + \frac{1}{2}\|x_{t_n-1}^{\mathsf{alg}}\|,$$

yielding the bound $\|x_{t_n}^{\mathsf{alg}}\| \le 2R_\star R_w$ for all $n$. *Part b:* Next, let us bound $\|\bar{x}_{n;t}(K)\|$ for some $t \ge t_n$. We have

$$
\begin{aligned}
\|\bar{x}_{n;t}(K)\| &= \left\| \sum_{i=t_n+1}^{t} \Phi_{i:t}(K)w_t t + \Phi_{t_n:t}(K)x_{t_n}^{\mathsf{alg}} \right\| \\
&\le R_w \Big( \sum_{0}^{t-t_n-1} \mathcal{C}_\star \rho_\star^i \Big) + \mathcal{C}_\star \rho_\star^{t_n} \|x_{t_n}^{\mathsf{alg}}\| \\
&\le R_w \sum_{0}^{t-t_n-1} \mathcal{C}_\star \rho_\star^i + 2\rho_\star^{t_n} \mathcal{C}_\star R_\star R_w.
\end{aligned}
$$

Using $R_\star = \frac{\mathcal{C}_\star}{\rho_\star}$, $\mathcal{C}_\star \ge 1$, and $\sum_{0}^{t-t_n-1} + \frac{\rho_\star^{t_n}}{1-\rho_\star} = \frac{1}{1-\rho_\star}$, the above simplifies to $2\frac{\mathcal{C}_\star^2 R_w}{1-\rho_\star} = 2\mathcal{C}_\star R_w R_\star$.

*Part c:* This follows from the fact that, for any $x \in \mathbb{R}^{d_x}$ and $K \in \mathcal{K}$, $\|(x, Kx)\| \le (1 + \|K\|)\|x\| \le R_{\mathcal{K}}\|x\|$. $\qquad\square$

**Bounding $R_1$** The term $R_1$ corresponds to the simple regret on the sequence of losses $\ell_n(K)$ over the discrete enumeration of controllers $K \in \mathcal{K}_\varepsilon$. Examining *Algorithm* 8, we simply run the Exp3 algorithm on these losses. By appealing to a standard regret bound for this algorithm with appropriate step size $\eta$, we ensure that

$$
R_1 \le 2B\sqrt{\frac{T}{H}|\mathcal{K}_\varepsilon| \log |\mathcal{K}_\varepsilon|},
$$

provided that, for all $n$ and $K \in \mathcal{K}_\varepsilon$, $\ell_n(K) \in [0, B]$. To find the appropriate bound $B$, we note that from the growth condition on the costs, Assumption 3, we have

$$
0 \le \ell_n(K) = \sum_{t=t_n}^{t_{n+1}-1} c_t(\bar{x}_{n;t_n}(K), \bar{u}_{n;t_n}(K)) \le H \max_{t \ge t_n} c_t(\bar{x}_{n;t_n}(K), \bar{u}_{n;t_n}(K))
$$

$$
\le LH \max\{1, \|(\bar{x}_{n;t_n}(K), \bar{u}_{n;t_n}(K))\|^2\} \le 8LH(R_{\mathcal{K}}\mathcal{C}_\star R_\star R_w)^2,
$$

where the last inequality uses Lemma E.1. Hence,

$$
R_1 \le 16L(R_{\mathcal{K}}\mathcal{C}_\star R_\star R_w)^2 \sqrt{TH|\mathcal{K}_\varepsilon| \log |\mathcal{K}_\varepsilon|}.
$$

**Bounding $R_2$:** To bound $R_2$, it suffices to find a probability-one upper bound on

$$
\sup_{K \in \mathcal{K}_\varepsilon} \left| \sum_{n=1}^{T/H} \ell_n(K) - \sum_{t=1}^{T} c_t(x_t^K, u_t^K) \right| = \sup_{K \in \mathcal{K}_\varepsilon} \left| \sum_{n=1}^{T/H} \sum_{t=t_n}^{t_{n+1}-1} c_t(\bar{x}_{n;t}(K), \bar{u}_{n;t}(K)) - c_t(x_t^K, u_t^K) \right|
$$

$$
\le \frac{T}{H} \sup_{K \in \mathcal{K}_\varepsilon} \max_n \sum_{t=t_n}^{t_{n+1}-1} \left| c_t(\bar{x}_{n;t}(K), \bar{u}_{n;t}(K)) - c_t(x_t^K, u_t^K) \right|.
$$

Using the Lipschitz conditions on $c_t$, the bounds from Lemma E.1, and the bound $1 + \|K\| \le R_{\mathcal{K}}$,

$$
\begin{aligned}
\sum_{t=t_n}^{t_{n+1}-1} \left| c_t(\bar{x}_{n;t}(K), \bar{u}_{n;t}(K)) - c_t(x_t^K, u_t^K) \right| &\le L(R_{\mathcal{K}}\mathcal{C}_\star R_\star R_w) \sum_{t=t_n}^{t_{n+1}-1} \|(\bar{x}_{n;t}(K), \bar{u}_{n;t}(K)) - (x_t^K, u_t^K)\| \\
&= L(R_{\mathcal{K}}\mathcal{C}_\star R_\star R_w) \sum_{t=t_n}^{t_{n+1}-1} \|(\bar{x}_{n;t}(K), K\bar{x}_{n;t}(K)) - (x_t^K, Kx_t^K)\| \\
&= L(R_{\mathcal{K}}\mathcal{C}_\star R_\star R_w)R_{\mathcal{K}} \sum_{t=t_n}^{t_{n+1}-1} \|\bar{x}_{n;t}(K) - x_t^K\|.
\end{aligned}
$$

Finally, we can compute that the difference $\bar{x}_{n;t}(K) - x_t^K = \Phi_{t_n}^t(x_{t_n}^{\mathsf{alg}} - x_{t_n}^K)$ depends only on the response to the state difference at time $t_n$. Hence, using Assumption 4 and Lemma E.1, the above is

at most

$$L(R_{\mathcal{K}}\mathcal{C}_\star R_\star R_w)R_{\mathcal{K}} \cdot \sum_{i \geq 0} \mathcal{C}_\star \rho_\star^i \|(x_{t_n}^{\mathsf{alg}} - x_{t_n}^K)\| \leq L(R_{\mathcal{K}}\mathcal{C}_\star R_\star R_w)R_{\mathcal{K}} \cdot \underbrace{\frac{\mathcal{C}_\star}{1 - \rho_\star}}_{= R_\star \cdot \mathcal{C}_\star R_\star R_w}$$

$$\leq L(R_{\mathcal{K}}\mathcal{C}_\star R_\star R_w)^2 R_\star.$$

Concluding, we find

$$R_2 \leq L(R_{\mathcal{K}}\mathcal{C}_\star R_\star R_w)^2 \cdot \frac{TR_\star}{H}.$$

**Bounding $R_3$.** We now turn to bounding $R_3$, which captures the approximation error of approximating $\mathcal{K}$ with $\mathcal{K}_\varepsilon$. We require the following technical lemma:

**Lemma E.2.** *Let $K, K' \in \mathcal{K}$ satisfy $\|K - K'\|_{\mathrm{op}} \leq \varepsilon$. Then, for all $t \geq 1$,*

$$\|x_t^K - x_t^{K'}\| \leq 4\varepsilon R_w R_\star^2 R_B.$$

*Hence,*

$$\|c_t(x_t^K, u_t^K) - c_t(x_t^{K'}, u_t^{K'})\| \leq 4\varepsilon L R_w^2 R_{\mathcal{K}}^2 R_\star^3 (1 + R_B).$$

Using the fact that $\mathcal{K}_\varepsilon$ is an $\varepsilon$-covering of $\mathcal{K}$ in the operator norm means that for any $K \in \mathcal{K}$, we can find a $K' \in \mathcal{K}_\varepsilon$ for which $\|K - K'\|_{\mathrm{op}} \leq \varepsilon$. Hence, from the above lemma

$$|\sum_{t=1}^T c_t(x_t^K, u_t^K) - c_t(x_t^{K'}, u_t^{K'})| \leq 4T\varepsilon L R_w^2 R_{\mathcal{K}}^2 R_\star^3 (1 + R_B).$$

Since $R_3 \leq \sup_{K \in \mathcal{K}} \inf_{K' \in \mathcal{K}_\varepsilon} |\sum_{t=1}^T c_t(x_t^K, u_t^K) - c_t(x_t^{K'}, u_t^{K'})|$, we conclude

$$R_3 \leq 4T\varepsilon L R_w^2 R_{\mathcal{K}}^2 R_\star^3 (1 + R_B).$$

**Concluding the proof** In sum, we found

$$\mathbb{E}[\mathrm{Reg}_T(\mathcal{K})] = R_1 + R_2 + R_3$$

$$\leq \mathcal{O}\left(L(R_{\mathcal{K}}\mathcal{C}_\star R_\star R_w)^2\right) \left(\sqrt{TH|\mathcal{K}_\varepsilon|\log|\mathcal{K}_\varepsilon|} + \frac{TR_\star}{H} + (1 + R_B)T\varepsilon\right),$$

$$\leq \mathcal{O}\left(L(R_{\mathcal{K}}\mathcal{C}_\star R_\star R_w)^2(1 + R_B)R_\star\right) \left(\sqrt{TH|\mathcal{K}_\varepsilon|\log|\mathcal{K}_\varepsilon|} + \frac{T}{H} + T\varepsilon\right),$$

where in the last line, we use $R_\star \geq 1$ and $1 + R_B \geq 1$. Setting $H = T^{1/4}$,

$$\mathbb{E}[\mathrm{Reg}_T(\mathcal{K})] \leq \mathcal{O}\left(L(R_{\mathcal{K}}\mathcal{C}_\star R_\star R_w)^2(1 + R_B)R_\star\right) T^{3/4} \cdot \left(\sqrt{|\mathcal{K}_\varepsilon|\log|\mathcal{K}_\varepsilon|} + T^{1/4}\varepsilon\right),$$

We bound the cardinality of $\mathcal{K}_\varepsilon$. It suffices to ensure $\mathcal{K}_\varepsilon$ is an $\varepsilon$-covering in the larger Frobenius norm, which is just the Euclidean norm on $R^{d_x d_u}$. Since $\mathcal{K}$ is a bounded subset of this space, with radius at most $R_{\mathcal{K}}$, we can find a covering such that $|\mathcal{K}_\varepsilon| \leq (\frac{5R_{\mathcal{K}}}{\varepsilon})^{d_x d_u}$ (see, e.g. Chapter 4.2 in [43]). This yields

$$|\mathcal{K}_\varepsilon|\log|\mathcal{K}_\varepsilon| \leq d_x d_u \left(\frac{5R_{\mathcal{K}}}{\varepsilon}\right)^{d_x d_u} \log\left(\frac{5R_{\mathcal{K}}}{\varepsilon}\right) \leq d_x d_u \left(\frac{5R_{\mathcal{K}}}{\varepsilon}\right)^{d_x d_u + 1} = 5R_{\mathcal{K}} \cdot (5R_{\mathcal{K}})^{d_x d_u} \varepsilon^{-d_x d_u + 1}$$

where we use $\log x \leq x$. Hence, we can bound

$$\mathbb{E}[\mathrm{Reg}_T(\mathcal{K})] \leq \mathcal{O}\left(L R_{\mathcal{K}} R_\star (R_{\mathcal{K}}\mathcal{C}_\star R_\star R_w)^2 (1 + R_B)\sqrt{d_x d_u}\right) (5R_{\mathcal{K}})^{d_x d_u/2} \cdot T^{3/4} \left(\left(\frac{1}{\varepsilon}\right)^{(1 + d_x d_u)/2} + \varepsilon T^{1/4}\right)$$

Setting $\varepsilon = T^{-\frac{1}{2(d_x d_u + 3)}}$ gives

$$\mathbb{E}[\mathrm{Reg}_T(\mathcal{K})] \leq \mathcal{O}\left(L R_\star (R_{\mathcal{K}}\mathcal{C}_\star R_\star R_w)^2 (1 + R_B)\sqrt{d_x d_u}\right) (5R_{\mathcal{K}})^{d_x d_u + 1} \cdot T^{1 - \frac{1}{2(d_x d_u + 3)}}$$

$$= \mathcal{O}\left(L \frac{\mathcal{C}_\star^5}{(1 - \rho_\star)^3} R_{\mathcal{K}}^3 R_w^2 (1 + R_B)\sqrt{d_x d_u}\right) (5R_{\mathcal{K}})^{d_x d_u/2} \cdot T^{1 - \frac{1}{2(d_x d_u + 3)}},$$

## E.2 Ommited Proofs

*Proof of Lemma E.1. Part a:* All such iterates can be realized by dynamics of the form $x_1 = 0$, $x_{t+1} = (A_t + B_t K_t)x_t + w_t$ and $u_t = K_t x_t$ for any appropriate sequence $(K_1, K_2, \dots)$ of elements of $\mathcal{K}$. For such dynamics, we find

$$x_t = \sum_{s=1}^{t-1} \left( \prod_{i=s+1}^{t} (A_i + B_i K_i) \right) w_s = \sum_{s=1}^{t-1} \Phi_{s+1;t}(K_{s+1:t}) w_t.$$

Using $\|w_s\| \le R_w$ and the assumption $\Phi_{s+1;t}(K_{s+1:t}) w_t \le c_\star \rho_\star^{t-s-1}$ from Assumption 4, we find

$$\|x_t\| \le R_w c_\star \sum_{s=1}^{t-1} \rho_\star^{t-s-1} \le \frac{c_\star R_w}{1 - \rho_\star} = R_\star R_w$$

$$\|u_t\| + \|x_t\| = \|x_t\| + \|K_t x_t\| \le \|x_t\|(1 + \|K_t\|) \le R_\star R_{\mathcal{K}} R_w$$

*Part b:* Since the closed-loop dynamics for $\bar{\mathbf{x}}_{k;t}^K$ and $x_t^K$ concincide for $t \ge t_k$ and are given by $x_{t+1} = (A_t + B_t K)x_t + w_t$, we can compute

$$\bar{\mathbf{x}}_{k;t}^K - x_t^K = \left( \prod_{i=t_k}^{t} (A_t + B_t K) \right) (\bar{\mathbf{x}}_{t_k;k}^K - x_t^K).$$

Bounding $\|\bar{\mathbf{x}}_{k;t}^K - x_t^K)\| \le 2R_w R_K$ from part (a) and $\|(A_t + B_t K)^{t-t_k}\| \le c_\star \rho_\star^{t-t_k}$ from Assumption 4 yields $\|\bar{\mathbf{x}}_{k;t}^K - x_t^K)\| \le 2R_x c_\star \rho_\star^{t-t_k}$. Summing over $t \ge t_k$ yields $\sum_{t \ge t_k} \|\bar{\mathbf{x}}_{k;t}^K - x_t^K\| \le 2R_w R_\star^2$. Finally, using $\bar{u}_{k;t}^K = K\bar{\mathbf{x}}_{k;t}^K$ and $u_t^K = K x_t^K$ gives

$$\sum_{t \ge t_k} \|\bar{\mathbf{x}}_{t;k}^K - x_t^K\| + \|\bar{u}_{t;k}^K - u_t^K\| \le (1 + \|K\|) \sum_{t \ge t_k} \|\bar{\mathbf{x}}_{t;k}^K - x_t^K\| \le 2R_w R_{\mathcal{K}} R_\star^2.$$

$\square$

*Proof of Lemma E.2.* Introducing the short hand $X_i = A_i + B_i K$ and $Y_i = A_i + B_i K'$, and expanding the dynamics, and introducing the short hand

$$\|x_t^K - x_t^{K'}\| = \left\| \sum_{s=1}^{t-1} \left( \prod_{i=s+1}^{t} \underbrace{(A_i + B_i K)}_{=X_i} - \prod_{i=s+1}^{t} \underbrace{(A_i + B_i K')}_{=Y_i} \right) w_s \right\|$$

$$\le R_w \sum_{s=1}^{t-1} \left\| \prod_{i=s+1}^{t} X_i - \prod_{i=s+1}^{t} Y_i \right\|_{\mathrm{op}}$$

Using an elementary matrix telescoping identiy,

$$\prod_{i=s+1}^{t} X_i - \prod_{i=s+1}^{t} Y_i = \sum_{j=s+1}^{t} \left( \prod_{i=j+1}^{t} X_j \right) (X_j - Y_j) \prod_{i=s+1}^{j-1} Y_j.$$

Thus, invoking stability assumption, Assumption 4, and setting $R_B \ge \max_t \|B_t\|_{\mathrm{op}}$,

$$\left\| \prod_{i=s+1}^{t} X_i - \prod_{i=s+1}^{t} Y_i \right\|_{\mathrm{op}} \le \sum_{j=s+1}^{t} \left\| \prod_{i=j+1}^{t} X_j \right\|_{\mathrm{op}} \left\| \prod_{i=s+1}^{j-1} Y_j \right\|_{\mathrm{op}} \|X_j - Y_j\|_{\mathrm{op}}$$

$$= \sum_{j=s+1}^{t} c_\star \rho_\star^{t-j+1} c_\star \rho_\star^{j-1-(s+1)} \|B_j(K - K')\|_{\mathrm{op}}$$

$$= \frac{c_\star^2}{\rho_\star^2} \sum_{j=s+1}^{t} \rho_\star^{t-(s+1)} \|B_j(K - K')\|_{\mathrm{op}}$$

$$\le \varepsilon R_B \frac{c_\star^2}{\rho_\star^2} (t - s + 1) \rho_\star^{t-(s+1)}$$

Thus, we find

$$
\|x_t^K - x_t^{K'}\| \leq \varepsilon R_w R_B \frac{c_\star^2}{\rho_\star^2} \sum_{s=1}^{t-1} (t - s + 1) \rho_\star^{t-(s+1)}
$$

$$
\leq \varepsilon R_w R_B \frac{c_\star^2}{\rho_\star^2 (1 - \rho_\star)^2} \leq 4\varepsilon R_w R_B \frac{c_\star^2}{(1 - \rho_\star)^2} = 4\varepsilon R_w R_\star^2 R_B,
$$

where in the last step, we use $\rho_\star \geq 1/2$. Thus, applying Assumption 3 and **??**,

$$
\|c_t(x_t^K, u_t^K) - c_t(x_t^{K'}, u_t^{K'})\| \leq L \max\{1, \|(x_t^K, u_t^K)\|, \|(x_t^{K'}, u_t^{K'})\|\} \cdot \|(x_t^K, u_t^K) - (x_t^{K'}, u_t^{K'})\|
$$

$$
\leq L R_w R_{\mathcal{K}} R_\star \cdot \|(x_t^K, u_t^K) - (x_t^{K'}, u_t^{K'})\|.
$$

Continuing, we bound

$$
\|(x_t^K, u_t^K) - (x_t^{K'}, u_t^{K'})\| = \|(x_t^K, K x_t^K) - (x_t^{K'}, K' x_t^{K'})\|
$$

$$
\leq \|(x_t^K, K x_t^K) - (x_t^{K'}, K x_t^{K'})\| + \|(K - K') x_t^{K'}\|
$$

$$
\leq R_{\mathcal{K}} \|x_t^{K'} - x_t^{K'}\| + \|(K - K') x_t^{K'}\|.
$$

Finally, using the bound $\|x_t^{K'} - x_t^{K'}\| \leq 4\varepsilon R_w R_\star^2 R_B$ derived above, and bounding $\|(K - K') x_t^{K'}\| \leq \varepsilon \|x_t^{K'}\| \leq \varepsilon R_w R_\star$ in view of Lemma E.1. Hence,

$$
\|c_t(x_t^K, u_t^K) - c_t(x_t^{K'}, u_t^{K'})\| \leq L R_w R_{\mathcal{K}} R_\star (4\varepsilon R_{\mathcal{K}} R_w R_\star^2 R_B + \varepsilon R_w R_\star)
$$

$$
\leq 4\varepsilon L R_w^2 R_{\mathcal{K}}^2 R_\star^3 (R_B + 1),
$$

where above we use that $R_{\mathcal{K}}, R_\star \geq 1$ by assumption. $\qquad\square$

# F   Lower Bounds and Separations

## F.1   Separation between policy classes

Let $\mathcal{Z} = (c_t, w_t, A_t, B_t)_{t \geq 1}$ denote sequences over costs, disturbances, and dynamics. We let $J_T(\pi; \mathcal{Z})$ denote the cost of policy $\pi$ on the sequence $\mathcal{Z}$. Our lower bounds hold even against sequences which enjoy the following *regularity* condition.

**Definition F.1.** We say that $\mathcal{Z}$ is *regular* if, for all $t$, $c_t(\cdot, \cdot)$ satisfies **??** with $L \leq 1$, and that for all $t$, $\|w_t\| \leq 1$, $\|B_t\|_{\mathrm{op}} \leq 1$ and $\|A_t\|_{\mathrm{op}} \leq 1/2$.

We define the policy classes

$$
\Pi_{\mathrm{drc}}^+(h) := \left\{ \pi : u_t^\pi = u_0 + \sum_{i=0}^{h-1} M^{[i]} x_{t-i}^{\mathrm{nat}}, \ \forall t \right\}
$$

$$
\Pi_{\mathrm{dac}}^+(h) := \left\{ \pi : u_t^\pi = u_0 + \sum_{i=0}^{h-1} M^{[i]} w_{t-i-1}, \ \forall t \right\}
$$

$$
\Pi_{\mathrm{feed}}^+(h) := \left\{ \pi : u_t^\pi = u_0 + \sum_{i=0}^{h-1} M^{[i]} x_{t-i}^\pi, \ \forall t \right\}
$$

$$
\Pi_{\mathrm{feed}}(h, R) := \left\{ \pi : u_t^\pi = u_0 + \sum_{i=0}^{h-1} M^{[i]} x_{t-i}^\pi, \ \forall t, \quad \sum_{i=0}^{h-1} \|M^{[i]}\|_{\mathrm{op}} \leq R. \right\}.
$$

That is, $\Pi_{\mathrm{drc}}^+(h)$ are all length $h$ DRC policies of unbounded norm and allowing affine offsets, $\Pi_{\mathrm{dac}}^+(h)$ are all length $h$ DAC policies of unbounded norm allowing affine offsets, and $\Pi_{\mathrm{feed}}^+(h)$ are all static feedback policies of unbounded norm allowing affine offsets, and $\Pi_{\mathrm{feed}}(h, R)$ are feedback policies of bounded norm and horizon.

The following theorem demonstrates that the DAC, DRC, and feedback parametrizations are fundamentally incommensurate.

**Theorem F.1.** *Let $\mathcal{C}_0 > 0$ denote a universal constant. There exists three regular sequences $\mathcal{Z}_1, \mathcal{Z}_2, \mathcal{Z}_3$ in $d_u = d_x = 1$ which separate DAC, DRC, and feedback controllers, in the following sense:*

*(a) Under $\mathcal{Z}_1$, the static feedback policy $\pi$ selecting $u_t^\pi = \frac{1}{4} x_t^\pi$ satisfies $J_T(\pi; \mathcal{Z}_1) = 0$, but*

$$\inf_{\pi \in \Pi_{\mathrm{drc}}^+(h) \cup \Pi_{\mathrm{dac}}^+(h)} J_T(\pi; \mathcal{Z}_1) \geq \mathcal{C}_0(T - h - 2)$$

*(b) Under $\mathcal{Z}_2$, the DAC policy $\pi$ selecting $u_t^\pi = w_t$ satisfies $J_T(\pi; \mathcal{Z}_2) = 0$, but*

$$\inf_{\pi \in \Pi_{\mathrm{drc}}^+(h) \cup \Pi_{\mathrm{feed}}^+(h)} J_T(\pi; \mathcal{Z}_2) \geq \mathcal{C}_0(T - h - 2)$$

*(c) Under $\mathcal{Z}_3$, the DRC policy $\pi$ selecting $u_t^\pi = x_t^{\mathrm{nat}}$ satifies $J_T(\pi; \mathcal{Z}_3) = 0$, but*

$$\inf_{\pi \in \Pi_{\mathrm{dac}}^+(h)} J_T(\pi; \mathcal{Z}_3) \geq \mathcal{C}_0(T - h - 3)$$

*Moreover, for any $h \in N$, $R > 0$ and $T \geq 10h$, we have*

$$\inf_{\pi \in \Pi_{\mathrm{feed}}(h,R)} J_T(\pi; \mathcal{Z}_3) \geq \mathcal{C}_0 \frac{T}{h \max\{R, 1\}}.$$

*Proof.* We establish the separations for each part with different constant factors. One can choose $\mathcal{C}_0$ to be the minimum of all constants which arise.

**Proof of part a.** We set $\mathcal{Z}_1$ to be the sequence with $A_t = 0$ for all $t$, $w_t = 1$ for all $t$, $c_t(x, u) = \frac{1}{8}(u - \frac{1}{4}x)^2$, and

$$B_t = \begin{cases} 1 & t \text{ odd} \\ -1 & t \text{ even.} \end{cases}$$

This sequence is clearly regular, and is clear that the policy $u_t^\pi = \frac{1}{4} x_t^\pi$ has $J_T(\pi; \mathcal{Z}_1) = 0$. On the other hand, let $\pi \in \Pi_{\mathrm{drc}}^+(h) \cup \Pi_{\mathrm{dac}}^+(h)$, $J_T(\pi; \mathcal{Z}_1)$. Since $A_t \equiv 0$, $x_t^{\mathrm{nat}} = w_t$ so $\Pi_{\mathrm{drc}}^+(h) = \Pi_{\mathrm{dac}}^+(h)$. Moreover, since $w_t = 1$ for all $t \geq 1$, any $\pi \in \Pi_{\mathrm{dac}}^+(h)$ has $u_t^\pi = \bar{u}$ for some fixed $\bar{u}$ for all $t > h$. Then, for all $t > h + 1$, $x_t = 1 + B_{t-1}\bar{u}$. Thus, $c_t(x_t^\pi, u_t^\pi) = \frac{1}{8}(\bar{u} - \frac{1 + B_{t-1}\bar{u}}{4})^2$. Using the definition of $B_t$,

$$c_t(x_t^\pi, u_t^\pi) + c_{t+1}(x_{t+1}^\pi, u_{t+1}^\pi) = \frac{1}{8}(\bar{u} - \frac{1 - \bar{u}}{4})^2 + \frac{1}{8}(\bar{u} - \frac{1 + \bar{u}}{4})^2 = \frac{1}{128}\left((3\bar{u} + 1)^2 + (5\bar{u} + 1)\right)^2 = \Omega(1).$$

The bound follows.

**Proof of part b.** Set $c_t(x, u) = (u - w_{t-1})^2$. Then the DAC policy $u_t^\pi = w_{t-1}$ has zero cost. Further, set $B_t \equiv 0$, thus, $x_t \equiv x_t^{\mathrm{nat}}$, so $\Pi_{\mathrm{feed}}^+(h)$ and $\Pi_{\mathrm{drc}}^+(h)$ are equivalent on this system. Finally, let $n = 2m + 1$, and set $w_1 = 1$, and for $t \geq 1$, set

$$(A_t, w_t) = \begin{cases} (\frac{1}{2}, \frac{1}{2}) & t \text{ is even} \\ (\frac{1}{4}, \frac{3}{4}) & t \text{ is odd} \end{cases}.$$

Then, one can verify via induction that $x_t = x_t^{\mathrm{nat}} = 1$ for all $t \geq 2$. Hence, for all $t \geq h + 1$, any $\pi \in \Pi_{\mathrm{feed}}^+(h) \cup \Pi_{\mathrm{drc}}^+(h)$ has a constant input $u_t^\pi = \bar{u}$. However, $c_t(x_t^\pi, u_t^\pi) + c_{t+1}(x_t^\pi, u_t^\pi) = (\bar{u} - \frac{1}{2})^2 + (\bar{u} - \frac{3}{4})^2$, which is greater than a universal constant. Hence, the regret must spaces as $\Omega(T - (h + 2))$.

**Proof of part c.** Fix policity $\pi_\star$ to select $u_t^{\pi_\star} = x_{t-1}^{\mathrm{nat}}$. Denote the sequences that arise from this policy as $(x_t^\star, u_t^\star)$. We set

$$c_t(x, u) = \frac{1}{4}\left((u - \frac{1}{2}u_t^\star)^2 + |x - x_t^\star|\right)$$

By construction $\pi_\star$ has zero cost on $c_t$. Now, set $w_t = 1$ for all $t$, and

$$A_t = \begin{cases} \frac{1}{4} & t \mod 3 = 1 \\ \frac{1}{4} & t \mod 3 = 2 \ , \\ 0 & t \mod 3 = 0. \end{cases} \quad B_t = \begin{cases} -\frac{1}{4} & t \mod 3 = 1 \\ -\frac{1}{20} & t \mod 3 = 1 \\ 0 & t \mod 3 = 0. \end{cases}$$

$$u^\star_{3k+1} = x^{\mathrm{nat}}_{3k+1} = 1$$
$$u^\star_{3k+2} = x^{\mathrm{nat}}_{3k+2} = \frac{5}{4}$$
$$u^\star_{3k+3} = x^{\mathrm{nat}}_{3k+3} = \frac{21}{16}.$$

Hence, a similar argument as in part (b), using the fact that that $w_t$ is constant but $u^\star_t$ is periodic, shows that any $\pi \in \Pi^+_{\mathrm{dac}}(h)$ suffers cost $\Omega(T - h - 3)$.

Let us now analyze the performance of policies $\pi \in \Pi_{\mathrm{feed}}(m, R_M)$. First, observe that $x^\star_t = 1$ for all $t \geq 2$.

$$x^\star_{3k+1} = 1$$
$$x^\star_{3k+2} = \frac{5}{4} - \frac{1}{4}u^\star_{3k+1} = 1.$$
$$x^\star_{3k+2} = \frac{21}{16} - \frac{1}{4}u^\star_{3k+1} - \frac{1}{20}u^\star_{3k+2}$$
$$= \frac{21}{16} - \frac{1}{4} - \frac{1}{20} \cdot \frac{5}{4} = 1.$$

Next, observe that for $\pi \in \Pi_{\mathrm{feed}}(h, R)$, and $t \geq h + 1$,

$$u^\pi_t = c + \sum_{i=0}^{h-1} M^{[i]} x^\pi_t = c + \sum_{i=0}^{h-1} M^{[i]} x^\star_t + \sum_{i=0}^{h-1} M^{[i]}(x^\pi_t - x^\star_t) = \underbrace{(c + \sum_{i=0}^{h-1} M^{[i]})}_{:=\bar{u}} + \sum_{i=0}^{h-1} M^{[i]}(x^\pi_t - x^\star_t)$$

Defining $\epsilon_t = u^\pi_t - \bar{u}$, we have

$$|\epsilon_t| \leq \left| \sum_{i=0}^{h-1} M^{[i]}(x^\pi_t - x^\star_t) \right| \leq R \max_{i=0}^{h-1} |x^\pi_{t-i} - x^\star_{t-i}|.$$

Hence, for integers $k$,

$$\max_{i \in [3]} |\epsilon_{3k+i}| \leq R \max_{t=3k-h+2}^{3k+3} |x^\pi_t - x^\star_t|. \tag{F.1}$$

We now argue a dichotomoty on the size of $\max_{i \in [3]} |\epsilon_{3k+i}|$. First, we show that if the epsilons are large, the costs incurred on a past window of $h$ must be as well. This is Eq. (F.1) would necessitate that $x^\pi_t$ differs from $x^\star_t$ over the previous window.

**Claim F.1.** *Suppose $\max_{i \in [3]} |\epsilon_{3k+i}| \geq \frac{1}{32}$. Then, $\sum_{t=3k-h+2}^{3k+3} c_t(x^\pi_t, u^\pi_t) \geq \frac{1}{2^7 R}$.*

*Proof.* By Eq. (F.1), we have that if $\max_{i \in [3]} |\epsilon_{3k+i}| \geq \frac{1}{32}$, then $\max_{t=3k-h+2}^{3k+3} |x^\pi_t - x^\star_t| \geq \frac{1}{32R}$. Since $c_t(x^\pi_t, u^\pi_t) \geq \frac{1}{4}|x^\pi_t - x^\star_t|$, the bound follows by upper bounding the maximum with the sum. $\square$

On the other hand, we show that if the $\epsilon$-terms are small, then the costs on $t \in \{3k+1, 3k+2, 3k+3\}$ are at least a small constant. This is because the inputs selected by $\pi$, $\bar{u} + \epsilon_t$, are close to constant, and therefore can fit the periodic values of $u^{\pi\star}_t$.

**Claim F.2.** *Suppose $\max_{i \in [3]} |\epsilon_{3k+i}| \leq \frac{1}{32}$. Then, $\sum_{i=1}^{3} c_{3k+i}(x^\pi_t, u^\pi_t) \geq 2^{-12}$.*

*Proof.* We expand

$$\sum_{i=1}^{3} c_{3k+i}(x_t^\pi, u_t^\pi) \geq \sum_{i=1}^{3} \frac{1}{4} \left(u_t^\star - u_t^\pi\right)^2$$

$$\geq \sum_{i=1}^{3} \frac{1}{4} \left(u_{3k+i}^\star - \bar{u} - \epsilon_{3k+i}\right)^2.$$

$$= \frac{1}{4}\left((1 - \bar{u} - \epsilon_{3k+1})^2 + \left(1 + \frac{1}{4} - \bar{u} - \epsilon_{3k+1}\right)^2 + \left(1 + \frac{1}{4} + \frac{1}{16} - \bar{u} - \epsilon_{3k+i}\right)^2\right).$$

In particular, suppose $\max_{i\in[3]} |\epsilon_{3k+i}| \leq \frac{1}{32}$. Then unless $|\bar{u} - 1| \leq \frac{1}{16}$, the above is at least $\frac{1}{4}\left(\frac{1}{32}\right)^2 = 2^{-12}$. On the other hand, if $|\bar{u} - 1| \leq \frac{1}{16}$. Then, $\frac{1}{4}\left(1 + \frac{1}{4} - \bar{u} - \epsilon_{3k+1}\right)^2 \geq (\frac{1}{4} - \frac{1}{16} - \frac{1}{32})^2 \geq \frac{1}{4}(\frac{1}{8})^2 \geq 2^{-12}$. $\square$

Combining both cases, we find that, for all $k$ such that $3k - h + 2 \geq 1$,

$$\sum_{t=3k-h+2}^{3k+3} c_t(x_t^\pi, u_t^\pi) \geq \frac{2^{-17}}{\max\{R, 1\}}.$$

In particular, we find that for $J_T(\pi; \mathcal{Z}_3) \geq \Omega(\frac{T}{h\max\{1,R\}})$ provided (say) $T \geq 10h$. $\square$

## F.2 Linear Regret Against DRC and DAC

In this section, we demonstrate linear regret against DRC and DAC. We consider a distribution over instances of the following form

$$A_t = 0, \quad B_t = \begin{bmatrix} 1 & 0 & 0 \\ 0 & \beta_t & 0 \\ 0 & 0 & 1 \end{bmatrix}, \quad w_t = -\begin{bmatrix} \omega_{t-1} \\ \omega_t \\ 1 \end{bmatrix}, t \geq 0. \tag{F.2}$$

Define the constant $\alpha = 1/4$. For a given $\sigma \in (0, 1/8)$, let $\mathcal{D}_\sigma$ denote the distribution over $(A_t, B_t, w_t)$ induced by drawing

$$\beta_t \overset{\text{i.i.d}}{\sim} [1 - \sigma, 1 + \sigma], \quad \omega_t \overset{\text{i.i.d}}{\sim} \{1 - \alpha\sigma, 1 + \alpha\sigma\}.$$

Note that these instances are (a) controllable, (b) stable, and (c) have variance scaling like $T\sigma^2$, and (d) the DRC and DAC parametrizations coincide. Letting $v[i]$ denote the $i$-th coordinate of vectors $v$, we consider cost of the form

$$c_f(x, u) = x[2]^2 + u[2]^2 + f(x[1]). \tag{F.3}$$

for either $f(z) = |z|$ or $f(z) = z^2$. Note that both choices of $f$ ensure that $c_f$ satisfies **??**, and the latter choice ensures that $c_f(x, u)$ is second order smooth.

**Theorem F.2.** *Let* alg *be any online learning algorithm. Let $c_f(x, u)$ as in (F.3). Then, for any $\sigma > 0$, there exist a DRC policy $\pi_\star \in \Pi_{\mathrm{drc}}(1, 1)$[4] such that expected regret incurred by* alg *under the distribution $\mathcal{D}_\sigma$ and cost $c_f(x, u)$ is at least*

$$\mathbb{E}_{\mathcal{D}_\sigma, \mathsf{alg}}[J_T(\mathsf{alg}) - J_T(\pi_\star)] \geq \cdot \begin{cases} C_1 T\sigma^2 & \text{for } f(z) = z^2, \ T \geq \frac{C_0}{\sigma^2} \\ C_1 T\sigma & \text{for } f(z) = z, \ T \geq \frac{C_0}{\sigma}, \end{cases}$$

*where above, $C_0, C_1$ are universal, positive constants.*

*Proof.* In what follows, $\mathbb{E}[\cdot]$ denotes expectation under the nstance from $\mathcal{D}_\sigma$, and any algorithmic randomness.

---

[4]Equivalently, in $\Pi_{\mathrm{dac}}(1, 1)$ since $A_t \equiv 0$

We expand $x_t$ and $u_t$ into its coordinates $x_t = (x_{t;1}, x_{t;2}, x_{t;3})$ and of $u_t = (u_{t;1}, u_{t;2}, u_{t;3})$. For any policy $\pi$, we can decompose its cost as

$$J_T(\pi) = \sum_{t=1}^{T} c_f(x_t^\pi, u_t^\pi) = (u_{T;2}^\pi)^2 + f(x_{T;1}^\pi) + \sum_{t=1}^{T-1} c_q(x_{t+1;2}^\pi, u_{t;2}^\pi) + f(x_{t;1}^\pi)$$

$$\text{where } c_q(x_{t+1;2}^\pi, u_{t;2}^\pi) = (x_{t+1;2}^\pi)^2 + (u_{t;2}^\pi)^2.$$

Note that $x_1 = 0$ in the above. The following lemma characterizes the conditional expectation of the $c_q$

**Lemma F.3.** *There exist a constant $c_{q;\star}$ such that*

$$\mathbb{E}[c_q(x_{t+1;2}, u_{t;2})] - c_{q;\star} = (2 + \sigma^2/3)\mathbb{E}[(u_{t;2} - \bar{u})^2],$$

*where $\bar{u} = \frac{1}{2(1+\sigma^2/6)}$.*

The proof of Lemma F.3 is deferred to the end of the section.

**Bounding the cost of $\pi^\star$**  We select $\pi^\star$ to be DAC (or equivalently, DRC since $A_t \equiv 0$) policy given by

$$M = \begin{bmatrix} 0 & 1 & 0 \\ 0 & 0 & \bar{u} \\ 0 & 0 & 0 \end{bmatrix}, \quad u_t^M = Mw_{t-1}.$$

We find that

$$\mathbb{E}[c_q(x_{t+1;2}^{\pi_\star}, u_{t;2}^{\pi_\star})] - c_{q;\star} = (2 + \sigma^2/3)\mathbb{E}[(u_{t;2}^M - \bar{u})^2] = 0, \quad 2 \le t \le T-1$$

$$\mathbb{E}[f(x_{t;1}^M)] = \mathbb{E}[f(u_{t-1;1}^M - \omega_{t-2})] = 0, \quad 3 \le t \le T.$$

Since the noise is uniformly bounded independent for all $\sigma, \alpha \le 1$, we conclude

$$\mathbb{E}[J_T(\pi^M)] - (T-1)c_{q;\star} = \underbrace{\mathbb{E}[f(x_{t;2}^\pi)]}_{\mathbb{E}[f(\omega_t)]} + (u_{T;2}^{\pi_\star})^2 + \underbrace{\mathbb{E}[f(x_{T;1}^\pi)]}_{=0} + \underbrace{\mathbb{E}[c_f(x_1, u_1)]}_{=0}$$

$$= \frac{f(1-\alpha\sigma) + f(1+\alpha\sigma)}{2} + \underbrace{(\bar{u} \cdot w_{t-1;3})^2}_{} = \bar{u}^2 \le 2., \qquad \text{(F.4)}$$

where we use $\alpha \le 1/24$, $\sigma \le 1/8$, and $\bar{u} \le \frac{1}{2(1+\sigma^2/6)}$, and $f(z) \le \max\{|z|, |z|^2\}$ to achieve the bound Eq. (F.4).

**Bounding the cost of adaptive policies**  Fix any online learning algorithm alg; we lower bound its performance. Because $B_t$ and $w_t$ are drawn from a fixed probability distribution, and are therefore oblivious to the learner's actions, we may assume without loss of generality that alg is deterministic.

The first step is to argue that any algorithm with small cost must select inputs where are bounded away from zero. Specifically, define the devent $\mathcal{E}_t := \{u_{t;2}^{\text{alg}} \ge 1/6\}$. Using Lemma F.3 together with $\bar{u} \ge 1/3$,

$$\mathbb{E}[J_T(\text{alg})] - Tc_{q;\star} \ge 2\sum_{t=1}^{T} \mathbb{E}\left[f(x_{t;1} + (u_{t;2}^{\text{alg}} - \bar{u})^2)\right]$$

$$\ge 2\sum_{t=1}^{T} \mathbb{E}\left[f(x_{t;1}^{\text{alg}})\right] + (\frac{1}{3} - \frac{1}{6})^2 \mathbb{P}[\mathcal{E}_t^c]$$

$$= \sum_{t=1}^{T} \mathbb{E}\left[f(x_{t;1}^{\text{alg}})\right] + \frac{1}{18}\mathbb{P}[\mathcal{E}_t^c]$$

$$\ge \sum_{t=3}^{T} \mathbb{E}\left[f(x_{t;1}^{\text{alg}}) \mid \mathcal{E}_{t-1}\right] \mathbb{P}[\mathcal{E}_{t-2}] + \frac{1}{18}\sum_{t=1}^{3} \mathbb{P}[\mathcal{E}_t^c]. \qquad \text{(F.5)}$$

We now lower bound $\mathbb{E}\left[f(x_{t;1}^{\text{alg}}) \mid \mathcal{E}_{t-2}\right]$, again deferring the proof to the end of the section.

**Lemma F.4.** *For any executable policy $\pi$ $\mathbb{E}[f(x_{t;1}^{\pi}) \mid \mathcal{E}_{t-2}] \geq \frac{1}{2}f(\alpha\sigma)$, provided $\alpha \leq 1/24$.*

Combining Lemma F.4 with the above bound, we have

$$\mathbb{E}[J_T(\mathsf{alg})] - (T-1)c_{\mathsf{q};\star} \geq \sum_{t=1}^{T} \frac{f(\alpha\sigma)}{2} \mathbb{P}[\mathcal{E}_{t-2}] + \frac{1}{18} \sum_{t=1}^{T} \mathbb{P}[\mathcal{E}_t^c]$$

$$\geq \min\left\{\frac{f(\alpha\sigma)}{2}, \frac{1}{18}\right\} \sum_{t=1}^{T-2} \mathbb{P}[\mathcal{E}_t] + \mathbb{P}[\mathcal{E}_t^c]$$

$$\geq (T-2)\min\left\{\frac{f(\alpha\sigma)}{2}, \frac{1}{18}\right\}.$$

With $\sigma \leq 1$ and $\alpha = \frac{1}{24}$, $\frac{f(\alpha\sigma)}{2} \leq \frac{1}{18}$. Combining with Eq. (F.4), we have

$$\mathbb{E}[J_T(\mathsf{alg})] - \mathbb{E}[J_T(\pi^\star)] \geq (T-2)\frac{f(\alpha\sigma)}{2} - 2.$$

The bound follows. $\qquad\qquad\square$

### F.2.1 Omitted proofs

*Proof of Lemma F.3.* Let $(\overline{\mathcal{F}}_t)_{t\geq 1}$ denote the filtration induced by setting $\overline{\mathcal{F}}_t$ to be the sigma-algebra generated by $(\beta_1, \ldots, \beta_{t-1}, \omega_1, \ldots, \omega_t)$. We have

$$\mathbb{E}[\beta_t^2 \mid \overline{\mathcal{F}}_t] = \frac{1}{2\sigma}\int_{u=1-\sigma}^{1+\sigma} u^2 \mathrm{d}u = \frac{(1+\sigma)^3 - (1-\sigma)^3}{6\sigma}$$

$$= \frac{1 + 3\sigma + 3\sigma^2 + \sigma^3 - (1 - 3\sigma^2 + 3\sigma^2 - \sigma^3)}{6\sigma}$$

$$= 1 + \sigma^3/3.$$

Set $c_{\mathsf{q}}(x,u) = x^2 + u^2$.

$$\mathbb{E}[c_{\mathsf{q}}(x_{t+1;2}, u_{t;2}) \mid \overline{\mathcal{F}}_t] = u_{t;2}^2 + \mathbb{E}[(\beta_t u_{t;2} - \omega_t)^2 \mid \overline{\mathcal{F}}_t]$$

$$= u_{t;2}^2(1 + \mathbb{E}[\beta_t^2 \mid \overline{\mathcal{F}}_t]) - 2u_{t;2}\mathbb{E}[\beta_t w_t \mid \overline{\mathcal{F}}_t] + \mathbb{E}[\omega_t^2 \mid \overline{\mathcal{F}}_t]$$

$$= u_{t;2}^2(2 + \sigma^2/3) - 2u_{t;2} + (1 + c\sigma^2)$$

at $u_{\star;2} = \frac{1}{2(1+\sigma^2/6)}$. Define $c_{\mathsf{q};\star} := \min_{u_{t;2}} \mathbb{E}[\ell_2(x_{t;2}, u_{t;2}) \mid \overline{\mathcal{F}}_t]$, we then have

$$\mathbb{E}[c_{\mathsf{q}}(x_{t+1;2}, u_{t;2}) \mid \overline{\mathcal{F}}_t] - \ell_{2;\star} = (2 + \sigma^2/3)(u_{t;2} - u_{\star;2})^2. \qquad\text{(F.6)}$$

$$\square$$

*Proof of Lemma F.4.* Let us introduce a second event, $\tilde{\mathcal{E}}$, defined as

$$\tilde{\mathcal{E}}_{t-1} := \left\{(1+\alpha\sigma) + u_{t;2}^{\mathsf{alg}}(1-\sigma) \leq x_{t-1;2}^{\mathsf{alg}} \leq (1-\alpha\sigma) + u_{t;2}^{\mathsf{alg}}(1+\sigma)\right\}.$$

Then, since $f$ is non-negative,

$$\mathbb{E}[f(x_{t;1}^{\pi}) \mid \mathcal{E}_{t-2}] \geq \mathbb{E}[\mathbb{I}\{\tilde{\mathcal{E}}_{t-1}\}f(x_{t;1}^{\pi}) \mid \mathcal{E}_{t-2}]$$

$$= \mathbb{P}[\tilde{\mathcal{E}}_{t-1} \mid \mathcal{E}_2]\mathbb{E}[\cdot\mathbb{E}[f(x_{t;1}^{\pi}) \mid \tilde{\mathcal{E}}_{t-1} \cap \mathcal{E}_{t-2}].$$

Let's first lower bound $\mathbb{P}[\tilde{\mathcal{E}}_{t-1} \mid \mathcal{E}_2]$. Writing $x_{t-1;2}^{\mathsf{alg}} = \beta_{t-2}u_{t;2}^{\mathsf{alg}} + \omega_{t-2}$, $\tilde{\mathcal{E}}_{t-1}$ occurs as soon as

$$(1+\alpha\sigma) + u_{t-2;2}^{\mathsf{alg}}(1-\sigma) \leq \beta_{t-2}u_{t-2;2}^{\mathsf{alg}} + \omega_{t-2}$$

$$(1-\alpha\sigma) + u_{t-2;2}^{\mathsf{alg}}(1+\sigma) \geq \beta_{t-2}u_{t-2;2}^{\mathsf{alg}} + \omega_{t-2}.$$

Using that $1 - \alpha\sigma \leq \omega_{t-2} \leq 1 + \alpha\sigma$ and rearranging the above, it is enough that

$$2\alpha\sigma \leq (\beta_t - (1-\sigma))u_{t-2;2}^{\mathsf{alg}}$$

$$-2\alpha\sigma \geq -((1+\sigma) - \beta_t)u_{t-2;2}^{\mathsf{alg}}.$$

Now, if $\mathcal{E}_{t-2}$ holds, that $u^{\text{alg}}_{t;2} \geq 1/6$. Furthermore, by construction $1 - \sigma \leq \beta_t \leq 1 + \sigma$. Therefore, if $\mathcal{E}_{t-2}$ holds, then $\tilde{\mathcal{E}}_{t-1}$ holds as long as

$$\beta_t - (1 - \sigma) \geq 12\alpha\sigma \quad \text{and} \quad (1 + \sigma) - \beta_t \geq 12\alpha\sigma.$$

In particular, for $\alpha = \frac{1}{24}$, then

$$\mathbb{P}[\tilde{\mathcal{E}}_{t-1} \mid \mathcal{E}_2] \geq \mathbb{P}[\beta_t \in [1 - \tfrac{2}{\sigma}, 1 + \tfrac{2}{\sigma}]] = \frac{1}{2}. \tag{F.7}$$

Next, we lower bound $\mathbb{E}[f(x^\pi_{t;1}) \mid \tilde{\mathcal{E}}_{t-1} \cap \mathcal{E}_{t-2}]$. To do so, we observe that $x_{t;1} = \omega_{t-2} - u^{\text{alg}}_{t-1;1}$. Moreover, since alg is deterministic (see discussion above), $u^{\text{alg}}_{t-1;1}$ is a deterministic function of $x^{\text{alg}}_{1:t-1}$, and moreover, $\tilde{\mathcal{E}}_{t-1}, \mathcal{E}_{t-2}$ are determinied by $x^{\text{alg}}_{1:t-1}$. Hence,

$$
\begin{aligned}
\mathbb{E}[f(x^\pi_{t;1}) \mid \tilde{\mathcal{E}}_{t-1}, \mathcal{E}_{t-2}] &= \mathbb{E}[\mathbb{E}[f(x^\pi_{t;1}) \mid x^{\text{alg}}_{1:t-1}] \mid \tilde{\mathcal{E}}_{t-1} \cap \mathcal{E}_{t-2}] \\
&= \mathbb{E}[\mathbb{E}[f(\omega_{t-2} - u^{\text{alg}}_{t-1;1}) \mid x^{\text{alg}}_{1:t-1}] \mid \tilde{\mathcal{E}}_{t-1} \cap \mathcal{E}_{t-2}] \\
&\geq \mathbb{E}[\min_{u \in \mathbb{R}} \mathbb{E}[f(\omega_{t-2} - u) \mid x^{\text{alg}}_{1:t-1}] \mid \tilde{\mathcal{E}}_{t-1} \cap \mathcal{E}_{t-2}].
\end{aligned}
$$

Thus, it suffices to characterize the distribution of $\omega_{t-1} \mid x^{\text{alg}}_{1:t-1}$ whenever the events $\tilde{\mathcal{E}}_{t-1} \cap \mathcal{E}_{t-2}$. Indeed, we claim that $\omega_{t-2} \mid x^{\text{alg}}_{1:t-1}$ is *uniform* on $\{1 - \alpha\sigma, 1 + \alpha\sigma\}$ on whenever $\tilde{\mathcal{E}}_{t-1}$ holds.

**Claim F.5.** *Let $\omega^- = 1 - \alpha\sigma$ and $\omega^+ = 1 + \alpha\sigma$. then (with probability one) $\mathbb{P}[\omega_{t-2} = \omega^- \mid x^{\text{alg}}_{1:t-1}] = \mathbb{P}[\omega_{t-2} = \omega^+ \mid x^{\text{alg}}_{1:t-1}] = \frac{1}{2}$ if $\tilde{\mathcal{E}}_{t-1}$ holds.*

*Proof.* If $\tilde{\mathcal{E}}_{t-1}$ holds, then there exists exactly two values $\beta$ and $\beta'$ in $[1 - \sigma, 1 + \sigma]$ such that

$$\omega^+ + \beta u^{\text{alg}}_{t-2;2} = \omega^- + \beta' u^{\text{alg}}_{t-2;2} = x^{\text{alg}}_{t-1;2}.$$

Even conditioned on $x^{\text{alg}}_{1:t-2}$, $\omega_{t-2}$ is uniformly distributed on $\{\omega^-, \omega^+\}$, and since $\beta$ and $\beta'$ have the same probability mass under the uniform distribution of $\beta_{t-2}$, it follows that $\mathbb{P}[\omega_{t-2} = \omega^+ \mid x^{\text{alg}}_{1:t-1}] = [\omega_{t-2} = \omega^- \mid x^{\text{alg}}_{1:t-1}]$ when $\tilde{\mathcal{E}}_{t-1}$ holds. $\square$

Hence,

$$\mathbb{E}[f(x^\pi_{t;1}) \mid \tilde{\mathcal{E}}_{t-1}, \mathcal{E}_{t-2}] \geq \min_{u \in \mathbb{R}} \frac{1}{2}(f(1 + \alpha\sigma - u) + f(1 - \alpha\sigma - u)).$$

For $f(z) = z^2$ or $f(z) = |z|$, the minimum is attained at $u = 1$, with values $\alpha^2\sigma^2$ and $\alpha\sigma$, respectively, both equal to $f(\alpha\sigma)$. Thus, combining with Eq. (F.7), we conclude

$$\mathbb{E}[f(x^\pi_{t;1}) \mid \mathcal{E}_{t-2}] \geq \frac{1}{2}f(\alpha\sigma).$$

$\square$

### F.3 Lower Bound without Stability

**Theorem F.3.** *Consider a scalar LTV system with $A = \rho \in [0,1]$, and $B_t$ drawn independently and uniformly at random from $\{-1\}$. Suppose that $w_1 = 1$, and $w_t = 0$ for all $t > 1$. Finally, let $c_t(x, u) = x^2$ be a fixed costs. Then,*

(a) *There then DRC policy $u^\pi_t = -\rho B_2 x^{\text{nat}}_t$, DAC policy $u^\pi_t = -\rho B_2 w_{t-1}$, and static-feedback policy $u^\pi_t = -\rho B_2 A$ (all chosen with foreknowledge of the $(B_t)_{t \geq 1}$ sequence) all enjoy:*

$$J_T(\pi) = 1, \quad \text{with probability 1.}$$

(b) *Any online learning algorithm without foreknowledge of $(B_t)_{t \geq 1}$ must suffer expected cost*

$$\mathbb{E}[J_T(\pi)] = \Omega\left(\min\left\{T, \frac{1}{1 - \rho}\right\}\right).$$

*Proof.* In part (a), all policies choose $u_2 = -B_2\rho w_1$, so that $x_3 = \rho x_2 - \rho w_1 = 0$. Since $0$ is an equilibrium point and $w_t = 0$ for all $t \geq 2$, the system remains at zero. Hence, the only cost incurred is at times 1 and 2, which are costs of zero and 1 respectively. In part (b), we use the unbiasedness of $B_t$ to recurse

$$\mathbb{E}[x_2] = Ax_1 + \mathbb{E}[B_1]\mathbb{E}[u_1] + w_1 = 1$$
$$\mathbb{E}[x_{t+1}] = A\mathbb{E}[x_t] + \mathbb{E}[B_t]\mathbb{E}[u_t] + \underbrace{w_t}_{=0} = \rho\mathbb{E}[x_t], \quad \forall t \geq 2,$$

yielding $\mathbb{E}[x_t] = \rho^{t-2}$ for all $t \geq 2$. Hence, the expected cost of the policy is

$$\mathbb{E}[J_T(\pi)] = \sum_{t=2}^{T} \rho^{(2t-2)}.$$

Considering the cases where $\rho \geq \frac{1}{T-2}$ and $\rho \leq \frac{1}{T-2}$, we find the above is $\Omega\left(\min\left\{T, \frac{1}{1-\rho}\right\}\right)$. □

### F.4 Hardness of Computing Best State Feedback Controller

Consider a time-varying linear dynamical system with no noise:

$$x_{t+1} = A_t x_t + B_t u_t + w_t, \quad \forall t, \, w_t \equiv 0$$

subject to changing convex costs $c_t(x, u)$. We show that even in the no-noise setting properly learning the *optimal state feedback* policy is computationally hard. This statement holds with the control agent having full prior knowledge of the dynamics $(A_t, B_t, c_t)$. It relies on a reduction to the MAX-3SAT problem which is NP-hard. Our lower bound is inspired by the analogous one for discrete MDPs by [14].

**Theorem F.4.** *There exists a reduction from* MAX-3SAT *on $m$-clauses and $n$-literals to the problem of finding the state feedback policy $K$ optimal for the cost $\sum_{t=1}^{T} c_t(x_t^K, u_t^K)$, over sequentially stable dynamics given by $(A_t, B_t, c_t)$: a solution to* MAX-3SAT *with $k$ value implies optimal cost of at most $-k$, and a solution $K$ to the control problem with $-k - \epsilon$ value implies optimal value of* MAX-3SAT *at least $k$ for any known $\epsilon > 0$.*

Let us first describe the construction of the dynamics that reduce the optimal control problem to the MAX-3SAT problem. Consider a 3-CNF formula $\phi$ with $m$ clauses $C_1, \ldots, C_m$ and $n$ literals $y_1, \ldots, y_n$. The state space is of dimensionality $d_x = n + 1$ and the action space is of dimensionality $d_u = 2$. The control problem is given as a sequence of $m$ episodes corresponding to the clauses of the formula $\phi$.

For a single clause $C_j$ with $j \in [m]$, let the dynamics $(A_t, B_t, c_t)_{t=1}^{n+2}$ be an episode of length $n + 2$ constructed as follows. The initial state is $x_1 = [1, \mathbf{0}_n]^\top$. The state transitions are independent of the clause itself given by the following $A_t \in \mathbb{R}^{n+1 \times n+1}$:

- for $1 \leq t < n$, $A_t(t+1) = [\mathbf{1}_n, 0]^\top$, $A_t(n+1) = [\mathbf{0}_n, 1]^\top$, $A_t(i) = [\mathbf{0}_{n+1}]^\top$ for all other $i \neq t+1, n+1$.
- for $t = n$, it becomes $A_t(n+1) = [\mathbf{1}_{n+1}]^\top$ and $A_t(i) = [\mathbf{0}_{n+1}]^\top$ for all other $i \neq n+1$.
- for $t = n+1$ take $A_t(1) = [\mathbf{1}_{n+1}]^\top$ and $A_t(i) = [\mathbf{0}_{n+1}]^\top$ for all other $i \neq 1$; for $t = n+2$ take $A_t = \mathbf{0}_{n+1 \times n+1}$ to ensure sequential stability.

The action matrices $B_t$ along with the costs $c_t$, on the other hand, depend on the content of the clause $C_j$ itself. In particular, let $I_j$ be the set of indices of the literals that are in clause $C_j$. We define the regularity cost to be $c(x, u) = S_x(x) + (1 - x(n+1))^2 \cdot S_u(u)$, where $S_x(\cdot) = \text{dist}(\cdot, \Delta_{n+1})$ and $S_u(\cdot) = \text{dist}(\cdot, \Delta_2)$ are the distance functions to the simplex sets of corresponding dimensionality. The action matrices and costs are given as follows:

- for $1 \leq t \leq n$ and $t \notin I_j$, $B_t = \mathbf{0}_{n+1 \times 2}$ and $c_t(x, u) = c(x, u)$.
- for $1 \leq t \leq n$ and $t \in I_j$, if $y_t \in C_j$: then $B_t(t+1) = [-1, 0]^\top$, $B_t(n+1) = [1, 0]^\top$ and $B_t(i) = [0, 0]^\top$ for all the other $i \neq t+1, n+1$; the cost is $c_t(x, u) = c(x, u) - u(1) \cdot (1 - x(n+1))$ rewarding the action $[1, 0]$ which corresponds to assigning the literal a value $y_t = 1$.

- for $1 \leq t \leq n$ and $t \in I_j$, if $\neg y_t \in C_j$: then $B_t(t+1) = [0, -1]^\top$, $B_t(n+1) = [0, 1]^\top$ and $B_t(i) = [0, 0]^\top$ for all the other $i \neq t+1, n+1$; the cost is $c_t(x, u) = c(x, u) - u(2) \cdot (1 - x(n+1))$ rewarding the action $[0, 1]$ which corresponds to assigning the literal a value $y_t = 0$.

- for $t = n + 1$, $B_t = \mathbf{0}_{n+1 \times 2}$ and for $t = n + 2$, $B_t(1) = [1, 1]^\top$ and $B_t(i) = [0, 0]^\top$ for all other $i \neq 1$; for both $t = n + 1, n + 2$, the costs are $c_t(x, u) = c(x, u)$.

Note that the last two rounds $t = n + 1, n + 2$ for a clause ensure sequential stability and identical starting state $x_1 = [1, \mathbf{0}_n]^\top$.

**Lemma F.6.** *The described system $(A_t, B_t)$ is sequentially stable and the costs $c_t$ are convex in $x, u$.*

*Proof.* By the construction of the state matrices $A_t$, we know that for any $t \geq n + 2$ the operator $\Phi_t^{[n+2]} = 0$ implying sequential stability of the system. To show convexity, note that the distance function $g(\cdot) = \text{dist}(\cdot, \mathcal{S})$ for any convex and compact set $\mathcal{S}$ is a convex function. More specifically, for $z \in \mathbb{R}^{d_z}$ it is given by $g(z) = \min_{w \in \mathcal{S}} \|z - w\|$. This is straightforward to show: take any $z_1, z_2 \in \mathbb{R}^{d_z}$, and let $w_1, w_2 \in \mathcal{S}$ be the closest points to $z_1, z_2$ respectively, i.e. $\|w_1 - z_1\| = g(z_1)$ and $\|w_2 - z_2\| = g(z_2)$. For any $\lambda \in [0, 1]$, given the convexity of the set $\mathcal{S}$, we know that $\lambda w_1 + (1 - \lambda)w_2 \in \mathcal{S}$, which concludes the convexity proof for $g$:

$$
\begin{aligned}
\lambda g(z_1) + (1 - \lambda)g(z_2) &= \lambda \|z_1 - w_1\| + (1 - \lambda)\|z_2 - w_2\| \\
&\geq \|\lambda(z_1 - w_1) + (1 - \lambda)(z_2 - w_2)\| \\
&= \|\lambda z_1 + (1 - \lambda)z_2 - (\lambda w_1 + (1 - \lambda)w_2)\| \\
&\geq g(\lambda z_1 + (1 - \lambda)z_2).
\end{aligned}
$$

This means that both $S_x(\cdot)$ and $S_u(\cdot)$ are convex functions since the simplex is convex in any dimension. The construction of the costs $c_t$ is based on these two functions as well as linear components in $x, u$, hence all $c_t$ costs are convex in $x, u$. $\square$

**Lemma F.7.** *If there exists an assignment of literals $y \leftarrow v$ s.t. the formula $\phi$ has $k \in [1, m]$ satisfied clauses, then there is a corresponding linear policy $K \in \mathbb{R}^{2 \times n+1}$ that suffers the exact cost of $-k$.*

*Proof.* This should be evident from the construction itself. Let $v_i = 1$ assignment correspond to $\bar{v}_i = [1, 0]^\top$ and $v_i = 0$ to $\bar{v}_i = [0, 1]^\top$. Then consider the linear policy $K = [\bar{v}_1, \ldots, \bar{v}_n, \mathbf{0}_2^\top]$. Denote $e_1, \ldots, e_{n+1} \in \mathbb{R}^{n+1}$ to be the basis vectors of the space. Note that according to the defined $K$, if $x_t$ is a basis vector of $\mathbb{R}^{n+1}$, then $u_t = K x_t$ is a basis vector of $\mathbb{R}^2$. It is straightforward to check by our construction that $u_t$ being a basis vector of $\mathbb{R}^2$ implies $x_{t+1}$ is a basis vector of $\mathbb{R}^{n+1}$. Since $x_1 = e_1$, then the state-action pairs when following policy $K$ are both basis vectors, and satisfy the regularity conditions of $c(x, u)$. This means that the policy $K$ plays $\bar{v}_t$ if $x_t(t) = 1$ and plays $\mathbf{0}_2$ if $x_t(n+1) = 1$. Hence, if the clause $C_j$ is satisfied by the assignment $y \leftarrow v$, then the cost of $K$ over the episode is exactly $-1$, i.e. once the clause is satisfied, $-1$ is accrued and the state moves to the sink $e_{n+1}$. If the clause is not satisfied, then the cost is 0 since $c(x, u)$ is 0 throughout. This means that the constructed linear policy $K$ over the whole control sequence suffers cost $-k$.i k $\square$

**Lemma F.8.** *If there exists a state feedback policy $K \in \mathbb{R}^{2 \times n+1}$ s.t. following the actions $u_t = K x_t$ results in cost at most $-k - \epsilon$ for any $k \in [1, m]$ and any $\epsilon \in (0, 1)$, then there is a literal assignment $y \leftarrow v$ s.t. the formula $\phi$ has at least $k$ satisfied clauses.*

*Proof.* Let the linear policy matrix be given as $K = [\bar{v}_1, \ldots, \bar{v}_n, \bar{v}_{n+1}]$. The proof consists of two main components: (i) we argue that the policy $K^*$ with $\bar{v}_i^* = \arg\min_{v \in \Delta_2} \|v - \bar{v}_i\|$ for $1 \leq i \leq n$ and $\bar{v}_{n+1}^* = \mathbf{0}_2$ is at least as good as $K$ in terms of cost up to an approximation factor $\epsilon$; (ii) we show that for $K$ that satisfies the constraints, the randomized policy $\hat{K}$ that has $\hat{K}(i) = [1, 0]^\top$ w.p. $\bar{v}_i(1)$ and $\hat{K}(i) = [0, 1]^\top$ w.p. $\bar{v}_i(2)$ (as well as $\hat{K}(n+1) = \mathbf{0}_2$) suffers expected cost at most that of $K$ itself.

Suppose these two claims are true, then the described randomized linear policy $\hat{K}$ has expected cost at most $-k$, which means that there exists a deterministic linear policy with first $n$ columns as basis vectors, i.e. $[1, 0]^\top$ or $[0, 1]^\top$, that suffers cost at most $-k$. It follows that the corresponding assignment of literals given by the first $n$ columns of the linear policy $y \leftarrow v$ satisfies at least $k$ out of the $m$ clauses, so $\phi$ has at least $k$ satisfied clauses.

To prove (i), first notice that the policy $K^*$ suffers non-positive cost over the entire horizon since it satisfies the necessary constraints given by the regulatory cost $c(x, u)$. Note also that said $c(x, u)$ can be scaled by any constant $M_\epsilon > 0$. Now suppose that under the condition $\min_i \|\bar{v}_i - \bar{v}_i^*\| \le \ell_\epsilon$ the cost difference of $K^*$ and $K$ is bounded by $\epsilon$: the choice of $\ell_\epsilon$ can depend on any problem parameters, and since the construction is over $T = \Theta(mn)$ overall rounds (finite), such a choice is always possible. Hence, for all such $K^*$ we automatically infer that it suffers cost at most $-k$ and satisfies the necessary constraints. On the other hand, if the condition does not hold, then the distance of $\bar{v}_i$ from $\Delta_2$ is bounded from below by $\ell_\epsilon$, meaning that for a sufficiently large choice of $M_\epsilon > 0$ (given knowledge of $\epsilon$ and all other parameters), the overall cost suffered by $K$ will be positive due to $c(x, u)$, i.e. it will have a higher cost than $K^*$. Therefore, any state feedback policy $K$ can be approximately replaced by $K^*$ that satisfies the constraints ensured by $c(x, u)$.

To show (ii), for a policy $K$ that does satisfy these constraints, i.e. $\bar{v}_i \in \Delta_2$ for $1 \le i \le n$ and $\bar{v}_{n+1} = \mathbf{0}_2$, we show that its randomized version $\hat{K}$ is at least just as good in terms of expected cost. Proving this claim is a matter of unrolling the dynamics for a single clause $C_j$. The order, the indices and negation or not of the literals in $C_j$ does not affect the cost, so w.l.o.g. assume we have $C_1 = y_1 \vee y_2 \vee y_3$. The cost of a general policy $K$ over first 3 iterations is given by

$$-\bar{v}_1(1) - (1 - \bar{v}_1(1))^2 \cdot \bar{v}_2(1) - (1 - \bar{v}_1(1))^2 \cdot (1 - \bar{v}_2(1))^2 \cdot \bar{v}_3(1)$$

The alternative randomized linear policy instead suffer expected cost given by

$$-\bar{v}_1(1) - (1 - \bar{v}_1(1)) \cdot \bar{v}_2(1) - (1 - \bar{v}_1(1)) \cdot (1 - \bar{v}_2(1)) \cdot \bar{v}_3(1)$$

which is straightforward to show to be not larger than the original cost. $\square$

*Proof of Theorem F.4.* Lemma F.6 above show that the given LTV system construction along with the costs satisfies the theorem conditions. Lemmas F.7 and F.8 indicate that the MAX-3SAT problem can be reduced to the LTV control, in particular proper learning of state feedback policies in this setting. Given that MAX-3SAT is NP-Hard even in its decision form implies the computational hardness of the offline optimization of LTV optimal state feedback policies. $\square$