# OpenReview forum: "Online Control of Unknown Time-Varying Dynamical Systems"
_NeurIPS.cc/2021/Conference — NeurIPS 2021 Poster_

### Official Review · Reviewer_w6Bk · 2021-07-15

**Rating:** 8
**Confidence:** 4

**Summary:**

This paper studies the online control of unknown Linear Time-Varying (LTV) systems in the nonstochastic control setting. They present in-depth theoretical results given the aforementioned problem setting that shows the following (in the order presented in the paper):
1. Statement that the regret for the DAC/DRC policies cannot scale sub linearly in the time horizon (lower bound)
2. An algorithmic upper-bound for online prediction
3. Applying (2) to the online control problem
4.Theoretical analysis showing that sublinear regret is possible but in non-polynomial time (computationally inefficient) unless P=NP)

**Limitations And Societal Impact:**

No issues were noted.

**Main Review:**

Quality:
+The authors provide a comprehensive yet concise literature survey, distinguishing the results by problem setting (i.e. LTV/LTI, known/unknown dynamics, nonstochastic/stochastic noise, full-state/partial-state observation)
+ The authors provide in-depth theoretical results
+ The authors are very clear on the assumptions made in their analysis
- The paper could be improved to have simulation results similar to citation #16 (Adaptive regret for control of time-varying dynamics) though I understand that the main result is sufficient.
- It would be helpful to note that citation #20 (The nonstochastic control problem) studies unknown time-invariant dynamics (to clarify the difference with the authors’ work)


Clarity:
+ Overall, the paper is well written and organized (the flow is readable i.e. 1. Statement that the regret for the DAC/DRC policies cannot scale sub linearly in the time horizon (lower bound) -> 2. Proposing algorithmic upper-bound for online prediction -> 3.Applying this algorithm to the online control problem -> 4.Theoretical analysis showing that sublinear regret is possible but in non-polynomial time (computationally inefficient) unless P=NP)
+ They provide very detailed theoretical analysis

Originality:
+ After a brief search, it does appear that they are the first to look at the setting: Unknown LTV system dynamics in the non stochastic control setting.
+ The results of studying and comparing the three classes of the controllers are novel, fresh, and interesting. The reviewer also gains new insights about the three classes of controllers.



**Time Spent Reviewing:**

9 hours

---

> ### Author Response · Authors · 2021-08-10
> **Review rebuttal**
>
> Thank you! We will note that [20] is time-invariant in the revision. We chose not to include simulation results as in our opinion it would distract from the theoretical results we believe are important to emphasize (e.g. the lower bounds).

---

### Official Review · Reviewer_vpa4 · 2021-07-16

**Rating:** 7
**Confidence:** 3

**Summary:**

This paper studies online control in linear time-varying (LTV) systems where the dynamics are unknown. The performance metric is the adaptive regret against three common classes of policies. When the benchmark policy class is DRC/DAC, the authors show a lower bound of the adaptive regret of any online controllers that grows linearly with respect to the length of horizon. This negative result demonstrate that the unknown LTV setting is qualitatively harder than either unknown linear time invariant (LTI) or known LTV, where sublinear regrets are achievable. On the other hand, they also propose an online control algorithm whose adaptive regret matches the lower bound in the same setting.  When the benchmark policy class is static feedback controllers, they show that although sublinear regret is achievable under some restrictive assumptions, planning over this policy class is computational inefficient in general.

**Limitations And Societal Impact:**

Yes. I did not see any potential negative societal impact of this work.

**Main Review:**

This paper makes original and significant contribution to the online control literature. The problem setting under consideration is very important. As the authors discussed in the introduction section, since many positive results on adaptive regret have come out in either unknown linear time invariant (LTI) or known LTV settings, it is natural to ask whether these results extend to the unknown LTV setting, which is more common in practice. A negative lower bound result in this setting helps to demonstrate the hardness of this problem, and prompts us consider other performance metrics (e.g. competitive ratio) as discussed in Section 5. A limitation is that Assumption 1 and 2 seem to be restrictive. Although Appendix A.2 says the results still hold if a nominal stabilizing controller is known, it is unclear why such controller must exist and how to find it.

The paper is well-organized and written clearly. All claims are supported by theoretical results, and the assumptions are stated clearly. Although I did not check the proof in detail, the results look technically sound to me.


**Time Spent Reviewing:**

6

---

> ### Author Response · Authors · 2021-08-10
> **Review rebuttal**
>
> Thank you for the review. The existence and foreknowledge of a stabilizing controller is an interesting question, and one that is beyond the scope of this work. For systems which vary slowly enough, one could imagine periodic coarse system identification phases which identify stabilizing (but suboptimal) controllers, which can be refined further by online learning in the suitable convex parametrization. We can discuss this more in the appendix. We can also include further literature citations to express that stabilization of general LTV systems is an open and challenging problem more broadly.

---

### Official Review · Reviewer_xwEp · 2021-07-16

**Rating:** 7
**Confidence:** 4

**Summary:**

This paper studies LTV systems control, which is a very challenging problem in my opinion. A set of interesting results have been obtained, including the non-equivalence between different control classes, and algorithms with adaptive regret bounds against different controller classes. Computational complexity in achieving sublinear regret against state-feedback controller has also been discussed.

In general, I think this paper makes interesting contributions. However, I have some reservations on the notion of "adaptive regret". Further, I find some places in the presentation that need more discussions and clarifications.

I will be happy to revise the score after rebuttal if these points are clarified.

**Limitations And Societal Impact:**

More discussions on the notion of adaptive regret should be included, including its limitations.

**Main Review:**

On adaptive regret:

- In (2.1), the algorithm $\mathcal{A}$ is not allowed to depend on interval I, correct?
- In the second term in (2.1), the $u_t^\pi$ and $x_t^\pi$ is obtained by applying $\pi$ from the very beginning t=0, although the summation in (2.1) is only on the interval I?
- Why in the benchmark, one needs to consider a policy class? Why can’t we compare to the optimal offline solution when everything is known in hindsight?
- Also, in the second term in (2.1), the policy $\pi$ appears to be time-invariant, i.e. the policy parameters $M^{[i]}$ is not allowed to depend on the time instances  $t \in I$, is this correct? If so, if one only cares about $I = [T]$, the entire horizon, then comparing against a static, time-invariant policy doesn’t seem to make sense.
- I find it not straightforward to understand adaptive regret. Can a low adaptive regret guarantee stability, i.e. the state size does not blow up as T grows?

Assumptions: I find the open-loop stable assumption 1 to be strong. One of the basic (and oftentimes challenging) tasks in control is stabilization, yet this is put as an assumption. More discussions are needed.

Results: I find Theorem 2.1 to be very interesting. Is there any intuition about this result? Having more discussions will be highly helpful.

The control algorithm used in Algorithm 2 is DRC-OGD, but very few details are provided in the main text. I suggest adding more details here in the main text, e.g. what the algorithm is based on, and what is its guarantee for known systems. One question is does the control algorithm has to be DRC-OGD, or any algorithm that satisfies a certain regret guarantee for knonw systems will work?

Thm 4.1: under a suitable stabilization assumption? What is the meaning?

Line 313: What is the “nonparametric rate of regret” in the paragraph after Thm 4.1?

I am not sure what’s the point Thm 4.2 is trying to make. The paragraph before Thm 4.2 suggests the purpose of Thm 4.2 is to show the computational intractability in Thm 4.1 may be hard to avoid. But Thm 4.2 only shows the NP-hardness of finding the optimal state feedback control $K$. If I understand correctly, if one tries to compare against the best state feedback control, but the strategy is not limited to searching over a state feedback control, but search over some larger controller class instead, maybe it is still possible to achieve computational efficiency and sublinear regret?

In the paragraph after Thm 4.2, it says “…there may be more clever convex relaxations (other than DRC and DAC, which provably cannot work)…” I guess the reason that DRC and DAC would not is due to Thm 2.1, if I understand correctly?


**Time Spent Reviewing:**

5

---

> ### Author Response · Authors · 2021-08-10
> **Review rebuttal**
>
> Thank you for your review. We address each question in order:
> 1. Yes, the algorithm is fixed and must have low regret for all intervals.
> 2. Yes, that is correct; the costs are applied to a fixed trajectory full-length, though the summation is over I. This makes sense because you cannot restart the system at the beginning of each interval
> 3. It is impossible to get sublinear regret with respect to unrestricted policy classes, even for time invariant systems. This point is explained in a work ([*] Li et al. 2019), which we will cite in the revision. Recent work which compares to the best in hindsight policy either requires future lookahead, or considers competitive ratio ([**] Goel, Hassibi 2020).
> 4. Yes, the policy is time-invariant. Note that a fixed policy on the full interval is standard regret, and this is why adaptive regret is so important. A fixed policy might perform poorly on the whole time scale [1,T], while on sub-intervals, there may be better (fixed for each sub-interval) policies. Because of this, **adaptive** regret guarantees *do* imply low regret for changing policies.
>
> For illustrative purposes, consider the case of k-switching online LQR given by $(A_1, B_1), … (A_k, B_k)$ where one has no system information (including if a switch has occurred). In this case, our results guarantee $O(T^{2/3})$ regret on each subinterval $I_j$ (notice that the subinterval variability term is 0) against the best fixed DRC policy on that interval, and, due to class equivalence in LTI dynamics, for sufficiently large m, against the best state feedback controller $K_j$ too (because, over a window where dynamics aren’t changing, DRC does approximate feedback [39]). Hence, over the entire trajectory, despite not knowing when the switches occur and naturally also not what the new matrices are, we obtain $O(k T^{2/3})$ regret against the policy that switches between the optimal fixed subinterval policies $\pi_1, … \pi_k$ (either DRC or state feedback) *at exactly the correct time*. In this setup, this is the most natural and desirable policy to compete against. A standard regret guarantee would have only permitted us to compare to the best fixed joint DRC policy over the entire trajectory which is a significantly less powerful comparator (and in this case we would also no longer be able to compare to state feedback policies).
>
> More generally, the metric of adaptive regret [21] has been widely used in the online learning community to deal with changing environments [2, 11, 47]. Since adaptive regret is the supremum over local regret on any interval, a low rate bound ensures good performance against changing predictors (think optimal predictors for different subintervals): this implies adaptation to changes in the environment. As illustrated in the example above, we believe this intuition carries over to LTV systems.
>
> 5. Regarding Stability: Our assumption, which is common in the non stochastic control literature, is that one has access to pre-stabilizing controllers so that one can build parameterizations (based on Youla (DRC) or SLS (DAC) which ensure stability a priori (see e.g. [3] and [39])). For simplicity, the body of the paper considers open-loop stable systems; extensions to pre-stabilized systems are discussed at length in App A.
> 6. In light of point 3, regret must consider restricted policy benchmarks. The point of Thm 2.1. Is that, while 3 popular policy classes are roughly equivalent for time invariant linear systems (e.g. [3], [39]), they can have very different performance when dynamics vary.
> 7. The algorithm used is Algorithm 1 from [39] (which proves its standard regret guarantee for known LTI systems). The algorithm DRC-OGD (called DRC-GD in the original paper) simply runs online gradient descent on the DRC parameterization. In Appendix B we show that this algorithm naturally attains essentially the same adaptive regret rate on LTV systems as well. We will add this note in the main text (currently in App B) — thank you for noting this unclarity.
>
> To obtain Thm 3.4, beyond having an algorithm with low regret against some policy class $\Pi$, one needs to analyze the error sensitivity of the chosen control algorithm and the policy class, i.e. how their performance degrades as a function of the error in system estimation. In practice the proposed adaptive estimation could be put on top of any control algorithm C, but the way to get a final regret bound is not black-box: one needs to actually bound the extra cost incurred by C when acting on the estimated rather than true system. We therefore used an explicit control algorithm, DRC-OGD, in order to get precise rates and dependence on all parameters of an actually executable algorithm.
> 8. As stated below theorem 4.1, the formal statement is given in Appendix E. This contains the ``suitable stabilization assumption’’.
> 9. Regret guarantees where epsilon regret requires T = eps^{-poly(dimension)} are called non-parameteric because they reflect common statistical rates in nonparametric regression (see citation [33] provided for further details).
> 10. Your intuition is mostly correct in the assessment of Thm 4.2. We provide Theorem 4.2 as, to quote our text, “strong evidence” that competing with the best K is hard, but acknowledge on lines 329-330 that this does not rule out stronger algorithms. As you noted, these stronger algorithms cannot be based on the DRC/DAC parameterizations in the LTV case given Theorem 2.1. However, Theorem 4.2 does rule out a large class of possible algorithms, and suggests there are at least fundamental algorithmic challenges to competing with this class of policies. Even if there are algorithms which yield better regret by relaxation, the impossibility of optimizing over K remains of independent interest.
>
>
>
> [2]	Dmitry Adamskiy, Wouter M Koolen, Alexey Chernov, and Vladimir Vovk. A closer look at adaptive regret. The Journal of Machine Learning Research, 17(1):706–726, 2016.
>
> [3] 	Naman Agarwal, Brian Bullins, Elad Hazan, Sham Kakade, and Karan Singh. Online control with adversarial disturbances. In International Conference on Machine Learning, pages 111– 119, 2019.
>
> [11]	Amit Daniely, Alon Gonen, and Shai Shalev-Shwartz. Strongly adaptive online learning. In Francis Bach and David Blei, editors, Proceedings of the 32nd International Conference on Machine Learning, volume 37 of Proceedings of Machine Learning Research, pages 1405–1411, Lille, France, 07–09 Jul 2015. PMLR.
>
> [21]	Elad Hazan and Comandur Seshadhri. Efficient learning algorithms for changing environments. In Proceedings of the 26th annual international conference on machine learning, pages 393–400. ACM, 2009.
>
> [33]	Alexander Rakhlin and Karthik Sridharan. Online non-parametric regression. In Conference on Learning Theory, pages 1232–1264. PMLR, 2014.
>
> [39]	Max Simchowitz, Karan Singh, and Elad Hazan. Improper learning for non-stochastic control, 2020. In Conference in Learning Theory, pages 3320-3436, 2020.
>
> [47]	Lijun Zhang, Tie-Yan Liu, and Zhi-Hua Zhou. Adaptive regret of convex and smooth functions. arXiv preprint arXiv:1904.11681, 2019.
>
> [*]	Yingying Li, Xin Chen, and Na Li. Online optimal control with linear dynamics and predictions: Algorithms and regret analysis. In NeurIPS, pages 14858–14870, 2019.
>
> [**]	Gautam Goel, Babak Hassibi. Regret-optimal control in dynamic environments. arXiv:2010.10473v2, 2020.

---

> > ### Comment · Reviewer_xwEp · 2021-08-25
> > **Good response**
> >
> > Thanks for the response! Many of the discussions and go to the final version (if accepted).

---

### Official Review · Reviewer_kAKa · 2021-07-18

**Rating:** 3
**Confidence:** 5

**Summary:**

The paper studies RL policies for linear time-varying systems.

**Ethics Review Area:**

["I don’t know"]

**Main Review:**

Most of the points discussed in the first round of reviewing are not addressed in the rebuttal. Further, the explanations in the rebuttal together with the antagonistic responses of the authors show that the manuscript suffers from serious presentation issues.
___________________________________________________________________
I think that the work is in an early stage of preparation and so cannot be recommended for publication. In general, connections to relevant literature and where this work stands in a bigger picture is missing, presentation is hard to read and harder to grasp for an expected reader, and motivations of the work and the directions along which it can be helpful are not clear.

To the reader, the technical parts look isolated and different pieces do not connect well to deliver the potential messages. The authors rely too much on the supplementary file so that the paper lost its self-sufficiency. Accordingly, the language is too strong and super confident while the writing suffers from a sub-standard presentation by leaving the burden of decoding multiple claims in every line and justifying them to the reader. I suggest rewriting major portions and providing more concise presentation, using google scholar to find more-relevant literature and clearly and comprehensively discussing the connections. Unfortunately, the impression of the reader from the current version is that the manuscript focuses on technicalities and details of a problem, while presenting as if it is providing deep conceptual understandings of an extensive area.

From a technical point of view, the results are expected, e.g., if the environment keeps changing, then clearly it is not learnable! The adaptive regret that the manuscript relies on in several places is not sufficiently controlled, and the assumptions appear to be so rich for the presented results. I think it is good that some classes of policies are compared, but to provide a nice tutorial to the reader, the current presentation is too compact and needs more intuitions and insights. The premises are not clear not to be very restrictive. E.g., in (2.1), the benchmark has general notation but later in the manuscript, it depends on the interval under study. It is said that the disturbance is revealed to the learner, which is a consequential assumption.

Most of the discussions on pages 6, 7, 9 are in a different context than that of the problem under study on pages 1-5.

Finally, there are multiple typos and frequent unclear sentences that I refrain mentioning here due to abundance.




**Time Spent Reviewing:**

10

---

> ### Author Response · Authors · 2021-08-10
> **Review rebuttal**
>
> Thank you for the review. For a bit of background, this paper studies the nonstochastic control setting, which has received substantial attention in the past two years. In the paper, we direct the reader to previous sources [3,20,39] which explain this setting and its motivation in fuller detail. However, given (a) the space constraints of the NeurIPS format and (b) the substantial body of work in this space (see later half of “Low-Regret Control” passage in 1.2 Related work), we elected to provide a more compact description of our problem setting and notion of performance. We would appreciate any detailed feedback about specific areas in which our exposition could be improved, which technical matters were unclear, or which further references were missing.
>
> Regarding the strength of claims, we will be sure to explain that they pertain to the nonstochastic control setting in particular; of course, there are other notions of online learning for control (e.g. with lookahead or competitive analysis) and we will be clear that our results do not have bearing on these equally interesting settings.
>
> We concur that if the system is changing, it is not learnable in a natural sense. However, our results are considerably more fine grained: first, one needs to define a notion of learnability, since sublinear regret wrt to unrestricted policy classes is impossible *even* for fixed systems ([*] Li et al 2019, a reference we will include). Second, for changing LTV systems, we show that the common policy classes can differ substantially in their performance as opposed to the fixed LTI case. Third, for the convex policies, we show upper and lower bounds which match in their scaling with the system variability, a measure of the “system change” that determines whether sublinear regret is possible. In other words, we characterize *precisely* the optimal worst-case dependence of regret on system change. These results all together validate our claim: the unknown LTV system problem has a clear distinction in learnability from the previously studied known LTV and unknown LTI cases.
>
> Further comments:
> 1. Attaining these results worst case requires several algorithmic innovations, which we detail. We would greatly appreciate it if you clarified for us what it means that the adaptive regret is not sufficiently controlled.
> 2. Regarding assumptions, they are standard in the nonstochastic control literature [3,20,39]; the only unusually strong assumption is internal stability of the system, to which we devote an entire appendix (Appendix A) describing how such assumptions can be weakened considerably via various convex parametrizations.
> 3. The Benchmark in 2.1 does include dependence on the interval, in both the summation and the subscript on the regret term on the LHS.
> 4. The algorithm does not assume that the disturbances w_t are revealed to the learner; this is made clear on lines 116-118. In fact, a key technical challenge to our algorithm is recovering estimates of hat{w}_t of w_t. In fact, if w_t’s were revealed, one can show that one can obtain sublinear regret *without* the system variability term via a bandit convex optimizations strategy, so the fact that w_t’s are not revealed (i.e. the system is unknown and only state is observed) is crucial to the separations uncovered in this paper.
> 5. As explained in the discussion surrounding our first theorem (Thm 2.1), there are three different policy classes of interest. The first two are convex, but the third, which uses feedback, is not. The paper is split naturally into the study of convex parametrizations (DRC and DAC), and feedback parametrizations. The differences in convexity necessitate different techniques.
>
> [3] 	Naman Agarwal, Brian Bullins, Elad Hazan, Sham Kakade, and Karan Singh. Online control with adversarial disturbances. In International Conference on Machine Learning, pages 111– 119, 2019.
>
> [20]	Elad Hazan, Sham Kakade, and Karan Singh. The nonstochastic control problem. In Proceedings of the 31st International Conference on Algorithmic Learning Theory, pages 408–421. PMLR, 2020.
>
> [39]	Max Simchowitz, Karan Singh, and Elad Hazan. Improper learning for non-stochastic control, 2020. In Conference in Learning Theory, pages 3320-3436, 2020.
>
> [*]	Yingying Li, Xin Chen, and Na Li. Online optimal control with linear dynamics and predictions: Algorithms and regret analysis. In NeurIPS, pages 14858–14870, 2019.

---

### Decision · Program_Chairs · 2021-09-27

**Decision:**

Accept (Poster)

**Comment:**

Three of the four reviews were quite positive.  There was one negative review, and I think the authors response was adequate to address concerns raised.